# Growing Tiny Networks: Spotting Expressivity Bottlenecks and Fixing Them Optimally

**Manon Verbockhaven, Théo Rudkiewicz, Sylvain Chevallier, Guillaume Charpiat**
*TAU team, LISN, Université Paris-Saclay, CNRS, Inria, 91405, Orsay, France*
`firstname.name@inria.fr`

**Reviewed on OpenReview:** *https://openreview.net/forum?id=hbtG6s6e7r*

## Abstract

Machine learning tasks are generally formulated as optimization problems, where one searches for an optimal function within a certain functional space. In practice, parameterized functional spaces are considered, in order to be able to perform gradient descent. Typically, a neural network architecture is chosen and fixed, and its parameters (connection weights) are optimized, yielding an architecture-dependent result. This way of proceeding however forces the evolution of the function during training to lie within the realm of what is expressible with the chosen architecture, and prevents any optimization across architectures. Costly architectural hyper-parameter optimization is often performed to compensate for this. Instead, we propose to adapt the architecture on the fly during training. We show that the information about desirable architectural changes, due to expressivity bottlenecks when attempting to follow the functional gradient, can be extracted from backpropagation. To do this, we propose a mathematical definition of expressivity bottlenecks, which enables us to detect, quantify and solve them while training, by adding suitable neurons. Thus, while the standard approach requires large networks, in terms of number of neurons per layer, for expressivity and optimization reasons, we provide tools and properties to develop an architecture starting with a very small number of neurons. As a proof of concept, we show results on the CIFAR dataset, matching large neural network accuracy, with competitive training time, while removing the need for standard architectural hyper-parameter search.

## 1 Introduction

**Issues with the fixed-architecture paradigm.** Universal approximation theorems such as (Hornik et al., 1989; Cybenko, 1989) are historically among the first theoretical results obtained on neural networks, stating the family of neural networks with arbitrary width as a good candidate for a parameterized space of functions to be used in machine learning. However the current common practice in neural network training consists in choosing a fixed architecture, and training it, without any possible architecture modification meanwhile. This inconveniently prevents the direct application of these universal approximation theorems, as expressivity bottlenecks that might arise in a given layer during training will not be able to be fixed. There are two approaches to circumvent this in daily practice. Either one chooses a (very) large width, to be sure to avoid expressivity and optimization issues (Hanin & Rolnick, 2019b; Raghu et al., 2017), to the cost of extra computational power consumption for training and applying such big models; to mitigate this cost, model reduction techniques are often used afterwards, using pruning, tensor factorization, quantization (Louizos et al., 2017) or distillation (Hinton et al., 2015). Or one tries different architectures and keeps the most suitable one (in terms of performance-size compromise for instance), which multiplies the computational burden by the number of trials. This latter approach relates to the Auto-DeepLearning field (Liu et al., 2020), where different exploration strategies over the space of architecture hyper-parameters (among other ones) have been tested, including reinforcement learning (Baker et al., 2017; Zoph & Le, 2016), Bayesian optimization techniques (Mendoza et al., 2016), and evolutionary approaches (Miller et al., 1989; Stanley et al., 2009; Miikkulainen et al., 2017; Bennet et al., 2021), that all rely on random tries and consequently

take time for exploration. Within that line, Net2Net (Chen et al., 2015), AdaptNet (Yang et al., 2018) and MorphNet (Gordon et al., 2018) propose different strategies to explore possible variations of a given architecture, possibly guided by model size constraints. Instead, we aim at providing a way to locate precisely expressivity bottlenecks in a trained network, which might speed up neural architecture search significantly. Moreover, based on such observations, we aim at modifying the architecture *on the fly* during training, in a single run (no re-training), using first-order derivatives only, while avoiding neuron redundancy. Related work on architecture adaptation while training includes probabilistic edges (Liu et al., 2019) or sparsifying priors (Wolinski et al., 2020). Yet the training is done on the largest architecture allowed, which is resource-consuming. On the opposite we aim at starting from the simplest architecture possible.

**Optimization properties.** An important reason for common practice to choose wide architectures is the associated optimization properties: sufficiently larger networks are proved theoretically and shown empirically to be better optimized than small ones (Jacot et al., 2018). Typically, small networks exhibit issues with spurious local minima, while wide ones find good nearly-global minima. One of our goals is to train small networks without suffering from such optimization difficulties.

**Neural architecture growth.** A related line of work consists in growing networks neuron by neuron, by iteratively estimating the best possible neurons to add, according to a certain criterion. For instance, approaches such as (Wu et al., 2019) or Firefly (Wu et al., 2020) aim at escaping local minima by adding neurons that minimize the loss under neighborhood constraints. These neurons are found by gradient descent or by solving quadratic problems involving second-order derivatives. Other approaches (Causse et al., 2019; Bashtova et al., 2022), including GradMax (Evci et al., 2022), seek to minimize the loss as fast as possible and involve another quadratic problem. However the neurons added by these approaches are possibly redundant with existing neurons, especially if one does not wait for training convergence to a local minimum (which is time consuming) before adding neurons, therefore producing larger-than-needed architectures.

**Redundancy.** To our knowledge, the only approach tackling redundancy in neural architecture growth adds random neurons that are orthogonal in some sense to the ones already present (Maile et al., 2022). More precisely, the new neurons are picked within the *kernel* (preimage of $\{0\}$) of an application describing already existing neurons. Two such applications are proposed, respectively the matrix of fan-in weights and the pre-activation matrix, yielding two different notions of orthogonality. The latter formulation is close to the one of GradMax, in that both study first-order loss variations and use the same pre-activation matrix, with an important difference though: GradMax optimally decreases the loss without caring about redundancy, while the other one avoids redundancy but picks random directions instead of optimal ones. In this paper we bridge the gap between these two approaches, picking optimal directions that avoid redundancy in the pre-activation space.

**Notions of expressivity.** Several concepts of expressivity or complexity exist in the Machine Learning literature, ranging from Vapnik-Chervonenkis dimension (Vapnik & Chervonenkis, 1971) and Rademacher complexity (Koltchinskii, 2001) to the number of pieces in a piecewise affine function (as networks with ReLU activations are) (Serra et al., 2018; Hanin & Rolnick, 2019a). Bottlenecks have been also studied from the point of view of Information Theory, through mutual information between the activities of different layers (Tishby & Zaslavsky, 2015; Dai et al., 2018); this quantity is difficult to estimate though. Also relevant and from Information Theory, the Minimum Description Length paradigm (Rissanen, 1978; Grünwald & Roos, 2019) and Kolmogorov complexity (Kolmogorov, 1965; Li et al., 2008) enable to define trade-offs between performance and model complexity.

In this article, we aim at measuring lacks of expressivity as the difference between what the backpropagation asks for and what can be done by a small parameter update (such as a gradient step), that is, between the desired variation for each activation in each layer (for each sample) and the best one that can be realized by a parameter update. Intuitively, differences arise when a layer does not have sufficient expressive power to realize the desired variation. Our main contributions are that we:

- adopt a functional analysis perspective on gradient descent in neural networks, advocating to follow the functional gradient. We not only optimize the weights of the current architecture but also

dynamically adjust the architecture itself to progress towards a suitable parameterized functional space. This approach mitigates optimization challenges like local minima that are due to thin architectures;

- properly define and quantify the concept of expressivity bottlenecks, both globally at the neural network output and locally at individual layers, in a computationally accessible manner. This methodology enables to localize expressivity bottlenecks within a neural network;

- formally define as a quadratic problem the best possible neurons to add to a given layer to decrease lacks of expressivity ; solve it and compute the associated expressivity gain;

- use a naive strategy on the basis of our approach to expand a neural network while maintaining competitive computational complexity and performance compared to large and well-known model or other NAS search methods.

## 2 Main concepts

### 2.1 Notations

Let $\mathcal{F}$ be a functional space, e.g. $L_2(\mathbb{R}^p \to \mathbb{R}^d)$, and a loss function $\mathcal{L} : \mathcal{F} \to \mathbb{R}^+$ defined on it, of the form $\mathcal{L}(f) = \mathbb{E}_{(\boldsymbol{x},\boldsymbol{y})\sim\mathcal{D}}\left[\ell(f(\boldsymbol{x}),\boldsymbol{y})\right]$, where $\ell$ is the per-sample loss, assumed to be differentiable, and where $\mathcal{D}$ is the sample distribution, from which the dataset $\{(\boldsymbol{x}_1,\boldsymbol{y}_1),...,(\boldsymbol{x}_N,\boldsymbol{y}_N)\}$ is sampled, with $\boldsymbol{x}_i \in \mathbb{R}^p$ and $\boldsymbol{y}_i \in \mathbb{R}^d$.

For the sake of simplicity we consider a feedforward neural network $f_\theta : \mathbb{R}^p \to \mathbb{R}^d$ with $L$ hidden layers, each of which consisting of an affine layer with weights $\boldsymbol{W}_l$ followed by a differentiable activation function $\sigma_l$ which satisfies $\sigma_l(0) = 0$. The network parameters are then $\theta := (\boldsymbol{W}_l)_{l=1...L}$. The network iteratively computes:

$$\boldsymbol{b}_0(\boldsymbol{x}) = \begin{pmatrix} \boldsymbol{x} \\ 1 \end{pmatrix}$$

$$\forall l \in [1, L], \quad \begin{cases} \boldsymbol{a}_l(\boldsymbol{x}) = \boldsymbol{W}_l \, \boldsymbol{b}_{l-1}(\boldsymbol{x}) \\ \boldsymbol{b}_l(\boldsymbol{x}) = \begin{pmatrix} \sigma_l(\boldsymbol{a}_l(\boldsymbol{x})) \\ 1 \end{pmatrix} \end{cases}$$

$$f_\theta(\boldsymbol{x}) = \sigma_L(\boldsymbol{a}_L(\boldsymbol{x}))$$

Figure 1: Notations

To any vector-valued function noted $\boldsymbol{t}(\boldsymbol{x})$ and any batch of inputs $\boldsymbol{X} := [\boldsymbol{x}_1, ..., \boldsymbol{x}_n]$, we associate the concatenated matrix $\boldsymbol{T}(\boldsymbol{X}) := \begin{pmatrix} \boldsymbol{t}(\boldsymbol{x}_1) & ... & \boldsymbol{t}(\boldsymbol{x}_n) \end{pmatrix} \in \mathbb{R}^{|\boldsymbol{t}(.)|\times n}$. The matrices of pre-activation and post-activation activities at layer $l$ over a minibatch $\boldsymbol{X}$ are thus respectively: $\boldsymbol{A}_l(\boldsymbol{X}) = \begin{pmatrix} \boldsymbol{a}_l(\boldsymbol{x}_1) & ... & \boldsymbol{a}_l(\boldsymbol{x}_n) \end{pmatrix}$ and $\boldsymbol{B}_l(\boldsymbol{X}) = \begin{pmatrix} \boldsymbol{b}_l(\boldsymbol{x}_1) & ... & \boldsymbol{b}_l(\boldsymbol{x}_n) \end{pmatrix}$.

*NB: convolutions can also be considered, with appropriate representations (cf matrix $\boldsymbol{B}_l^c(\boldsymbol{x})$ in C.1).*

### 2.2 Approach

**Functional gradient descent.** We take a functional perspective on the use of neural networks. Ideally in a machine learning task, one would search for a function $f : \mathbb{R}^p \to \mathbb{R}^d$ that minimizes the loss $\mathcal{L}$ by gradient descent: $\frac{\partial f}{\partial t} = -\nabla_f \mathcal{L}(f)$ for some metric on the functional space $\mathcal{F}$ (typically, $L_2(\mathbb{R}^p \to \mathbb{R}^d)$), where $\nabla_f$ denotes the functional gradient and $t$ denotes the evolution time of the gradient descent. The descent direction $\boldsymbol{v}_{\text{goal}} := -\nabla_f \mathcal{L}(f)$ is a function of the same type as $f$ and whose value at $\boldsymbol{x}$ is easily computable as $\boldsymbol{v}_{\text{goal}}(\boldsymbol{x}) = -\left(\nabla_f \mathcal{L}(f)\right)(\boldsymbol{x}) = -\nabla_{\boldsymbol{u}}\ell(\boldsymbol{u},\boldsymbol{y}(\boldsymbol{x}))\big|_{\boldsymbol{u}=f(\boldsymbol{x})}$ (see Appendix A.1 for more details). This direction $\boldsymbol{v}_{\text{goal}}$ is the best infinitesimal variation in $\mathcal{F}$ to add to the output ($u = f(\boldsymbol{x})$) to decrease the loss $\mathcal{L}$, regardless of the parametrization of $f$.

**Parametric gradient descent reminder.** However in practice, to represent functions and to compute gradients, the infinite-dimensional functional space $\mathcal{F}$ has to be replaced with a finite-dimensional parametric space of functions, which is usually done by choosing a particular neural network architecture $\mathcal{A}$ with weights $\theta \in \Theta_{\mathcal{A}}$. The associated parametric search space $\mathcal{F}_{\mathcal{A}}$ then consists of all possible functions $f_\theta$ that can be represented with such a network for any parameter value $\theta$. Under standard weak assumptions (see Appendix A.2), the gradient descent is of the form:

$$\frac{\partial \theta}{\partial t} = -\nabla_\theta \mathcal{L}(f_\theta) = - \mathop{\mathbb{E}}_{(\boldsymbol{x}, \boldsymbol{y}) \sim \mathcal{D}} \left[ \nabla_\theta \ell(f_\theta(\boldsymbol{x}), \boldsymbol{y}) \right]. \tag{1}$$

Using the chain rule (on $\frac{\partial f_\theta}{\partial t}$ then on $\nabla_\theta \ell(f_\theta(\boldsymbol{x}), \boldsymbol{y})$), these parameter updates yield a functional evolution:

$$\boldsymbol{v}_{\mathrm{GD}} := \frac{\partial f_\theta}{\partial t} = \frac{\partial f_\theta}{\partial \theta} \frac{\partial \theta}{\partial t} = \frac{\partial f_\theta}{\partial \theta} \mathop{\mathbb{E}}_{(\boldsymbol{x}, \boldsymbol{y}) \sim \mathcal{D}} \left[ \frac{\partial f_\theta}{\partial \theta}^T (\boldsymbol{x}) \, \boldsymbol{v}_{\mathrm{goal}}(\boldsymbol{x}) \right] \tag{2}$$

which significantly differs from the original functional gradient descent. We will aim to augment the neural network architecture so that the parametric gradient descent can get closer to the functional one.

**Optimal move direction.** We name $\mathcal{T}_{\mathcal{A}}^{f_\theta}$, or just $\mathcal{T}_{\mathcal{A}}$, the tangent space of $\mathcal{F}_{\mathcal{A}}$ at $f_\theta$, that is, the set of all possible infinitesimal variations around $f_\theta$ under small parameter variations:

$$\mathcal{T}_{\mathcal{A}}^{f_\theta} := \left\{ \frac{\partial f_\theta}{\partial \theta} \, \delta\theta \;\middle|\; \text{s.t. } \delta\theta \in \Theta_{\mathcal{A}} \right\}$$

This linear functional[1] space is a first-order approximation of the neighborhood of $f_\theta$ within $\mathcal{F}_{\mathcal{A}}$. The direction $\boldsymbol{v}_{\mathrm{GD}}$ obtained above by gradient descent is actually not the best one to consider within $\mathcal{T}_{\mathcal{A}}$. Indeed, the best move $\boldsymbol{v}^*$ would be the orthogonal projection of the desired direction $\boldsymbol{v}_{\mathrm{goal}} := -\nabla_{f_\theta} \mathcal{L}(f_\theta)$ onto $\mathcal{T}_{\mathcal{A}}$. This projection is what a (generalization of the notion of) natural gradient would compute (Ollivier, 2017).

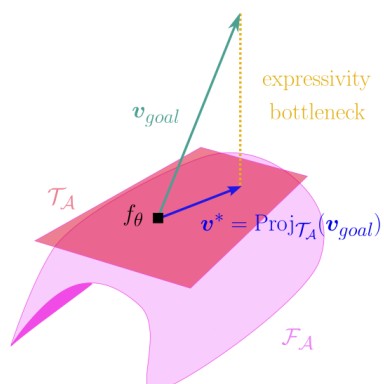

Figure 2: Expressivity bottleneck

Indeed, the parameter variation $\delta\theta^*$ associated to the functional variation $\boldsymbol{v}^* = \frac{\partial f_\theta}{\partial \theta} \delta\theta^*$ is the gradient $-\nabla_\theta^{\mathcal{T}_{\mathcal{A}}} \mathcal{L}(f_\theta)$ of $\mathcal{L} \circ f_\theta$ w.r.t. parameters $\theta$ when considering the $L_2$ metric on *functional* variations $\|\frac{\partial f_\theta}{\partial \theta} \delta\theta\|_{L_2(\mathcal{T}_{\mathcal{A}})}$, not to be confused with the usual gradient $\nabla_\theta \mathcal{L}(f_\theta)$, based on the $L_2$ metric on *parameter* variations $\|\delta\theta\|_{L_2(\mathbb{R}^{|\Theta_{\mathcal{A}}|})}$. This can be seen in a proximal formulation as:

$$\boldsymbol{v}^* = \operatorname*{arg\,min}_{\boldsymbol{v} \in \mathcal{T}_{\mathcal{A}}} \|\boldsymbol{v} - \boldsymbol{v}_{\mathrm{goal}}\|^2 = \operatorname*{arg\,min}_{\boldsymbol{v} \in \mathcal{T}_{\mathcal{A}}} \left\{ D_f \mathcal{L}(f)(\boldsymbol{v}) + \frac{1}{2} \|\boldsymbol{v}\|^2 \right\} \tag{3}$$

where $D$ is the directional derivative (see details in Appendix A.3), or equivalently as:

$$\delta\theta^* = \operatorname*{arg\,min}_{\delta\theta \in \Theta_{\mathcal{A}}} \left\| \frac{\partial f_\theta}{\partial \theta} \delta\theta - \boldsymbol{v}_{\mathrm{goal}} \right\|^2 = \operatorname*{arg\,min}_{\delta\theta \in \Theta_{\mathcal{A}}} \left\{ D_\theta \mathcal{L}(f_\theta)(\delta\theta) + \frac{1}{2} \left\| \frac{\partial f_\theta}{\partial \theta} \delta\theta \right\|^2 \right\} =: -\nabla_\theta^{\mathcal{T}_{\mathcal{A}}} \mathcal{L}(f_\theta).$$

**Lack of expressivity.** When $\boldsymbol{v}_{\mathrm{goal}}$ does not belong to the reachable subspace $\mathcal{T}_{\mathcal{A}}$, there is a lack of expressivity, that is, the parametric space $\mathcal{A}$ is not rich enough to follow the ideal functional gradient descent. This happens frequently with small neural networks (see Appendix A.4 for an example). The expressivity bottleneck is then quantified as the distance $\|\boldsymbol{v}^* - \boldsymbol{v}_{\mathrm{goal}}\|$ between the functional gradient $\boldsymbol{v}_{\mathrm{goal}}$ and the optimal functional move $\boldsymbol{v}^*$ given the architecture $\mathcal{A}$ (in the sense of Eq. 3).

---

[1] $\frac{\partial f_\theta}{\partial \theta} \delta\theta : \mathbb{R}^p \to \mathbb{R}^d, \; \boldsymbol{x} \mapsto \frac{\partial f_\theta(\boldsymbol{x})}{\partial \theta} \delta\theta$

### 2.3 Generalizing to all layers

**Ideal updates.** The same reasoning can be applied to the pre-activations $\boldsymbol{a}_l$ at each layer $l$, seen as functions $\boldsymbol{a}_l : \boldsymbol{x} \in \mathbb{R}^p \mapsto \boldsymbol{a}_l(\boldsymbol{x}) \in \mathbb{R}^{d_l}$ defined over the input space of the neural network. The optimal parameter update for a given layer $l$ then follows the projection of the desired update $-\nabla_{\boldsymbol{a}_l}\mathcal{L}(f_\theta)$ of the pre-activation functions $\boldsymbol{a}_l$ onto the linear subspace $\mathcal{T}_{\mathcal{A}}^{\boldsymbol{a}_l}$ of pre-activation variations that are possible with the architecture, as we will detail now.

Given a sample $(\boldsymbol{x}, \boldsymbol{y}) \in \mathcal{D}$, standard backpropagation already iteratively computes $\boldsymbol{v}_{\text{goal}}^l(\boldsymbol{x}) := -(\nabla_{\boldsymbol{a}_l}\mathcal{L}(f_\theta))(\boldsymbol{x}) = -\nabla_{\boldsymbol{u}}\ell(\sigma_L(\boldsymbol{W}_L\,\sigma_{L-1}(\boldsymbol{W}_{L-1}\ldots\sigma_l(\boldsymbol{u}))), \boldsymbol{y})|_{\boldsymbol{u}=\boldsymbol{a}_l(\boldsymbol{x})}$, which is the derivative of the loss $\ell(f_\theta(\boldsymbol{x}), \boldsymbol{y})$ with respect to the pre-activations $\boldsymbol{u} = \boldsymbol{a}_l(\boldsymbol{x})$ of each layer. This is usually performed in order to compute the gradients w.r.t. model parameters $\boldsymbol{W}_l$, as $\nabla_{\boldsymbol{W}_l}\ell(f_\theta(\boldsymbol{x}), \boldsymbol{y}) = \frac{\partial \boldsymbol{a}_l(\boldsymbol{x})}{\partial W_l}\,\nabla_{\boldsymbol{a}_l}\ell(f_\theta(\boldsymbol{x}), \boldsymbol{y})$.

$\boldsymbol{v}_{\text{goal}}^l(\boldsymbol{x}) := -(\nabla_{\boldsymbol{a}_l}\mathcal{L}(f_\theta))(\boldsymbol{x})$ indicates the direction in which one would like to change the layer pre-activations $\boldsymbol{a}_l(\boldsymbol{x})$ in order to decrease the loss at point $\boldsymbol{x}$. However, given a minibatch of points $(\boldsymbol{x}_i)$, most of the time no parameter move $\delta\theta$ is able to induce this progression for each $\boldsymbol{x}_i$ simultaneously, because the $\theta$-parameterized family of functions $\boldsymbol{a}_l$ is not expressive enough.

**Activity update resulting from a parameter change.** Given a subset of parameters $\tilde{\theta}$ (such as the ones specific to a layer: $\tilde{\theta} = \boldsymbol{W}_l$), and an incremental direction $\delta\tilde{\theta}$ to update these parameters (e.g. the one resulting from a gradient descent: $\delta\tilde{\theta} = -\sum_{(\boldsymbol{x}, \boldsymbol{y})\in\text{minibatch}} \nabla_{\tilde{\theta}}\ell(f_\theta(\boldsymbol{x}), \boldsymbol{y})$ ), the impact of the parameter update $\gamma\delta\tilde{\theta}$ on the pre-activations $\boldsymbol{a}_l$ at layer $l$ at order 1 in $\gamma$, where $\gamma$ an amplitude factor, is as $\boldsymbol{v}^l(\boldsymbol{x}, \gamma\delta\tilde{\theta}) := \gamma\frac{\partial \boldsymbol{a}_l(\boldsymbol{x})}{\partial\tilde{\theta}}\,\delta\tilde{\theta}$. The factor $\gamma$ is the analogue of the learning rate and is considered to be small.

## 3 Expressivity bottlenecks

We now quantify expressivity bottlenecks at any layer $l$ as the distance between the desired activity update $\boldsymbol{v}_{\text{goal}}^l(.)$ and the best realizable one $\boldsymbol{v}^l(.)$ (cf. Figure 2):

**Definition 3.1** (Lack of expressivity). *For a neural network $f_\theta$ and a minibatch of points $\boldsymbol{X} = \{(\boldsymbol{x}_i, \boldsymbol{y}_i)\}_{i=1}^n$, we define the lack of expressivity at layer $l$ as how far the desired activity update $\boldsymbol{V}_{goal}^l = (\boldsymbol{v}_{goal}^l(\boldsymbol{x}_1), \boldsymbol{v}_{goal}^l(\boldsymbol{x}_2), \ldots)$ is from the closest possible activity update $\boldsymbol{V}^l = (\boldsymbol{v}^l(\boldsymbol{x}_1), \boldsymbol{v}^l(\boldsymbol{x}_2), \ldots)$ realizable by a parameter change $\delta\theta$:*

$$\Psi^l := \min_{\boldsymbol{v}^l \in \mathcal{T}_{\mathcal{A}}^{\boldsymbol{a}_l}} \frac{1}{n}\sum_{i=1}^n \left\| \boldsymbol{v}^l(\boldsymbol{x}_i) - \boldsymbol{v}_{\text{goal}}^l(\boldsymbol{x}_i) \right\|^2 = \min_{\delta\theta} \frac{1}{n}\left\| \boldsymbol{V}^l(\boldsymbol{X}, \delta\theta) - \boldsymbol{V}_{\text{goal}}^l(\boldsymbol{X}) \right\|_{\text{Tr}}^2 \tag{4}$$

where $||.||$ stands for the $L_2$ norm, $||.||_{\text{Tr}}$ for the Frobenius norm, and $\boldsymbol{V}^l(\boldsymbol{X}, \delta\theta)$ is the activity update resulting from parameter change $\delta\theta$ as defined in previous section. In the two following parts we fix the minibatch $\boldsymbol{X}$ and simplify notations accordingly by removing the dependency on $\boldsymbol{X}$. Proofs of this section are deferred to Appendix C.

### 3.1 Best move without modifying the architecture of the network

Let $\delta\boldsymbol{W}_l^*$ be the solution of Equation (4) when the parameter variation $\delta\theta$ is restricted to involve only layer $l$ parameters, i.e. $\boldsymbol{W}_l$. This move is sub-optimal in that it does not result from an update of all architecture parameters but only of the current layer ones. In this case the activity update simplifies as $\boldsymbol{V}^l(\boldsymbol{W}_l) = \delta\boldsymbol{W}_l\,\boldsymbol{B}_{l-1}$ and the problem becomes:

$$\delta\boldsymbol{W}_l^* = \underset{\delta\boldsymbol{W}}{\arg\min} \frac{1}{n}\left\| \boldsymbol{V}^l(\delta\boldsymbol{W}) - \boldsymbol{V}_{\text{goal}}^l \right\|_{\text{Tr}}^2. \tag{5}$$

**Proposition 3.1.** *The solution of Problem equation 5 is: $\delta\boldsymbol{W}_l^* = \frac{1}{n}\boldsymbol{V}_{goal}^l\boldsymbol{B}_{l-1}^T(\frac{1}{n}\boldsymbol{B}_{l-1}\boldsymbol{B}_{l-1}^T)^+$ where $P^+$ denotes the pseudoinverse of matrix $P$.*

This update $\delta\boldsymbol{W}_l^*$ is not equivalent to the usual gradient descent update, whose form is $\delta\boldsymbol{W}_l^{\text{GD}} \propto \boldsymbol{V}_{\text{goal}}^l\boldsymbol{B}_{l-1}^T$. In fact the associated activity variation, $\delta\boldsymbol{W}_l^*\boldsymbol{B}_{l-1}$, is the projection of $\boldsymbol{V}_{\text{goal}}^l$ on the post-activation matrix

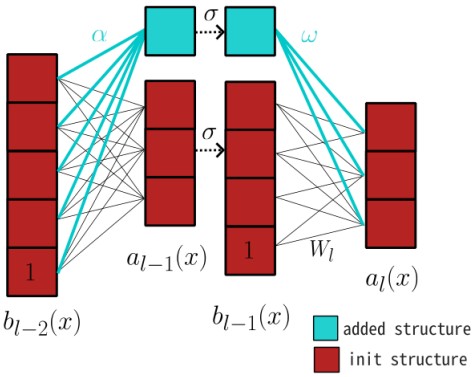

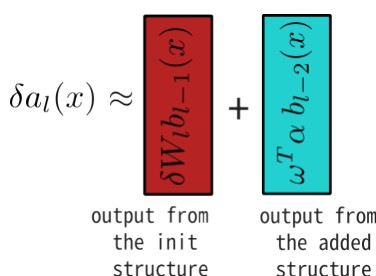

Figure 3: Adding one neuron to layer $l$ in cyan ($K = 1$), with connections in cyan. Here, $\boldsymbol{\alpha} \in \mathrm{R}^5$ and $\boldsymbol{\omega} \in \mathrm{R}^3$.

Figure 4: Sum of functional moves

of layer $l-1$, that is to say onto the span of all possible post-activation directions, through the projector $\frac{1}{n}\boldsymbol{B}_{l-1}^T \left(\frac{1}{n}\boldsymbol{B}_{l-1}\boldsymbol{B}_{l-1}^T\right)^+ \boldsymbol{B}_{l-1}$. To increase expressivity if needed, we will aim at increasing this span with the most useful directions to close the gap between this best update and the desired one. Note that the update $\delta\boldsymbol{W}_l^*$ consists of a standard gradient ($\boldsymbol{V}_{\text{goal}}^l \boldsymbol{B}_{l-1}^T$) followed by a change of metric in the pre-activation space as an application of $\left(\frac{1}{n}\boldsymbol{B}_{l-1}\boldsymbol{B}_{l-1}^T\right)^+$, a structure which is reminiscent of the natural gradient.

### 3.2 Reducing expressivity bottleneck by modifying the architecture

To get as close as possible to $\boldsymbol{V}_{\text{goal}}^l$ and to increase the expressive power of the current neural network, we modify each layer of its structure. At layer $l-1$, we add $K$ neurons $n_1, ..., n_K$ with input weights $\boldsymbol{\alpha}_1, ..., \boldsymbol{\alpha}_k$ and output weights $\boldsymbol{\omega}_1, ..., \boldsymbol{\omega}_K$ (cf. Figure 3). We have the following expansions by concatenation: $\boldsymbol{W}_{l-1}^T \leftarrow \begin{pmatrix} \boldsymbol{W}_{l-1}^T & \boldsymbol{\alpha}_1 & ... & \boldsymbol{\alpha}_K \end{pmatrix}$ and $\boldsymbol{W}_l \leftarrow \begin{pmatrix} \boldsymbol{W}_l & \boldsymbol{\omega}_1 & ... & \boldsymbol{\omega}_K \end{pmatrix}$. We note this architecture modification $\theta \leftarrow \theta \oplus \theta_{\leftrightarrow}^K$ where $\oplus$ is the concatenation sign and $\theta_{\leftrightarrow}^K := (\boldsymbol{\alpha}_k, \boldsymbol{\omega}_k)_{k=1}^K$ are the $K$ added neurons.

The added neurons could be chosen randomly, as in usual neural network initialization, but this would not yield any guarantee regarding the impact on the system loss. Another possibility would be to set either input weights $(\boldsymbol{\alpha}_k)_{k=1}^K$ or output weights $(\boldsymbol{\omega}_k)_{k=1}^K$ to 0, so that the function $f_\theta(.)$ would not be modified, while its gradient w.r.t. $\theta$ would be enriched from the new parameters. Another option is to solve an optimization problem as in the previous section with the modified structure $\theta \leftarrow \theta \oplus \theta_{\leftrightarrow}^K$ and jointly search for both the optimal new parameters $\theta_{\leftrightarrow}^K$ and the optimal variation $\delta\boldsymbol{W}$ of the old ones:

$$\underset{\theta_{\leftrightarrow}^K, \, \delta\boldsymbol{W}}{\arg\min} \frac{1}{n} \left\| \boldsymbol{V}^l(\delta\boldsymbol{W} \oplus \theta_{\leftrightarrow}^K) - \boldsymbol{V}_{\text{goal}}^l \right\|_{\text{Tr}}^2 . \tag{6}$$

As shown in Figure 4, the displacement $\boldsymbol{V}^l$ at layer $l$ is actually a sum of the moves induced by the neurons already present ($\delta\boldsymbol{W}$) and by the added neurons ($\theta_{\leftrightarrow}^K$). Our problem rewrites as :

$$\underset{\theta_{\leftrightarrow}^K, \, \delta\boldsymbol{W}}{\arg\min} \frac{1}{n} \left\| \boldsymbol{V}^l(\theta_{\leftrightarrow}^K) + \boldsymbol{V}^l(\delta\boldsymbol{W}) - \boldsymbol{V}_{\text{goal}}^l \right\|_{\text{Tr}}^2 \tag{7}$$

with $\boldsymbol{v}^l(\boldsymbol{x}, \theta_{\leftrightarrow}^K) := \sum_{k=1}^K \boldsymbol{\omega}_k \left(\boldsymbol{b}_{l-2}(\boldsymbol{x})^T \boldsymbol{\alpha}_k\right)$ (see A.5). We choose $\delta\boldsymbol{W}$ as the best move of already-existing parameters as defined in Proposition 3.1 and we note $\boldsymbol{V}_{\text{goal}_{proj}}^l := \boldsymbol{V}_{\text{goal}}^l - \boldsymbol{V}^l(\delta\boldsymbol{W}^*)$. We are looking for the solution $\left(K^*, \theta_{\leftrightarrow}^{K*}\right)$ of the optimization problem :

$$\underset{K, \, \theta_{\leftrightarrow}^K}{\arg\min} \frac{1}{n} \left\| \boldsymbol{V}^l(\theta_{\leftrightarrow}^K) - \boldsymbol{V}_{\text{goal}_{proj}}^l \right\|_{\text{Tr}}^2 . \tag{8}$$

One should note that Problems (7) and (8) are generally not equivalent, though similar (cf. C.4).
This quadratic optimization problem can be solved thanks to the low-rank matrix approximation theorem

(Eckart & Young, 1936), using matrices $\boldsymbol{N} := \frac{1}{n}\boldsymbol{B}_{l-2}\left(\boldsymbol{V}^l_{\mathrm{goal}_{proj}}\right)^T$ and $\boldsymbol{S} := \frac{1}{n}\boldsymbol{B}_{l-2}\boldsymbol{B}^T_{l-2}$. As $\boldsymbol{S}$ is positive semi-definite, let its SVD be $\boldsymbol{S} = \boldsymbol{O}\boldsymbol{\Sigma}\boldsymbol{O}^T$, and define $\boldsymbol{S}^{-\frac{1}{2}} := \boldsymbol{O}\sqrt{\boldsymbol{\Sigma}}^{-1}\boldsymbol{O}^T$, with the convention that the inverse of 0 eigenvalues is 0. Finally, consider the SVD of matrix $\boldsymbol{S}^{-\frac{1}{2}}\boldsymbol{N} = \sum_{k=1}^R \lambda_k \boldsymbol{u}_k \boldsymbol{v}_k^T$, where $R$ is the rank of the matrix $\boldsymbol{S}^{-\frac{1}{2}}\boldsymbol{N}$. Then:

**Proposition 3.2.** *The solution of Problem (8) is:*

- *optimal number of neurons:* $K^* = R$
- *their optimal weights:* $\theta^{K^*}_{\leftrightarrow} = (\boldsymbol{\alpha}^*_k, \boldsymbol{\omega}^*_k)_{k=1}^{K^*} = \left(\sqrt{\lambda_k}\boldsymbol{S}^{-\frac{1}{2}}\boldsymbol{u}_k,\ \sqrt{\lambda_k}\boldsymbol{v}_k\right)_{k=1}^{K^*}$

*Moreover for any number of neurons $K \leqslant R$, and associated optimal weights $\theta^{K,*}_{\leftrightarrow}$ consisting of the first $K$ neurons of $\theta^{K^*}_{\leftrightarrow}$, the expressivity gain can be quantified very simply as a function of the singular values $\lambda_k$:*

$$\Psi^l_{\theta \oplus \theta^{K,*}_{\leftrightarrow}} = \Psi^l_\theta - \sum_{k=1}^K \lambda_k^2 \tag{9}$$

*where $\Psi^l_\theta$ is the expressivity bottleneck (defined in Eq. (4)). For convolutional layers instead of fully-connected ones, Equation (9) becomes an inequality ($\leqslant$).*

In practice before adding new neurons $(\alpha, \omega)$, we multiply them by an amplitude factor $\gamma$ found by a simple line search (see Appendix E.3), i.e. we add $(\sqrt{\gamma}\alpha, \sqrt{\gamma}\omega)$. The addition of each neuron $k$ has an impact on the bottleneck of the order of $\gamma\lambda_k^2$ provided $\gamma$ is small. We observe the same phenomenon with the loss as stated in the next proposition :

**Proposition 3.3.** *For $\gamma > 0$, solving (8) using $\boldsymbol{V}_{goal_{proj}} = \boldsymbol{V}_{goal} - \boldsymbol{V}(\gamma\delta\boldsymbol{W}^*)$ is equivalent to minimizing the loss $\mathcal{L}$ at order one in $\gamma\boldsymbol{V}^l$. Furthermore, performing an architecture update with $\gamma\delta\boldsymbol{W}^*$ (5) and a neuron addition with $\gamma\theta^{K,*}_{\leftrightarrow}$ (3.2) has an impact on the loss at first order in $\gamma$ as :*

$$\mathcal{L}(f_{\theta \oplus \gamma\theta^{K,*}_{\leftrightarrow}}) \ := \ \frac{1}{n}\sum_{i=1}^n \ell(f_{\theta \oplus \gamma\theta^{K,*}_{\leftrightarrow}}(\boldsymbol{x}_i), \boldsymbol{y}_i) \ = \ \mathcal{L}(f_\theta) - \gamma\left(\sigma'_{l-1}(0)\Delta_{\theta^{K,*}_{\leftrightarrow}} + \Delta_{\delta\boldsymbol{W}^*}\right) + o(\gamma) \tag{10}$$

*with*

$$\Delta_{\theta^{K,*}_{\leftrightarrow}} \ := \ \frac{1}{n}\left\langle \boldsymbol{V}^l_{goal_{proj}}, \boldsymbol{V}^l(\theta^{K,*}_{\leftrightarrow}) \right\rangle_{\mathrm{Tr}} \ = \ \sum_{k=1}^K \lambda_k^2 \tag{11}$$

$$\Delta_{\delta\boldsymbol{W}^*} \ := \ \frac{1}{n}\left\langle \boldsymbol{V}^l_{goal}, \boldsymbol{V}^l(\delta\boldsymbol{W}^*) \right\rangle_{\mathrm{Tr}} \ \geqslant \ 0 \,. \tag{12}$$

The $\lambda_k$ could be used in a selection criterion realizing a trade-off with computational complexity. A selection based on statistical significance of singular values can also be performed. The full algorithm and its complexity are detailed in Appendices E.4 and E.5. We finish this section by some additional propositions and remarks.

**Proposition 3.4.** *If $\boldsymbol{S}$ is positive definite, then solving (8) is equivalent to taking $\boldsymbol{\omega}_k = \boldsymbol{N}\boldsymbol{\alpha}_k$ and finding the $K$ first eigenvectors $\boldsymbol{\alpha}_k$ associated to the $K$ largest eigenvalues $\lambda$ of the generalized eigenvalue problem:*
$$\boldsymbol{N}\boldsymbol{N}^T\boldsymbol{\alpha}_k = \lambda\boldsymbol{S}\boldsymbol{\alpha}_k \,.$$

**Corollary 1.** *For all integers $m, m'$ such that $m + m' \leqslant R$, at order one in $\boldsymbol{V}$, adding $m + m'$ neurons simultaneously according to the previous method is equivalent to adding $m$ neurons then $m'$ neurons by applying successively the previous method twice.*

One should also note that the family $\{\boldsymbol{V}^{l+1}((\boldsymbol{\alpha}_k, \boldsymbol{\omega}_k))\}_{k=1}^K$ of pre-activity variations induced by adding the neurons $\theta^{K,*}_{\leftrightarrow}$ is orthogonal for the trace scalar product. We could say that the added neurons are orthogonal to each other (and to the already-present ones) in that sense. Interestingly, the GradMax method (Evci et al., 2022) also aims at minimizing the loss 10, but without avoiding redundancy (see Appendix B.1 for more details).

# 4 About greedy growth sufficiency and TINY convergence

One might wonder whether a greedy approach on layer growth might get stuck in a non-optimal state. By *greedy* we mean that every neuron added has to decrease the loss. Since in this work we add neurons layer per layer independently, we study here the case of a single hidden layer network, to spot potential layer growth issues. For the sake of simplicity, we consider the task of least square regression towards an explicit continuous target $f^*$, defined on a compact set. That is, we aim at minimizing the loss:

$$\inf_f \sum_{\boldsymbol{x} \in \mathcal{D}} ||f(\boldsymbol{x}) - f^*(\boldsymbol{x})||^2 \tag{13}$$

where $f(\boldsymbol{x})$ is the output of the neural network and $\mathcal{D}$ is the training set. Proofs and supplementary propositions are deferred to Appendix D, in particular D.4 and D.7.

First, if one allows only adding neurons but no modification of already existing ones:

**Proposition 4.1** (Exponential convergence to 0 training error by ReLU neuron additions). *It is possible to decrease the loss exponentially fast with the number $t$ of added neurons, i.e. as $\gamma^t \mathcal{L}(f)$, towards 0 training loss, and this in a greedy way, that is, such that each added neuron decreases the loss. The factor $\gamma$ is $\gamma = 1 - \frac{1}{n^3 d'} \left( \frac{d_m}{d_M} \right)^2$, where $d_m$ and $d_M$ are quantities solely dependent on the dataset geometry, $d'$ is the output dimension of the network, and $n$ is the dataset size.*

In particular, there exists no situation where one would need to add many neurons simultaneously to decrease the loss: it is always feasible with a single neuron.

TINY might get stuck when no correlation between inputs $\boldsymbol{x}_i$ and desired output variations $f^*(\boldsymbol{x}_i) - f(\boldsymbol{x}_i)$ can be found anymore. To prevent this, one can choose an auxiliary method to add neurons in such cases, for instance random neurons (with a line search over their amplitude, cf. Appendix D.3), or locally-optimal neurons found by gradient descent, or solutions of higher-order expressivity bottleneck formulations using further developments of the activation function. We will name *completed-TINY* the completion of TINY by any such auxiliary method.

Now, if we also update already existing weights when adding new neurons, we get a stronger result:

**Proposition 4.2** (Completed-TINY reaches 0 training error in at most $n$ neuron additions). *Under certain assumptions (full batch optimization, updating already existing parameters, and, more technically: polynomial activation function of order $\geqslant n^2$), completed-TINY reaches 0 training error in at most $n$ neuron additions almost surely.*

Hence we see the importance of updating existing parameters on the convergence speed. This optimization protocol is actually the one we follow in practice when training neural networks with TINY (except when comparing with other methods using their protocol).

Note that our approach shares similarity with gradient boosting Friedman (2001) somehow, as we grow the architecture based on the gradient of the loss. Note also that finding the optimal neuron to add is actually NP-hard (Bach, 2017), but that we do not need new neuron optimality to converge to 0 training error.

# 5 Results

## 5.1 Comparison with GradMax on CIFAR-100

The closest growing method to TINY is GradMax (Evci et al. (2022)), as it solves a quadratic problem similar to (8). By construction, the objective of GradMax is to decrease the loss as fast as possible considering an infinitesimal increment of new neurons. The main difference is that GradMax does not take into account the expressivity of the current architecture as TINY does in (8) by projecting $v_{\text{goal}}$. In-depth details about the difference between the GradMax and TINY are provided in Appendix B.1.

In this section, we show on the CIFAR-100 dataset that solving (8) instead of (23) (defined by GradMax) to grow a network using a naive strategy allows better final performance and almost full expressive power.

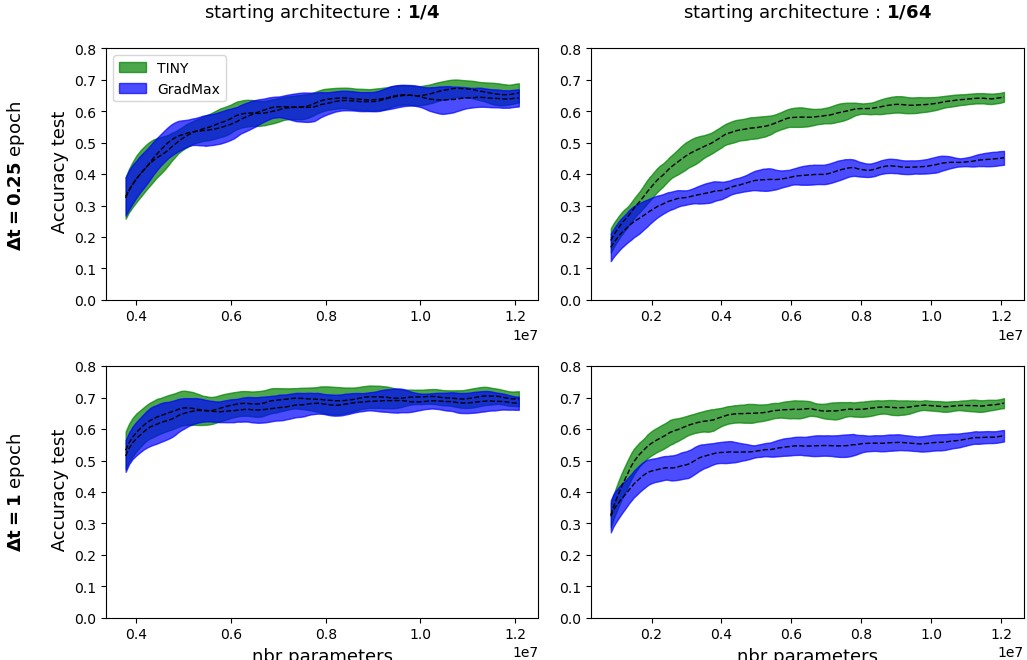

Figure 5: Test accuracy as a function of the number of parameters during architecture growth from ResNet$_s$ to ResNet18. The starting architecture in the left (resp. right) column is ResNet$_{1/4}$ (resp. ResNet$_{1/64}$). The amount of training time $\Delta t$ between consecutive neuron additions in the upper (resp. lower) row is 0.25 (resp. 1) epoch. Statistics over 4 runs are performed for each setting.

To do so, we have re-implemented the GradMax method and mimicked its growing process which consists in increasing the architecture of a thin ResNet18 until it reaches the architecture of the usual ResNet18. This process is described in the pseudo code 1, where two parameters can be chosen : the relative thinness $s$ of the starting architecture, w.r.t. the usual ResNet18 architecture ($s = 1/4$ or $s = 1/64$, cf. Table 3), and the amount of training time between consecutive neuron additions ($\Delta t = 1$ or $\Delta t = 0.25$ epochs). Then the number of parameters and the performance of the growing network are evaluated at regular intervals to plot Figure 5. We note that both methods reach their final accuracy within less than one GPU day, outperforming by far other NAS search methods in their category (cf. Table 5).

Once the models have reached the final architecture ResNet18, they are trained for 250 more epochs (or 500 epochs if they have not converged yet) on the training set. We have summarized the final performance in Table 1. We also added the column *Reference*, which gives the performance of a ResNet18 trained from scratch by usual gradient descent with all its neurons. We do not expect TINY or GradMax to achieve the performance of the reference as its architecture and optimisation process have been optimised for years. The column *Small references* corresponds to the training of the architecture ResNet18$_s$ for $s = 1/4$ and $s = 1/64$ using standard gradient descent without any increase in architecture.

The details of the protocol can be found in the annexes F.1, as well as other technical details such as the dynamic of the training batch size E.2, the normalization of the new neurons E.3, the number of examples used to estimate and solve the expressivity bottleneck E.1 and the complexity of the algorithms E.5. For both methods, all the latter apply so that the main differences between GradMax and TINY in this experiment is the mathematical definition of the new neurons.

For $s = 1/64$, we observe a significant difference in performance between TINY and GradMax methods. While TINY models almost achieve the reference's performance, GradMax remains stuck 10 points below. This suggests that the framework proposed by GradMax is not sufficient to be able to start with an archi-

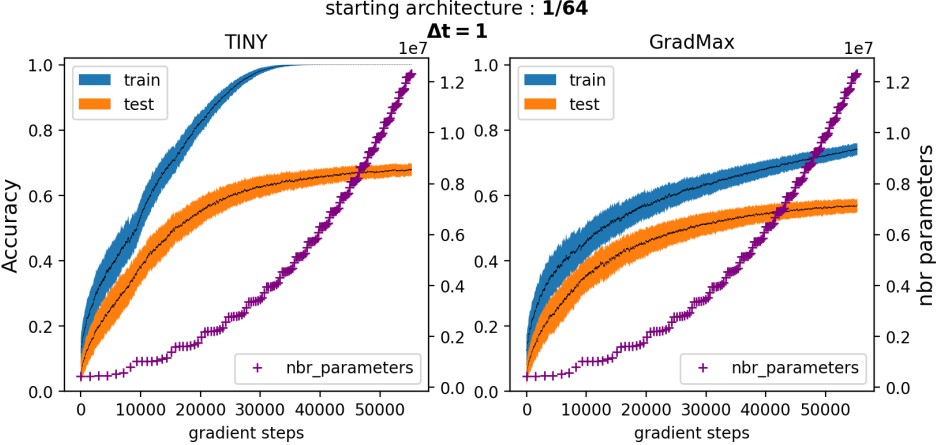

Figure 6: Evolution of accuracy and number of parameters as a function of gradient steps for the setting $\Delta t = 1$, $s = 1/64$, for TINY and GradMax. Mean and standard deviations are estimated over four runs. Results with other settings can be found in Fig. 13 in the appendix.

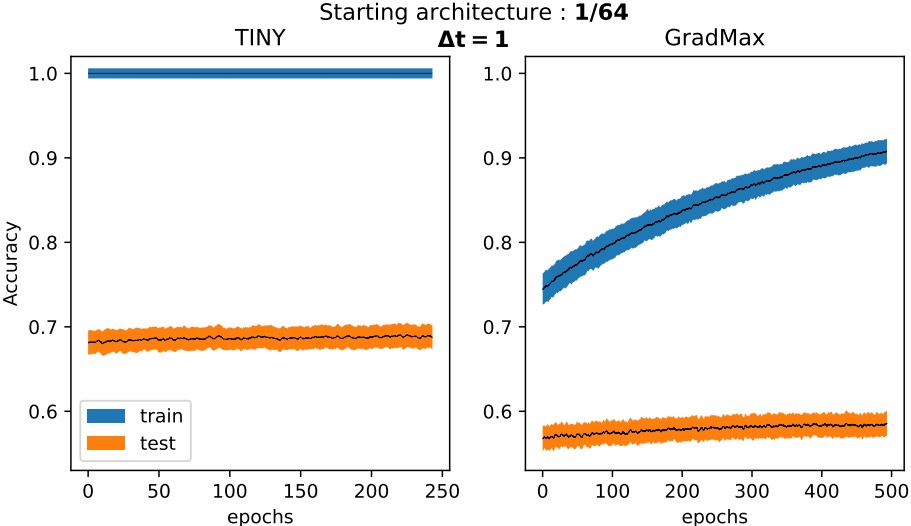

Figure 7: Evolution of accuracy as a function of epochs for the setting $\Delta t = 1$, $s = 1/64$ during extra training for TINY and GradMax. Mean and standard deviations are estimated over four runs. Results with other settings can be found in Figures 14 and 15 in the appendix.

| | $\Delta t$ | TINY | | GradMax | | Small references | Reference |
|---|---|---|---|---|---|---|---|
| | | 0.25 | 1 | 0.25 | 1 | | |
| s | $1/4$ | $67.2 \pm 0.1$ | $70.4 \pm 0.2$ | $65.1 \pm 0.2$ | $68.6 \pm 0.1$ | $68.6 \pm 0.4$ [5*] | $72.8$ |
| | $1/4$ | $70.3 \pm 0.2$ [5*] | $70.9 \pm 0.2$ [5*] | $67.0 \pm 0.2$ [5*] | $69.0 \pm 0.2$ [5*] | | $\pm$ |
| | $1/64$ | $65.8 \pm 0.1$ | $68.1 \pm 0.5$ | $45.0 \pm 0.4$ | $56.8 \pm 0.2$ | $63.4 \pm 0.3$ [5*] | $0.3$ [5*] |
| | $1/64$ | $69.5 \pm 0.2$ [5*] | $68.7 \pm 0.6$ [5*] | $57.0 \pm 0.4$ [10*] | $58.4 \pm 0.2$ [10*] | | |

Table 1: Final accuracy on test set of ResNet18 after architecture growth (*grey*) and after convergence (*blue*). The number of stars indicates the multiple of 50 epochs needed to achieve convergence. With the starting architecture ResNet$_{1/64}$ and $\Delta t = 0.25$, the method TINY achieves $65.8 \pm 0.1$ on test set after its growth and it reaches $69.5 \pm 0.2$ [5*] after $5* := 5 \times 50$ additional epochs (examples of training curves for the extra training can be found in Figure 14). Means and standard deviations are performed on 4 runs for each setting.

tecture far from full expressivity, i.e. ResNet$_{1/64}$, while TINY is able to handle it. As for the setting $s = 1/4$, both methods seem equivalent in terms of final performance and achieve full expressivity.

The curves on Figure 6, which are extracted from Figure 13 in the appendix, show that TINY models have converged at the end of the growing process, while GradMax ones have not. This latter effect contrasts with GradMax formulation which is to accelerate the gradient descent as fast as possible by adding neurons. Furthermore GradMax needs extra training to achieve full expressivity: for the particular setting $s = 1/64, \Delta t = 1$, the extra training time required by GradMax is twice as high as TINY's, as shown in Figure 7. This need for extra training also appears for all settings in Table 1. In particular for $s = 1/64, \Delta t = 0.25$, the difference in performance after and before extra training goes up to 20 % above the initial performance while it is only of 6% for TINY.

## 5.2 Comparison with Random on CIFAR-100 : initialisation impact

In this section, we focus on the impact of the new neurons' initialization. To do so, we consider as a baseline the Random method, which initializes the new neurons according to a Gaussian distribution: $(\alpha_k^*, \omega_k^*)_{k=1}^K \sim \mathcal{N}(0, I_d)$ or a uniform distribution $\mathcal{U}[-1, 1]$. Also, when adding new neurons, we now search for the best scaling using a line-search on the loss. Thus, we perform the operation $\theta_\leftrightarrow^K \leftarrow \gamma^* \theta_\leftrightarrow^K$, with the amplitude factor $\gamma^* \in \mathbb{R}$ defined as :

$$\gamma^* := \arg\min_{\gamma \in [-L, L]} \sum_i \ell(f_{\theta \oplus \gamma \theta_\leftrightarrow^K}(\boldsymbol{x}_i), \boldsymbol{y}_i) \qquad \text{with } \gamma \theta_\leftrightarrow^{K^*} = (\gamma \alpha_k^*, \gamma \omega_k^*)_{k=1}^K \qquad (14)$$

with $L$ a positive constant. More details can be found in Appendix E.3.2 and in Algorithm 1. With such an amplitude factor, one can measure the quality of the directions generated by TINY and Random by quantifying the maximal decrease of loss in these directions.

To better measure the impact of the initialisation method, and to distinguish it from the optimization process, we do not perform any gradient descent. This contrasts with the previous section where long training time after architecture growth was modifying the direction of the added neurons, dampening initialization impact with training time, especially as they were added with a small amplitude factor (cf Section E.3.1).

With these two modifications to the protocol of previous section, we obtain Figure 8. We see the crucial impact of TINY initialization compared to the Random one. Indeed, TINY reaches more than 17% accuracy just by adding neurons (without any further update), which accounts for about one quarter of the total accuracy with the full method (69% in Table 1 using in plus gradient descent). On the opposite, the Random initialization does not contribute to the accuracy through the growing process (just about 1%); this can be explained and quantified as follows. To study the random setting, we can model $\boldsymbol{v}(X)$ and $\boldsymbol{v}_{\text{goal}}(X)$ as independent variables where $\boldsymbol{v}_{\text{goal}} \sim \mathcal{N}(0_d, \frac{1}{d} I_d)$ and $\boldsymbol{v}$ either $\sim \mathcal{N}(0_d, \frac{1}{d} I_d)$ or $\sim \mathcal{U}[-\frac{1}{d}, \frac{1}{d}]$, $d$ being the dimension of $\boldsymbol{v}_{\text{goal}}$ and $\boldsymbol{v}$. From Equation (10), the scalar product $\langle \boldsymbol{V}(X), \boldsymbol{V}_{\text{goal}}(X) \rangle := \frac{1}{n} \sum_i \boldsymbol{v}_{\text{goal}}(\boldsymbol{x}_i)^T \boldsymbol{v}(\boldsymbol{x}_i)$ is a proxy for the expected decrease of loss after each architecture growth. This quantity can be approximated

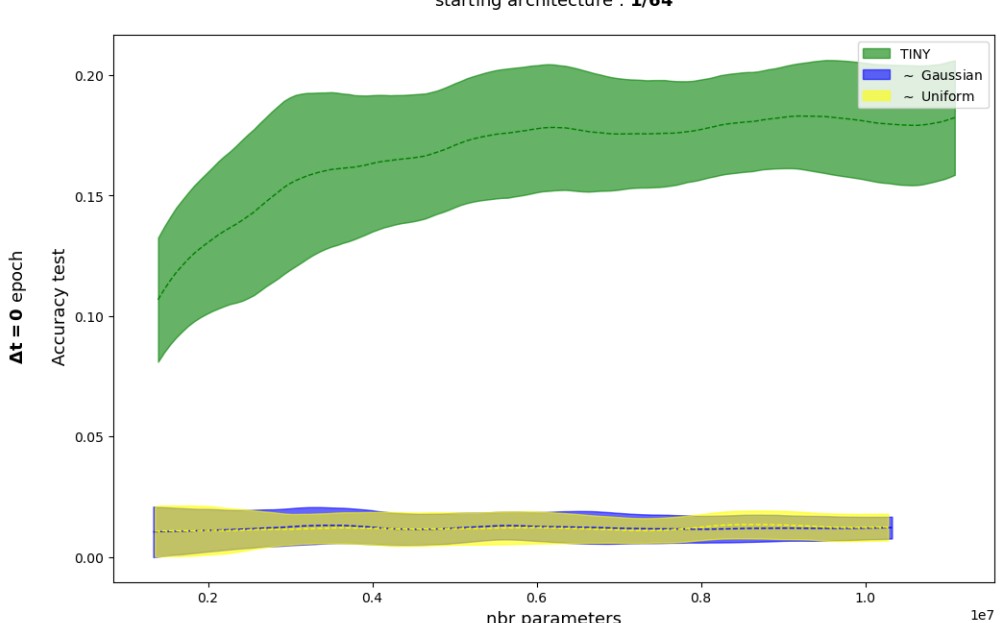

Figure 8: Test accuracy as a function of the number of parameters during pure architecture growth (no gradient steps performed, only neuron addition with a given initialization) from ResNet$_{1/64}$ to ResNet$_{18}$, averaged over four independent runs.

by its standard deviation, ie $\frac{1}{\sqrt{n\,d}}$, which makes the expected relative gain of loss (for a gradient step) of the order of magnitude of $\frac{1}{\sqrt{64}}$ for the first layer and $\frac{1}{\sqrt{512}}$ for the last layer when compared to the true gradient, and consequently when compared to TINY. Furthermore, one can take into account the effect of a line search over the random direction: in that case the expected relative loss gain is quadratic in the angle between the directions and therefore of the order of magnitude of $\frac{1}{64}$ or $\frac{1}{512}$ respectively (see Appendix D.3).

Note that the search interval of Equation (14) can be shrunk to $[0, L]$ with TINY initialization, as the first order development of the loss in Equation (10) is positive. This property is the direct consequence of the definition of $V^*$ as the minimizer of the expressivity bottleneck (Eq. (8)). One can also note that we do not include GradMax in Figure 8, because its protocol initializes the on-going weights to zero ($\alpha_k \leftarrow 0$) and imposes a small norm on its out-going weights ($||\omega_k|| = \varepsilon$). Those two aspects make the amplitude factor $\gamma^*$ meaningless and the impact of the new neurons initialization invisible without gradient descent.

The code is available at `https://gitlab.inria.fr/mverbock/tinypub`.

## 6 Conclusion

We provided the theoretical principles of TINY, a method to grow the architecture of a neural net while training it; starting from a very thin architecture, TINY adds neurons where needed and yields a fully trained architecture at the end. Our method relies on the functional gradient to find new directions that tackle the expressivity bottleneck, even for small networks, by expanding their space of parameters. This way, we combine in the same framework gradient descent and neural architecture search, that is, optimizing the network parameters and its architecture at the same time, and this, in a way that guarantees convergence to 0 training error, thus escaping expressivity bottlenecks indeed.

While transfer learning works well on ordinary tasks, for it to succeed, it needs to fine-tune and use large architectures at deployment in order to extract and manipulate common knowledge. Our method has the advantage of being generic and could also produce smaller models as it adapts the architecture to a single task.

It is already instantiated for linear and convolutional layers ; extension to self-attention mechanisms (transformers) is part of future works. Although common architectures consist of a succession of layers, a research direction is to develop tools handling general computational graphs (such as U-net, Inception, Dense-Net), which offers the possibility to let the architecture graph grow and bypass manual design.

Another possible development would be to study the statistical reliability of the TINY method, for instance using tools borrowed from random matrix theory. Indeed statistical tests can be applied to the intermediate computations from which the new neurons are obtained. An interesting byproduct of this approach would be to define a threshold to select neurons found by Prop. 3.2, based on statistical significance.

## 7   Acknowledgments

The authors address their deepest thanks to Stella Douka, Andrei Pantea, Stéphane Rivaud & François Landes for the exchanges and discussions on this project.

This work was supported by grants ANR-22-CE33-0015-01 and ANR-17-CONV-0003 operated by LISN to Sylvain Chevallier and by ANR-20-CE23-0025 operated by Inria to Guillaume Charpiat. This work was also funded by the European Union under GA no. 101135782 (MANOLO project).

Numerical computation was enabled by the scientific Python ecosystem: Matplotlib Hunter (2007), Numpy Harris et al. (2020), Scipy Virtanen et al. (2020), pandas pandas development team (2020), PyTorch Paszke et al. (2019).

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

**Appendix outline**

- Appendix A details the theoretical approach of TINY.

- Appendix B reformulates related works with a common theoretical framework and compares those approaches.

- Appendix C proves the propositions of Section 3.

- Appendix D details the proofs regarding the convergence (related to Section 4).

- Appendix E provides the hyper parameters for learning.

- Appendix F gives additional graphics associated to the result part.

For Appendices B and C, we use the trace scalar product and its associated norm. We note $\langle \, . \, , \, . \, \rangle := \langle \, . \, , \, . \, \rangle_{\mathrm{Tr}}$ and $|| \, . \, || := || \, . \, ||_{\mathrm{Tr}}$. One should remark that $|| \, . \, || = || \, . \, ||_{\mathrm{Tr}} = || \, . \, ||_2$.

# A  Theoretical details for part 2

## A.1  Functional gradient

The functional loss $\mathcal{L}$ is a functional that takes as input a function $f \in \mathcal{F}$ and outputs a real score:

$$\mathcal{L} : \ f \in \mathcal{F} \ \mapsto \ \mathcal{L}(f) = \underset{(\boldsymbol{x},\boldsymbol{y}) \sim \mathcal{D}}{\mathbb{E}} [\ell(f(\boldsymbol{x}), \boldsymbol{y})] \ \in \mathbb{R} \ .$$

The function space $\mathcal{F}$ can typically be chosen to be $L_2(\mathbb{R}^p \to \mathbb{R}^d)$, which is a Hilbert space. The directional derivative (or Gateaux derivative, or Fréchet derivative) of functional $\mathcal{L}$ at function $f$ in direction $v$ is defined as:

$$D\mathcal{L}(f)(v) = \lim_{\varepsilon \to 0} \frac{\mathcal{L}(f + \varepsilon v) - \mathcal{L}(f)}{\varepsilon}$$

if it exists. Here $v$ denotes any function in the Hilbert space $\mathcal{F}$ and stands for the direction in which we would like to update $f$, following an infinitesimal step (of size $\varepsilon$), resulting in a function $f + \varepsilon v$.

If this directional derivative exists in all possible directions $v \in \mathcal{F}$ and moreover is continuous in $v$, then the Riesz representation theorem implies that there exists a unique direction $v^* \in \mathcal{F}$ such that:

$$\forall v \in \mathcal{F}, \ \ D\mathcal{L}(f)(v) = \langle v^*, v \rangle \ .$$

This direction $v^*$ is named the *gradient* of the functional $\mathcal{L}$ at function $f$ and is denoted by $\nabla_f \mathcal{L}(f)$.

Note that while the inner product $\langle \cdot, \cdot \rangle$ considered is usually the $L_2$ one, it is possible to consider other ones, such as Sobolev ones (e.g., $H^1$). The gradient $\nabla_f \mathcal{L}(F)$ depends on the chosen inner product and should consequently rather be denoted by $\nabla_f^{L_2} \mathcal{L}(f)$ for instance.

Note that continuous functions from $\mathbb{R}^p$ to $\mathbb{R}^d$, as well as $C^\infty$ functions, are dense in $L_2(\mathbb{R}^p \to \mathbb{R}^d)$.

Let us now study properties specific to our loss design: $\mathcal{L}(f) = \mathbb{E}_{(\boldsymbol{x},\boldsymbol{y}) \sim \mathcal{D}} [\ell(f(\boldsymbol{x}), \boldsymbol{y})]$. Assuming sufficient $\ell$-loss differentiability and integrability, we get, for any function update direction $v \in \mathcal{F}$ and infinitesimal step size $\varepsilon \in \mathbb{R}$:

$$\mathcal{L}(f + \varepsilon v) - \mathcal{L}(f) = \underset{(\boldsymbol{x},\boldsymbol{y}) \sim \mathcal{D}}{\mathbb{E}} [\ell(f(\boldsymbol{x}) + \varepsilon v(\boldsymbol{x}), \boldsymbol{y}) - \ell(f(\boldsymbol{x}), \boldsymbol{y})]$$

$$= \underset{(\boldsymbol{x},\boldsymbol{y}) \sim \mathcal{D}}{\mathbb{E}} \left[ \nabla_{\boldsymbol{u}} \ell(\boldsymbol{u}, \boldsymbol{y})\big|_{\boldsymbol{u}=f(\boldsymbol{x})} \cdot \varepsilon v(\boldsymbol{x}) + O\left(\varepsilon^2 \|v(\boldsymbol{x})\|^2\right) \right]$$

using the usual gradient of function $\ell$ at point $(\boldsymbol{u} = f(\boldsymbol{x}), \boldsymbol{y})$ w.r.t. its first argument $\boldsymbol{u}$, with the standard Euclidean dot product $\cdot$ in $\mathbb{R}^p$. Then the directional derivative is:

$$D\mathcal{L}(f)(v) = \underset{(\boldsymbol{x},\boldsymbol{y}) \sim \mathcal{D}}{\mathbb{E}} \left[ \nabla_{\boldsymbol{u}} \ell(\boldsymbol{u}, \boldsymbol{y})\big|_{\boldsymbol{u}=f(\boldsymbol{x})} \cdot v(\boldsymbol{x}) \right] = \underset{\boldsymbol{x} \sim \mathcal{D}}{\mathbb{E}} \left[ \underset{\boldsymbol{y} \sim \mathcal{D}|\boldsymbol{x}}{\mathbb{E}} \left[ \nabla_{\boldsymbol{u}} \ell(\boldsymbol{u}, \boldsymbol{y})\big|_{\boldsymbol{u}=f(\boldsymbol{x})} \right] \cdot v(\boldsymbol{x}) \right]$$

and thus the functional gradient for the inner product $\langle v, v' \rangle_{\mathbb{E}} := \mathbb{E}_{\boldsymbol{x} \sim \mathcal{D}} [v(\boldsymbol{x}) \cdot v'(\boldsymbol{x})]$ is the function:

$$\nabla_f^{\mathbb{E}} \mathcal{L}(f) : \boldsymbol{x} \mapsto \underset{\boldsymbol{y} \sim \mathcal{D}|\boldsymbol{x}}{\mathbb{E}} \left[ \nabla_{\boldsymbol{u}} \ell(\boldsymbol{u}, \boldsymbol{y})\big|_{\boldsymbol{u}=f(\boldsymbol{x})} \right]$$

which simplifies into:

$$\nabla_f^{\mathbb{E}} \mathcal{L}(f) : \boldsymbol{x} \mapsto \nabla_{\boldsymbol{u}} \ell(\boldsymbol{u}, \boldsymbol{y}(\boldsymbol{x}))\big|_{\boldsymbol{u}=f(\boldsymbol{x})}$$

if there is no ambiguity in the dataset, i.e. if for each $\boldsymbol{x}$ there is a unique $\boldsymbol{y}(\boldsymbol{x})$.

Note that by considering the $L_2(\mathbb{R}^p \to \mathbb{R}^d)$ inner product $\int v \cdot v'$ instead, one would respectively get:

$$\nabla_f^{L_2} \mathcal{L}(f) : \boldsymbol{x} \mapsto p_{\mathcal{D}}(\boldsymbol{x}) \underset{\boldsymbol{y} \sim \mathcal{D}|\boldsymbol{x}}{\mathbb{E}} \left[ \nabla_{\boldsymbol{u}} \ell(\boldsymbol{u}, \boldsymbol{y})\big|_{\boldsymbol{u}=f(\boldsymbol{x})} \right]$$

and

$$\nabla_f^{L_2} \mathcal{L}(f) : \boldsymbol{x} \mapsto p_{\mathcal{D}}(\boldsymbol{x}) \nabla_{\boldsymbol{u}} \ell(\boldsymbol{u}, \boldsymbol{y}(\boldsymbol{x}))\big|_{\boldsymbol{u}=f(\boldsymbol{x})}$$

instead, where $p_{\mathcal{D}}(\boldsymbol{x})$ is the density of the dataset distribution at point $\boldsymbol{x}$. In practice one estimates such gradients using a minibatch of samples $(\boldsymbol{x}, \boldsymbol{y})$, obtained by picking uniformly at random within a finite dataset, and thus the formulas for the two inner products coincide (up to a constant factor).

## A.2   Differentiation under the integral sign

Let $X$ be an open subset of $\mathbb{R}$, and $\Omega$ be a measure space. Suppose $f : X \times \Omega \to \mathbb{R}$ satisfies the following conditions:

- $f(x, \omega)$ is a Lebesgue-integrable function of $\omega$ for each $x \in X$.

- For almost all $\omega \in \Omega$ , the partial derivative $\frac{\partial}{\partial x} f$ of $f$ according to $x$ exists for all $x \in X$.

- There is an integrable function $\theta : \Omega \to \mathbb{R}$ such that $\left| \frac{\partial}{\partial x}(x, \omega) \right| \leqslant \theta(\omega)$ for all $x \in X$ and almost every $\omega \in \Omega$.

Then, for all $x \in X$,

$$\frac{\partial}{\partial x} \int_{\Omega} f(x, \omega)\, d\omega = \int_{\Omega} \frac{\partial}{\partial x} f(x, \omega)\, d\omega \tag{15}$$

See proof and details: Le Gall (2006).

## A.3   Gradients and proximal point of view

Gradients with respect to standard variables such as vectors are defined in the same way as functional gradients above: given a sufficiently smooth loss $\widetilde{\mathcal{L}} : \theta \in \Theta_{\mathcal{A}} \mapsto \widetilde{\mathcal{L}}(\theta) = \mathcal{L}(f_{\theta}) \in \mathbb{R}$, and an inner product $\cdot$ in the space $\Theta_{\mathcal{A}}$ of parameters $\theta$, the gradient $\nabla_{\theta} \widetilde{\mathcal{L}}(\theta)$ is the unique vector $\boldsymbol{\tau} \in \Theta_{\mathcal{A}}$ such that:

$$\forall \delta\theta \in \Theta_{\mathcal{A}}, \quad \boldsymbol{\tau} \cdot \delta\theta = D_{\theta}\widetilde{\mathcal{L}}(\theta)(\delta\theta)$$

where $D_{\theta}\widetilde{\mathcal{L}}(\theta)(\delta\theta)$ is the directional derivative of $\widetilde{\mathcal{L}}$ at point $\theta$ in the direction $\delta\theta$, defined as in the previous section. This gradient depends on the inner product chosen, which can be highlighted by the following property. The opposite $-\nabla_{\theta}\widetilde{\mathcal{L}}(\theta)$ of the gradient is the unique solution of the problem:

$$\underset{\delta\theta \in \Theta_{\mathcal{A}}}{\arg\min} \left\{ D_{\theta}\widetilde{\mathcal{L}}(\theta)(\delta\theta) + \frac{1}{2}\, \|\delta\theta\|_P^2 \right\}$$

where $\| \ \|_P$ is the norm associated to the chosen inner product. Changing the inner product changes the way candidate directions $\delta\theta$ are penalized, leading to different gradients. This proximal formulation can be obtained as follows. For any $\delta\theta$, its distance to the gradient descent direction is:

$$\left\| \delta\theta - \left( -\nabla_{\theta}\widetilde{\mathcal{L}}(\theta) \right) \right\|^2 = \|\delta\theta\|^2 + 2\,\delta\theta \cdot \nabla_{\theta}\widetilde{\mathcal{L}}(\theta) + \left\| \nabla_{\theta}\widetilde{\mathcal{L}}(\theta) \right\|^2$$

$$= 2\left( \frac{1}{2}\, \|\delta\theta\|^2 + D_{\theta}\widetilde{\mathcal{L}}(\theta)(\delta\theta) \right) + K$$

where $K$ does not depend on $\delta\theta$. For the above to hold, the inner product used has to be the one from which the norm is derived. By minimizing this expression with respect to $\delta\theta$, one obtains the desired property.

In our case of study, for the norm over the space $\Theta_{\mathcal{A}}$ of parameter variations, we consider a norm in the space of associated functional variations, i.e.:

$$\|\delta\theta\|_P \ := \ \left\| \frac{\partial f_{\theta}}{\partial \theta}\, \delta\theta \right\|$$

which makes more sense from a physical point of view, as it is more intrinsic to the task to solve and depends as little as possible on the parameterization (i.e. on the architecture chosen). This results in a functional move that is the projection of the functional one to the set of possible moves given the architecture. On the opposite, the standard gradient (using Euclidean parameter norm $\|\delta\theta\|$ in parameter space) yields a

functional move obtained not only by projecting the functional gradient but also by multiplying it by a matrix $\frac{\partial f_\theta}{\partial \theta} \frac{\partial f_\theta}{\partial \theta}^T$ which can be seen as a strong architecture bias over optimization directions.

We consider here that the loss $\mathcal{L}$ to be minimized is the real loss that the user wants to optimize, possibly including regularizers to avoid overfitting, and since the architecture is evolving during training, possibly to architectures far from usual manual design and never tested before, one cannot assume architecture bias to be desirable. We aim at getting rid of it in order to follow the functional gradient descent as closely as possible.

Searching for

$$\boldsymbol{v}^* = \underset{\boldsymbol{v} \in \mathcal{T}_{\mathcal{A}}}{\arg\min} \|\boldsymbol{v} - \boldsymbol{v}_{\text{goal}}\|^2 = \underset{\boldsymbol{v} \in \mathcal{T}_{\mathcal{A}}}{\arg\min} \left\{ D\mathcal{L}(f)(\boldsymbol{v}) + \frac{1}{2}\|\boldsymbol{v}\|^2 \right\} \tag{16}$$

or equivalently for:

$$\delta\theta^* = \underset{\delta\theta \in \Theta_{\mathcal{A}}}{\arg\min} \left\| \frac{\partial f_\theta}{\partial \theta} \delta\theta - \boldsymbol{v}_{\text{goal}} \right\|^2 = \underset{\delta\theta \in \Theta_{\mathcal{A}}}{\arg\min} \left\{ D_\theta \mathcal{L}(f_\theta)(\delta\theta) + \frac{1}{2} \left\| \frac{\partial f_\theta}{\partial \theta} \delta\theta \right\|^2 \right\} =: -\nabla_\theta^{\mathcal{T}_{\mathcal{A}}} \mathcal{L}(f_\theta) \tag{17}$$

then appears as a natural goal.

### A.4  Example of expressivity bottleneck

**Example.** Suppose one tries to estimate the function $y = f_{\text{true}}(x) = 2\sin(x) + x$ with a linear model $f_{\text{predict}}(x) = ax + b$. Consider $(a, b) = (1, 0)$ and the square loss $\mathcal{L}$. For the dataset of inputs $(x_0, x_1, x_2, x_3) = (0, \frac{\pi}{2}, \pi, \frac{3\pi}{2})$, there exists no parameter update $(\delta a, \delta b)$ that would improve prediction at $x_0, x_1, x_2$ and $x_3$ simultaneously, as the space of linear functions $\{f : x \to ax + b \mid a, b \in \mathbb{R}\}$ is not expressive enough. To improve the prediction at $x_0, x_1, x_2$ **and** $x_3$, one should look for another, more expressive functional space such that for $i = 0, 1, 2, 3$ the functional update $\Delta f(x_i) := f^{t+1}(x_i) - f^t(x_i)$ goes into the same direction as the functional gradient $\boldsymbol{v}_{\text{goal}}(x_i) := -\nabla_{f(x_i)}\mathcal{L}(f(x_i), y_i) = -2(f(x_i) - y_i)$ where $y_i = f_{\text{true}}(x_i)$.

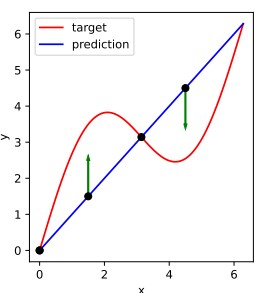

Figure 9: Linear interpolation

### A.5  Problem formulation and choice of pre-activities

There are several ways to design the problem of adding neurons, which we discuss now, in order to explain our choice of the pre-activities to express expressivity bottlenecks.

Suppose one wishes to add $K$ neurons $\theta_\leftrightarrow^K := (\boldsymbol{\alpha}_k, \boldsymbol{\omega}_k)_{k=1}^K$ to layer $l-1$, which impacts the activities $\boldsymbol{a}_l$ at the next layer, in order to improve its expressivity. These neurons could be chosen to have only null weights, or null input weights $\boldsymbol{\alpha}_k$ and non-null output weights $\boldsymbol{\omega}_k$, or the opposite, or both non-null weights. Searching for the best neurons to add for each of these cases will produce different optimization problems.

Let us remind first that adding such $K$ neurons with weights $\theta_\leftrightarrow^K := (\boldsymbol{\alpha}_k, \boldsymbol{\omega}_k)_{k=1}^K$ changes the activities $\boldsymbol{a}_l$ of the (next) layer by

$$\delta \boldsymbol{a}_l = \sum_{k=1}^K \boldsymbol{\omega}_k \, \sigma(\boldsymbol{\alpha}_k^T \boldsymbol{b}_{l-2}(\boldsymbol{x})) \tag{18}$$

**Small weight approximation**  Under the hypothesis of small input weights $\alpha_k$, and assuming smooth activation function $\sigma$ at 0, the activity variation 18 can be approximated by:

$$\sigma'(0) \sum_{k=1}^K \boldsymbol{\omega}_k \boldsymbol{\alpha}_k^T \boldsymbol{b}_{l-2}(\boldsymbol{x}) \tag{19}$$

at first order in $\|\boldsymbol{\alpha}_k\|$. We will drop the constant $\sigma'(0)$ in the sequel.

This quantity is linear both in $\alpha_k$ and $\boldsymbol{\omega}_k$, therefore the first-order parameter-induced activity variations are easy to compute:

$$\boldsymbol{v}^l\left(\boldsymbol{x}, (\boldsymbol{\alpha}_k)_{k=1}^K\right) = \frac{\partial \boldsymbol{a}_l(\boldsymbol{x})}{\partial\left((\boldsymbol{\alpha}_k)_{k=1}^K\right)}_{|(\boldsymbol{\alpha}_k)_{k=1}^K=0} (\boldsymbol{\alpha}_k)_{k=1}^K = \sum_{k=1}^K \boldsymbol{\omega}_k \boldsymbol{b}_{l-2}(\boldsymbol{x})^T \boldsymbol{\alpha}_k$$

$$\boldsymbol{v}^l\left(\boldsymbol{x}, (\boldsymbol{\omega}_k)_{k=1}^K\right) = \frac{\partial \boldsymbol{a}_l(\boldsymbol{x})}{\partial\left((\boldsymbol{\omega}_k)_{k=1}^K\right)}_{|(\boldsymbol{\omega}_k)_{k=1}^K=0} (\boldsymbol{\omega}_k)_{k=1}^K = \sum_{k=1}^K \boldsymbol{\omega}_k \boldsymbol{b}_{l-2}(\boldsymbol{x})^T \boldsymbol{\alpha}_k$$

so with a slight abuse of notation we have:

$$\boldsymbol{v}^l\left(\boldsymbol{x}, \theta_\leftrightarrow^K\right) = \sum_{k=1}^K \boldsymbol{\omega}_k \boldsymbol{\alpha}_k^T \boldsymbol{b}_{l-2}(\boldsymbol{x})$$

Note also that technically the quantity above is first-order in $\boldsymbol{\alpha}_k$ and in $\boldsymbol{\omega}_k$ but second-order in the joint variable $\theta_\leftrightarrow^K = (\boldsymbol{\alpha}_k, \boldsymbol{\omega}_k)$.

**Adding neurons with 0 weights (both input and output weights).** In that case, one increases the number of neurons in the layer, but without changing the function (since only null quantities are added) and also without changing the gradient with respect to the parameters, thus not improving expressivity. Indeed, the added quantity (Eq. 18) involves $0 \times 0$ multiplications, and consequently the derivative $\frac{\partial \boldsymbol{a}_l(\boldsymbol{x})}{\partial \theta_\leftrightarrow^K}\big|_{\theta_\leftrightarrow^K=0}$ w.r.t. these new parameters, that is, $\boldsymbol{b}_{l-2}(\boldsymbol{x})^T \boldsymbol{\alpha}_k$ w.r.t. $\boldsymbol{\omega}_k$ and $\boldsymbol{\omega}_k\,\boldsymbol{b}_{l-2}(\boldsymbol{x})^T$ w.r.t. $\boldsymbol{a}_k$ is 0, as both $\boldsymbol{a}_k$ and $\boldsymbol{\omega}_k$ are 0.

**Adding neurons with non-0 input weights and 0 output weights or the opposite.** In these cases, the addition of neurons will not change the function (because of multiplications by 0), but just the gradient. One of the 2 gradients (w.r.t. $\boldsymbol{a}_k$ or w.r.t $\boldsymbol{\omega}_k$) will be non-0, as the variable that is 0 has non-0 derivatives.

The question is then how to pick the best non-null variable, ($\boldsymbol{a}_k$ or $\boldsymbol{\omega}_k$) such that the added gradient will be the most useful. The problem can then be formulated similarly as what is done in the paper.

**Adding neurons with small yet non-0 weights.** In this case, both the function and its gradient will change when adding the neurons. Fortunately, Proposition 3.2 states that the best neurons to add in terms of expressivity (to get the gradient closer to the variation desired by the backpropagation) are also the best neurons to add to decrease the loss, i.e. the function change they will imply goes into the right direction.

For each family $(\boldsymbol{\omega}_k)_{k=1}^K$, the tangent space in $\boldsymbol{a}_l$ restricted to the family $(\boldsymbol{\alpha}_k)_{k=1}^K$, *i.e.* $\mathcal{T}_\mathcal{A}^{\boldsymbol{a}_l} := \left\{ \frac{\partial \boldsymbol{a}_l}{\partial(\boldsymbol{\alpha}_k)_{k=1}^K}|_{(\boldsymbol{\alpha}_k)_{k=1}^K=0}(.)(\boldsymbol{\alpha}_k)_{k=1}^K | (\boldsymbol{\alpha}_k)_{k=1}^K \in \left(\mathbb{R}^{|\boldsymbol{b}_{l-2}(\boldsymbol{x})|}\right)^K \right\}$ varies with the family $(\boldsymbol{\omega}_k)_{k=1}^K$, *i.e.* $\mathcal{T}_\mathcal{A}^{\boldsymbol{a}_l} := \mathcal{T}_\mathcal{A}^{\boldsymbol{a}_l}((\boldsymbol{\omega}_k)_{k=1}^K)$. Optimizing w.r.t. the $\boldsymbol{\omega}_k$ is equivalent to search for the best tangent space for the $\boldsymbol{\alpha}_k$, while symmetrically optimizing w.r.t. the $\boldsymbol{\alpha}_k$ is equivalent to find the best projection on the tangent space defined by the $\boldsymbol{\omega}_k$ (Figure 10).

**Pre-activities vs. post-activities.** The space of pre-activities $\boldsymbol{a}_l$ is a natural space for this framework, as they are formed with linear operations and we compute first-order variation quantities. Considering the space of post-activities $\boldsymbol{b}_l = \sigma(\boldsymbol{a}_l)$ is also possible, though computing variations will be more complex. Indeed, without first-order approximation, the obtained problem is not manageable, because of the non-linear activation function $\sigma$ added in front of all quantities (while in the cases of pre-activations, quantity 18 is linear in $\boldsymbol{\omega}_k$ and thus does not require approximation in $\boldsymbol{\omega}_k$, which allow considering large $\boldsymbol{\omega}_k$), and, with first-order approximation, it would add the derivative of the activation function, taken at various locations $\sigma'(\boldsymbol{a}_l)$ (while in the previous case the derivatives of the activation function were always taken at 0).

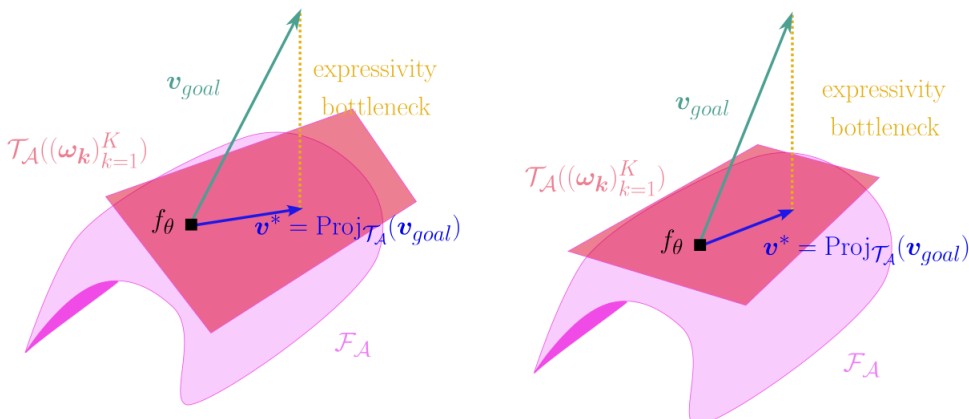

Figure 10: Changing the tangent space with different values of $(\boldsymbol{\omega}_k)_{k=1}^K$.

### A.6 Adding convolutional neurons

To add a convolutional neuron at layer $l-1$, one should add a kernel at layer $l-1$ and expand one dimension to all the kernels in layer $l$ to match the new dimension of the post-activity, see Figure 11.

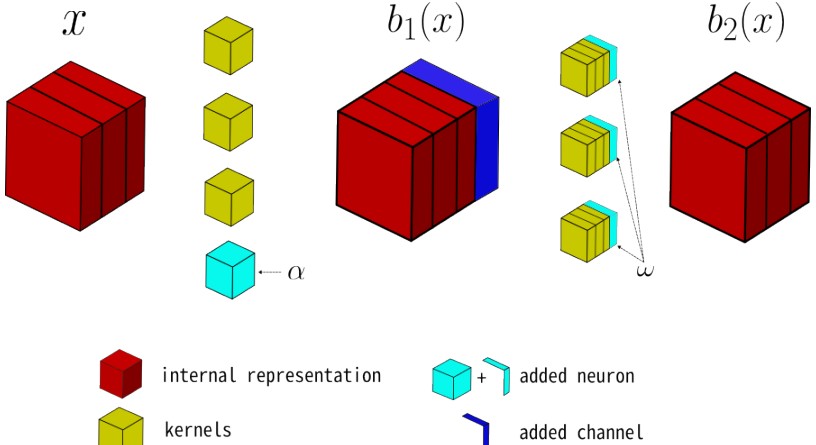

Figure 11: Adding one convolutional neuron at layer one for an input with tree channels.

## B   Theoretical comparison with other approaches

### B.1   GradMax method

To facilitate reading we remove the layer index of each quantity, *i.e.* $\boldsymbol{b} := \boldsymbol{b}_{l-2}$, $\boldsymbol{B} := \boldsymbol{B}_{l-2}$, $\boldsymbol{V}_{\text{goal}} := \boldsymbol{V}_{\text{goal}}^l$ and $\boldsymbol{V}_{\text{goal}_{proj}} := \boldsymbol{V}_{\text{goal}_{proj}}^l$.

The theoretical approach of GradMax is to add neurons with zero fan-in weights and choose the fan-out weights that would decrease the loss as much as possible after one gradient step. We note $\boldsymbol{\Omega}$ the fan-out weights of such neurons and perform the addition at layer $l-1$ at time $t$. After one gradient step, *i.e.* $t \to t+1$,

and considering a learning rate equal to 1, the decrease of loss is :

$$\mathcal{L}^{t+1} \approx \mathcal{L}^t + \langle \nabla_\theta \mathcal{L}, \delta\theta \rangle + \langle \nabla_\Omega \mathcal{L}, \delta\Omega \rangle \tag{20}$$

Taking the direction of the usual gradient descent, ie $\delta\theta = -\nabla_\theta \mathcal{L}$ and $\delta\Omega = -\nabla_\Omega \mathcal{L}$, we have :

$$\mathcal{L}^{t+1} \approx \mathcal{L}^t - ||\nabla_\theta \mathcal{L}||^2 - ||\nabla_\Omega \mathcal{L}||^2 \tag{21}$$

The output weights of the new neurons as formulated in the original paper Evci et al. (2022) at eq (11) are the solution of :

$$(\boldsymbol{\omega}_1^*, ..., \boldsymbol{\omega}_K^*) := \boldsymbol{\Omega}^* = \arg\max_{\boldsymbol{\Omega}} ||\nabla_\Omega \mathcal{L}||^2 \qquad s.t. \ \ ||\boldsymbol{\Omega}||^2 \leqslant c \tag{22}$$

we remark that :

$$\begin{aligned}
||\nabla_\Omega \mathcal{L}||^2 &= \left\lVert \sum_i \boldsymbol{b}(\boldsymbol{x}_i) \, \boldsymbol{v}_{\text{goal}}{}^T(\boldsymbol{x}_i) \, \boldsymbol{\Omega} \right\rVert^2 \\
&= \left\lVert \boldsymbol{B}\boldsymbol{V}_{\text{goal}}{}^T\boldsymbol{\Omega} \right\rVert^2 \\
&= \left\lVert \tilde{\boldsymbol{N}}\boldsymbol{\Omega} \right\rVert^2 \qquad\qquad\qquad\qquad \tilde{\boldsymbol{N}} := \boldsymbol{B}\boldsymbol{V}_{\text{goal}}{}^T
\end{aligned}$$

It follows that the fan-out of the neurons computed by GradMax are the solution of the problem :

$$\boldsymbol{\Omega}^* := \arg\max_{\boldsymbol{\Omega}} \left\lVert \tilde{\boldsymbol{N}}\boldsymbol{\Omega} \right\rVert \quad s.t. \ \ ||\boldsymbol{\Omega}||^2 \leqslant c \tag{23}$$

To compare this optimization problem with TINY, we use the following proposition:

**Proposition B.1.** $\forall \boldsymbol{D} \in \mathbb{R}^{p,q}, \boldsymbol{B} \in \mathbb{R}^{k,q}$,

$$\exists c \in \mathbb{R} \quad s.t. \quad \arg\min_{\boldsymbol{H} \in \mathbb{R}^{p,k}} ||\boldsymbol{D} - \boldsymbol{H}\boldsymbol{B}||^2 = \arg\max_{\boldsymbol{H}, \ ||\boldsymbol{H}\boldsymbol{B}||^2 \leqslant c} \langle \boldsymbol{D}, \boldsymbol{H}\boldsymbol{B} \rangle$$

*The proof can be found in B.1.*

Taking $\boldsymbol{V}_{\text{goal}}$ as $\boldsymbol{D}$ and $\boldsymbol{V} = \boldsymbol{\Omega}\boldsymbol{A}^T\boldsymbol{B}$ as $\boldsymbol{H}\boldsymbol{B}$, we can reformulate TINY optimization problem 8 as :

$$\boldsymbol{A}^*, \boldsymbol{\Omega}^* = \arg\max_{\boldsymbol{A}, \boldsymbol{\Omega}} \langle \boldsymbol{V}(\boldsymbol{A}, \boldsymbol{\Omega}), \boldsymbol{V}_{\text{goal}_{proj}} \rangle \qquad s.t. \ \ ||\boldsymbol{V}(\boldsymbol{A}, \boldsymbol{\Omega})||^2 \leqslant c \tag{24}$$

We remark that

$$\begin{aligned}
\left\langle \boldsymbol{V}(\boldsymbol{A}, \boldsymbol{\Omega}), \boldsymbol{V}_{\text{goal}_{proj}} \right\rangle &= \left\langle \boldsymbol{\Omega}\boldsymbol{A}^T\boldsymbol{B}, \boldsymbol{V}_{\text{goal}_{proj}} \right\rangle \\
&= \text{Tr}(\boldsymbol{B}^T\boldsymbol{A}\boldsymbol{\Omega}^T\boldsymbol{V}_{\text{goal}_{proj}}) \\
&= \text{Tr}(\boldsymbol{A}\boldsymbol{\Omega}^T\boldsymbol{V}_{\text{goal}_{proj}}\boldsymbol{B}^T) \\
&= \left\langle \boldsymbol{\Omega}\boldsymbol{A}^T, \boldsymbol{V}_{\text{goal}_{proj}}\boldsymbol{B}^T \right\rangle
\end{aligned}$$

With the definition $\boldsymbol{N} := \boldsymbol{B}\boldsymbol{V}_{\text{goal}_{proj}}^T$,

$$\left\langle \boldsymbol{V}(\boldsymbol{A}, \boldsymbol{\Omega}), \boldsymbol{V}_{\text{goal}_{proj}} \right\rangle = \left\langle \boldsymbol{A}\boldsymbol{\Omega}^T, \boldsymbol{N} \right\rangle \tag{25}$$

For the constrain on $\boldsymbol{V}(\boldsymbol{A}, \boldsymbol{\Omega})$, we perform the change of variable $\tilde{\boldsymbol{A}} := \boldsymbol{S}^{\frac{1}{2}}\boldsymbol{A}$, it follows that :

$$\begin{aligned}
||\boldsymbol{V}(\boldsymbol{A}, \boldsymbol{\Omega})||^2 &= \left\lVert \boldsymbol{\Omega}\boldsymbol{A}^T\boldsymbol{B} \right\rVert^2 \\
&= \text{Tr}(\boldsymbol{A}\boldsymbol{\Omega}^T\boldsymbol{\Omega}\boldsymbol{A}^T\boldsymbol{S}) \qquad\qquad \boldsymbol{S} = \boldsymbol{B}\boldsymbol{B}^T \\
&= \left\lVert \boldsymbol{\Omega}(\boldsymbol{S}^{\frac{1}{2}}\boldsymbol{A})^T \right\rVert \\
&= \left\lVert \boldsymbol{\Omega}\tilde{\boldsymbol{A}}^T \right\rVert
\end{aligned}$$

With the same change of variable, the initial scalar product Equation (25) is :

$$\langle \boldsymbol{V}(\boldsymbol{A}, \boldsymbol{\Omega}), \boldsymbol{V}_{\mathrm{goal}_{proj}} \rangle = \langle \boldsymbol{\Omega}\tilde{\boldsymbol{A}}^T \mathbf{S}^{-\frac{1}{2}}, \boldsymbol{N}^T \rangle = \langle \boldsymbol{\Omega}\tilde{\boldsymbol{A}}^T, \boldsymbol{N}^T \mathbf{S}^{-\frac{1}{2}} \rangle = \langle \tilde{\boldsymbol{A}}\boldsymbol{\Omega}^T, \mathbf{S}^{-\frac{1}{2}}\boldsymbol{N} \rangle \tag{26}$$

TINY optimization problem is now equivalent to :

$$\hat{\boldsymbol{A}}^*, \boldsymbol{\Omega}^* = \underset{\hat{\boldsymbol{A}}, \boldsymbol{\Omega}}{\arg\max} \langle \tilde{\boldsymbol{A}}\boldsymbol{\Omega}^T, \mathbf{S}^{-\frac{1}{2}}\boldsymbol{N} \rangle \qquad\qquad s.t. \ \left|\left| \boldsymbol{\Omega}\tilde{\boldsymbol{A}}^T \right|\right|^2 \leqslant c \tag{27}$$

To maximize the scalar product, we choose $\tilde{\boldsymbol{A}}\boldsymbol{\Omega}^T = \mathbf{S}^{-\frac{1}{2}}\boldsymbol{N}$. A solution for $(\tilde{\boldsymbol{A}}, \boldsymbol{\Omega})$ is the (left, right) eigenvectors of the matrix $\mathbf{S}^{-\frac{1}{2}}\boldsymbol{N}$. It implies that :

$$\boldsymbol{\Omega}^* := \underset{\boldsymbol{\Omega}}{\arg\max} \left|\left| \mathbf{S}^{-\frac{1}{2}}\boldsymbol{N}\boldsymbol{\Omega} \right|\right|^2 \quad s.t. \, ||\boldsymbol{\Omega}|| \leqslant \tilde{c} \tag{28}$$

One can note three differences between GradMax optimization problem and the last formulation of TINY (Equation (28)):

- First, the matrix $\tilde{\boldsymbol{N}}$ is not defined using the projection of the desired update $\boldsymbol{V}_{\mathrm{goal}_{proj}}^{l+1}$. As a consequence, GradMax does not take into account redundancy, and on the opposite will actually try to add new neurons that are as redundant as possible with the part of the goal update that is already feasible with already-existing neurons.

- Second, the constraint lies in the weight space for GradMax method while it lies in the pre-activation space in our case. The difference is that GradMax relies on the Euclidean metric in the space of parameters, which arguably offers less meaning that the Euclidean metric in the space of activities. Essentially this is the same difference as between the standard L2 gradient w.r.t. parameters and the natural gradient, which takes care of parameter redundancy and measures all quantities in the output space in order to be independent from the parameterization. In practice we do observe that the "natural" gradient direction improves the loss better than the usual L2 gradient.

- Third, our fan-in weights are not set to 0 but directly to their optimal values (at first order).

We now prove the proposition B.1.

*Proof.* Indeed,

$$\underset{\boldsymbol{H}}{\arg\min} ||\boldsymbol{D} - \boldsymbol{H}\boldsymbol{B}||^2 = \underset{\boldsymbol{H}}{\arg\min} ||\boldsymbol{H}\boldsymbol{B}||^2 - 2\langle \boldsymbol{D}, \boldsymbol{H}\boldsymbol{B} \rangle \tag{29}$$

$$= \underset{\boldsymbol{H}}{\arg\min} ||\boldsymbol{H}\boldsymbol{B}||^2 - 2\,||\boldsymbol{H}\boldsymbol{B}|| \left\langle \boldsymbol{D}, \frac{\boldsymbol{H}\boldsymbol{B}}{||\boldsymbol{H}\boldsymbol{B}||} \right\rangle \tag{30}$$

$$= \underset{h\boldsymbol{U}, \ ||\boldsymbol{H}\boldsymbol{B}||=h, \ \boldsymbol{U}=\frac{\boldsymbol{H}\boldsymbol{B}}{||\boldsymbol{H}\boldsymbol{B}||}}{\arg\min} h^2 - 2h\langle \boldsymbol{D}, \boldsymbol{U} \rangle \tag{31}$$

$$\tag{32}$$

We note $\boldsymbol{U}^* := \underset{\boldsymbol{U}=\frac{\boldsymbol{H}\boldsymbol{B}}{||\boldsymbol{H}\boldsymbol{B}||}}{\arg\max} \langle \boldsymbol{D}, \boldsymbol{U} \rangle$ which depends on $\boldsymbol{B}$ and $\boldsymbol{D}$ but does not depend on $h$. Then :

$$\underset{\boldsymbol{H}}{\arg\min} ||\boldsymbol{D} - \boldsymbol{H}\boldsymbol{B}||^2 = \underset{h\boldsymbol{U}^*, \ h \geqslant 0}{\arg\min} h^2 - 2h \langle \boldsymbol{D}, \boldsymbol{U}^* \rangle \tag{33}$$

$$= h^*\boldsymbol{U}^* \tag{34}$$

With the convention that $\frac{0}{||0||} = 0$. $\qquad\qquad\square$

## B.2 NORTH Preactivation

In paper Maile et al. (2022), fan-out weights are initialized to 0 while fan-in weights are initialized as $\boldsymbol{\alpha}_i = \boldsymbol{S}^{-1}\boldsymbol{B}_{l-2}\mathbf{V}_{\mathbf{A}_{l-1}}\boldsymbol{r}_i$ where $\boldsymbol{r}_i$ is a random vector and $\mathbf{V}_{\mathbf{Z}_{l-1}} \in \mathbb{R}^{n,|\ker(\boldsymbol{A}_{l-1}^T)|}$ is a matrix consisting of orthogonal vectors of the kernel of pre-activations $\boldsymbol{A}_{l-1}$. In our paper fan-in weights are initialized as $\boldsymbol{\alpha}_i = \mathbf{S}^{-\frac{1}{2}}\boldsymbol{B}_{l-2}\boldsymbol{V}_{\text{goal}}{}^T_{proj}\boldsymbol{v}_i = \mathbf{S}^{-\frac{1}{2}}\boldsymbol{B}_{l-2}\,\mathcal{P}_{\ker(\boldsymbol{B}_{l-1})}\,\boldsymbol{V}_{\text{goal}}{}^T\boldsymbol{v}_i$, where $\mathcal{P}_{Ker(\boldsymbol{B}_{l-1})} \in \mathbb{R}^{n,n}$ is the projector on the kernel of the linear application $\boldsymbol{B}_{l-1}$ and $\boldsymbol{v}_i$ are the right eigenvectors of the matrix $\boldsymbol{S}^{-\frac{1}{2}}\boldsymbol{N}$.

The main difference is thus that we use the backpropagation to find the best $\boldsymbol{v}_i$ or $\boldsymbol{r}_i$ directly, while the NORTH approach tries random directions $\boldsymbol{r}_i$ to explore the space of possible neuron additions.

# C   Proofs of Part 3

## C.1   Proposition 3.1

Denoting by $\boldsymbol{M}^+$ the generalized (pseudo-)inverse of $\boldsymbol{M}$, we have:

$$\delta\boldsymbol{W}_l^* = \frac{1}{n}\boldsymbol{V}_{\text{goal}}{}^l\boldsymbol{B}_{l-1}^T\left(\frac{1}{n}\boldsymbol{B}_{l-1}\boldsymbol{B}_{l-1}^T\right)^+ \text{ and } \boldsymbol{V}_0^l = \frac{1}{n}\boldsymbol{V}_{\text{goal}}{}^l\boldsymbol{B}_{l-1}^T\left(\frac{1}{n}\boldsymbol{B}_{l-1}\boldsymbol{B}_{l-1}^T\right)^+\boldsymbol{B}_{l-1}\,.$$

*Proof.* **Fully connected layers**

Consider the function

$$g(\delta\boldsymbol{W}) := \left|\left|\boldsymbol{V}_{\text{goal}}{}^l - \delta\boldsymbol{W}\boldsymbol{B}_{l-1}\right|\right|_F^2 \tag{35}$$

then:

$$g(\delta\boldsymbol{W} + \boldsymbol{H}) = ||\boldsymbol{V}_{\text{goal}}{}^l - \delta\boldsymbol{W}\boldsymbol{B}_{l-1} - \boldsymbol{H}\boldsymbol{B}_{l-1}||^2 \tag{36}$$

$$= g(\delta\boldsymbol{W}) - 2\left\langle\boldsymbol{V}_{\text{goal}}{}^l - \delta\boldsymbol{W}\boldsymbol{B}_{l-1}, \boldsymbol{H}\boldsymbol{B}_{l-1}\right\rangle + o(||\boldsymbol{H}||) \tag{37}$$

$$= g(\delta\boldsymbol{W}) - 2\left\langle\left(\boldsymbol{V}_{\text{goal}}{}^l - \delta\boldsymbol{W}\boldsymbol{B}_{l-1}\right)\boldsymbol{B}_{l-1}^T, \boldsymbol{H}\right\rangle + o(||\boldsymbol{H}||) \tag{38}$$

By identification $\nabla_{\delta\boldsymbol{W}}g(\delta\boldsymbol{W}) = -2\left(\boldsymbol{V}_{\text{goal}}{}^l - \delta\boldsymbol{W}\boldsymbol{B}_{l-1}\right)\boldsymbol{B}_{l-1}^T$, and thus:

$$\nabla_{\delta\boldsymbol{W}}g(\delta\boldsymbol{W}) = 0 \implies \boldsymbol{V}_{\text{goal}}{}^l\boldsymbol{B}_{l-1}^T = \delta\boldsymbol{W}\boldsymbol{B}_{l-1}\boldsymbol{B}_{l-1}^T$$

Using that $g$ is convex and the definition of the generalized inverse, we get:

$$\delta\boldsymbol{W}_l^* = \frac{1}{n}\boldsymbol{V}_{\text{goal}}{}^l\boldsymbol{B}_{l-1}^T\left(\frac{1}{n}\boldsymbol{B}_{l-1}\boldsymbol{B}_{l-1}^T\right)^+$$

as one solution.

**Convolutional layers**

For convolutional layers we aim to solve (where we dropped the index $l - 1$ for readability):

$$\underset{\delta\boldsymbol{W}}{\arg\min}\,||\boldsymbol{V}_{\text{goal}} - \mathbf{Conv}_{\delta\boldsymbol{W}}(\boldsymbol{B})||_F \tag{39}$$

We convert this in a linear regression by transforming the convolution in a matrix multiplication. First we reshape and permute:

- $\delta \boldsymbol{W} \in (C[+1], C, d[+1], d[+1])$ is transformed in $\delta \boldsymbol{W}_F \in (C[+1], Cd[+1]d[+1])$

- $\boldsymbol{V}_{\text{goal}} \in (n, C[+1], H[+1], W[+1])$ is transformed in $\boldsymbol{V}_{\text{goal}_F} \in (nH[+1]W[+1], C[+1])$

Then we can define $\boldsymbol{B}^c \in (n, Cd[+1]d[+1], H[+1]W[+1])$ and its reshaped version $\boldsymbol{B}_F^c \in (nH[+1]W[+1], Cd[+1]d[+1])$ that satisfies that $\boldsymbol{B}_F^c \delta \boldsymbol{W}_F^T \in (nH[+1]W[+1], C[+1])$ is a reshaped version of $\mathbf{Conv}_{\delta \boldsymbol{W}}(\boldsymbol{B})$. ($\boldsymbol{B}^c$ can be easily computed using `torch.nn.Unfold`.)

Hence Equation (39) becomes:

$$\underset{\delta \boldsymbol{W}_F}{\arg\min} \left\| \boldsymbol{V}_{\text{goal}_F} - \boldsymbol{B}_F^c \delta \boldsymbol{W}_F^T \right\|_F \tag{40}$$

Using the same reasoning that for fully connected layers, we get an optimal solution:

$$(\delta \boldsymbol{W}_F^*)^T = \left( (\boldsymbol{B}_F^c)^T \boldsymbol{B}_F^c \right)^+ (\boldsymbol{B}_F^c)^T \boldsymbol{V}_{\text{goal}_F} \tag{41}$$

$\square$

## C.2 Proposition 3.2

We define the matrices $\boldsymbol{N} := \frac{1}{n} \boldsymbol{B}_{l-2} \left( \boldsymbol{V}_{\text{goal}}{}^l_{proj} \right)^T$ and $\boldsymbol{S} := \frac{1}{n} \boldsymbol{B}_{l-2} \boldsymbol{B}_{l-2}^T$. Let us denote its SVD by $\boldsymbol{S} = \boldsymbol{O} \Sigma \boldsymbol{O}^T$, and note $\boldsymbol{S}^{-\frac{1}{2}} := \boldsymbol{O} \sqrt{\Sigma}^{-1} \boldsymbol{O}^T$ and consider the SVD of the matrix $\boldsymbol{S}^{-\frac{1}{2}} \boldsymbol{N} = \sum_{k=1}^{R} \lambda_k \boldsymbol{u}_k \boldsymbol{v}_k^T$ with $\lambda_1 \geqslant ... \geqslant \lambda_R \geqslant 0$, where $R$ is the rank of the matrix $\boldsymbol{N}$. Then:

**Proposition C.1** (3.2). *The solution of equation 8 can be written as:*

- *optimal number of neurons:* $K^* = R$

- *their optimal weights:* $(\boldsymbol{\alpha}_k^*, \boldsymbol{\omega}_k^*) = (\sqrt{\lambda_k} S^{-\frac{1}{2}} \boldsymbol{u}_k, \sqrt{\lambda_k} \boldsymbol{v}_k)$ *for* $k = 1, ..., R$.

*Moreover for any number of neurons* $K \leqslant R$, *and associated scaled weights* $\theta_{\leftrightarrow}^{K,*}$, *the expressivity gain and the first order in* $\eta$ *of the loss improvement due to the addition of these* $K$ *neurons are equal and can be quantified very simply as a function of the eigenvalues* $\lambda_k$:

$$\Psi^l_{\theta \oplus \theta_{\leftrightarrow}^{K,*}} = \Psi^l_\theta - \sum_{k=1}^{K} \lambda_k^2$$

*for fully-connected layers, with an inequality instead* ($\leqslant$) *for convolutional layers.*

*Proof.*
To facilitate reading we remove the layer index of each quantity, ie $\boldsymbol{B} := \boldsymbol{B}_{l-2}$, $\boldsymbol{V}^l(\boldsymbol{A}, \boldsymbol{\Omega}) := \boldsymbol{V}(\boldsymbol{A}, \boldsymbol{\Omega})$ and $\boldsymbol{V}_{\text{goal}}{}^l_{proj} := \boldsymbol{V}_{\text{goal}_{proj}}$. We fix $n$ and $\boldsymbol{x}_1, ...., \boldsymbol{x}_n$ on which we solve the expressivity bottleneck formula.

To solve this problem, we consider the input of the incoming connections $\boldsymbol{B}$ and the desired change in the output of the outgoing connections $\boldsymbol{V}_{\text{goal}_{proj}}$. Hence if we note $L(\boldsymbol{A})$ and $L(\boldsymbol{\Omega})$ the additional connections of the expanded representation and $\sigma$ the non linearity, we optimize the following proxy problem:

$$\underset{\boldsymbol{A}, \boldsymbol{\Omega}}{\arg\min} \frac{1}{n} \left\| (L(\boldsymbol{\Omega}) \circ \sigma \circ L(\boldsymbol{A}))(\boldsymbol{B}) - \boldsymbol{V}_{\text{goal}_{proj}} \right\|_{\text{Tr}} \tag{42}$$

We solve this problem at first order by linearizing the non-linearity $\sigma$. We denote $\mathbf{Lin}_{(a,b)}(W)$ the fully connected layer with input size $a$, output size $b$ and weight matrix $\boldsymbol{W}$. We also note $C[+1]$ and $C[-1]$ the

layer width at layer $l+1$ and $l-1$ with the convention that C[0] is the dimension of the input $\boldsymbol{x}$. With those notations, for fully connected layers, we have for the additions of $K$ neurons:

$$\underset{\boldsymbol{A},\boldsymbol{\Omega}}{\arg\min} \frac{1}{n} \left|\left|\mathbf{Lin}_{(C[+1],K)}(\boldsymbol{\Omega})(\mathbf{Lin}_{(K,C[-1])}(\boldsymbol{A})(\boldsymbol{B})) - \boldsymbol{V}_{\text{goal}_{proj}}\right|\right|_2 \tag{43}$$

With the same notations, for convolutional layers, we have for the additions of $K$ intermediate channels:

$$\underset{\boldsymbol{A},\boldsymbol{\Omega}}{\arg\min} \frac{1}{n} \left|\left|\mathbf{Conv}_{(C[+1],K)}(\boldsymbol{\Omega})(\mathbf{Conv}_{(K,C[-1])}(\boldsymbol{A})(\boldsymbol{B})) - \boldsymbol{V}_{\text{goal}_{proj}}\right|\right|_2 \tag{44}$$

If we note $\boldsymbol{V}(\boldsymbol{A},\boldsymbol{\Omega})$ the result of $\boldsymbol{B}$ after applying the layers parametrized by $\boldsymbol{A}$ and $\boldsymbol{\Omega}$, in both cases we aim to optimize:

$$\underset{\boldsymbol{A},\boldsymbol{\Omega}}{\arg\min} \frac{1}{n} \left|\left|\boldsymbol{V}(\boldsymbol{A},\boldsymbol{\Omega}) - \boldsymbol{V}_{\text{goal}_{proj}}\right|\right|_{\text{Tr}} \tag{45}$$

First we will transform the resolution of the problem in solving the following optimization problem:

$$\underset{\boldsymbol{A},\boldsymbol{\Omega}}{\arg\min} \left|\left|\mathbf{S}^{\frac{1}{2}}\boldsymbol{A}\boldsymbol{\Omega}^T - \mathbf{S}^{-\frac{1}{2}}\boldsymbol{N}\right|\right|_2 \tag{46}$$

where $\boldsymbol{S}$ depends of $\boldsymbol{B}$ and $\boldsymbol{N}$ of $\boldsymbol{B}$ and $\boldsymbol{V}_{\text{goal}_{proj}}$.

If we note $\boldsymbol{S} = \boldsymbol{O}\boldsymbol{\Lambda}\boldsymbol{O}^T$ the SVD of $\boldsymbol{S}$, we define the square root of $\boldsymbol{S}$ as $\mathbf{S}^{\frac{1}{2}} := \boldsymbol{O}\sqrt{\boldsymbol{\Lambda}}\boldsymbol{O}^T$ and $\mathbf{S}^{-\frac{1}{2}} := \boldsymbol{O}\sqrt{\boldsymbol{\Lambda}^{-1}}\boldsymbol{O}^T$ with the convention $0^{-1} = 0$.

### C.2.1 Fully connected layers

For a fully connected layer, we have

$$\boldsymbol{V}(\boldsymbol{A},\boldsymbol{\Omega}) = \mathbf{Lin}_{(C[+1],K)}(\boldsymbol{\Omega})(\mathbf{Lin}_{(K,C[-1])}(\boldsymbol{A})(\boldsymbol{B})) = \boldsymbol{\Omega}\boldsymbol{A}^T\boldsymbol{B} \tag{47}$$

**Lemma C.1.** *Let* $\boldsymbol{Y} \in \mathbb{R}(t,n), \boldsymbol{X} \in \mathbb{R}(s,n), \boldsymbol{C} \in \mathbb{R}(t,s)$

*We define:*

$$\boldsymbol{S} := \frac{1}{n}\boldsymbol{X}\boldsymbol{X}^T \in \mathbb{R}(s,s) \tag{48}$$

$$\boldsymbol{N} := \frac{1}{n}\boldsymbol{X}\boldsymbol{Y}^T \in \mathbb{R}(s,t) \tag{49}$$

$$\frac{1}{n}\left|\left|\boldsymbol{C}\boldsymbol{X} - \boldsymbol{Y}\right|\right|^2 = \left|\left|\boldsymbol{C}\mathbf{S}^{\frac{1}{2}} - \boldsymbol{N}^T\mathbf{S}^{-\frac{1}{2}}\right|\right|^2 - \left|\left|\mathbf{S}^{-\frac{1}{2}}\boldsymbol{N}\right|\right|^2 + \frac{1}{n}\left|\left|\boldsymbol{Y}\right|\right|^2 \tag{50}$$

Proof in C.2.2

Hence using Lemma C.5 with $\boldsymbol{C} \leftarrow \boldsymbol{\Omega}\boldsymbol{A}^T$, $\boldsymbol{Y} \leftarrow \boldsymbol{V}_{\text{goal}_{proj}}$ and $\boldsymbol{B} \leftarrow \boldsymbol{X}$, we have:

$$\frac{1}{n}\left|\left|\mathbf{Lin}_{(C[+1],K)}(\boldsymbol{\Omega})(\mathbf{Lin}_{(K,C[-1])}(\boldsymbol{A})(\boldsymbol{B})) - \boldsymbol{V}_{\text{goal}_{proj}}\right|\right|^2$$
$$=$$
$$\frac{1}{n}\left|\left|\boldsymbol{\Omega}\boldsymbol{A}^T\boldsymbol{B} - \boldsymbol{V}_{\text{goal}_{proj}}\right|\right|^2 \tag{51}$$
$$=$$
$$\left|\left|\boldsymbol{\Omega}\boldsymbol{A}^T\mathbf{S}^{\frac{1}{2}} - \boldsymbol{N}^T\mathbf{S}^{-\frac{1}{2}}\right|\right|^2 - \left|\left|\mathbf{S}^{-\frac{1}{2}}\boldsymbol{N}\right|\right|^2 + \frac{1}{n}\left|\left|\boldsymbol{V}_{\text{goal}_{proj}}\right|\right|^2$$

With:

$$\boldsymbol{S} := \frac{1}{n}\boldsymbol{B}\boldsymbol{B}^T \in \mathbb{R}(C[-1], C[-1]) \tag{52}$$

$$\boldsymbol{N} := \frac{1}{n}\boldsymbol{B}\boldsymbol{V}_{\text{goal}_{proj}}^T \in \mathbb{R}(C[-1], C[+1]) \tag{53}$$

### C.2.2 Convolutional connected layers

We have $\boldsymbol{A} \in \mathbb{R}(K, C[-1], d, d)$ and $\boldsymbol{\Omega} \in \mathbb{R}(C[+1], K, d[+1], d[+1])$ where $d, d[+1]$ is the kernel size at $l$ and $l+1$. We note $\boldsymbol{A}_F$ the flatten and transposed version of $\boldsymbol{A}$ of shape $(C[-1]dd, K)$ and $\boldsymbol{\alpha}_k := \boldsymbol{A}_F[:, k] \in (C[-1]dd, 1)$. We will now consider $\boldsymbol{\Omega}$ with the last order flatten $i.e.$ $\boldsymbol{\Omega} \in \mathbb{R}(C[+1], K, d[+1]d[+1])$. We also note $\boldsymbol{\omega}_{k,m} := \boldsymbol{\Omega}[m, k] \in (d[+1]d[+1], 1)$. Using this we define $\boldsymbol{\Omega}[m]_F := \begin{pmatrix} \boldsymbol{\omega}_{1,m}^T \\ \vdots \\ \boldsymbol{\omega}_{K,m}^T \end{pmatrix} \in (K, d[+1]d[+1])$ and

$\boldsymbol{\Omega}_F := \begin{pmatrix} \boldsymbol{\Omega}[1]_F & \cdots & \boldsymbol{\Omega}[m]_F \end{pmatrix} \in (K, C[+1]d[+1]d[+1])$.

We define the tensor $\boldsymbol{T}$ such that for a pixel $j$ of the output of the convolutional layer, $\boldsymbol{T}_j$ is a linear application that select the pixels of the input of the convolutional layer that are used to compute the pixel $j$ of the output in a flatten version image (flatten only on the space not on the channels). $\boldsymbol{T} \in \mathbb{R}(H[+1]W[+1], d[+1]d[+1], HW)$ where $H$ and $W$ are the height and width of the intermediate image and $H[+1]$ and $W[+1]$ are the height and width of the output image.

As previously, we have $\boldsymbol{B}^c$ the unfolded version of $\boldsymbol{B}$ such that $\boldsymbol{B}^c \in \mathbb{R}(n, C[-1]dd, HW)$ satisfying $\mathbf{Conv}(\boldsymbol{B}_i)$ is equal with the correct reshape to $\boldsymbol{A}\boldsymbol{B}_i^c$.

In addition, we use $j$ as an index on the space of pixels instead of having a couple $h, w$ for height and width. With those notations we have:

$$\boldsymbol{V}(\boldsymbol{A}, \boldsymbol{\Omega})[i, m, j] = \mathbf{Conv}_{(C[+1],K)}(\boldsymbol{\Omega})(\mathbf{Conv}_{(K,C[-1])}(\boldsymbol{A})(\boldsymbol{B}_i))[m, j] \tag{54}$$

$$= \sum_k^K \boldsymbol{\omega}_{m,k}^T \boldsymbol{T}_j (\boldsymbol{B}_i^c)^T \boldsymbol{\alpha}_k \tag{55}$$

In the following for simplicity, we note $\boldsymbol{B}_{i,j}^t := \boldsymbol{T}_j(\boldsymbol{B}_i^c)^T$. To find the best neurons to add we solve the expressivity bottleneck as :

$$\arg\min_{\boldsymbol{A},\boldsymbol{\Omega}} \frac{1}{n} \sum_i \sum_j \sum_m \left\| \boldsymbol{V}_{\text{goal}_{\text{proj}_i}}^{(j,m)} - \sum_{k=1}^K \boldsymbol{\omega}_{m,k}^T \boldsymbol{B}_{i,j}^t \boldsymbol{\alpha}_k \right\|^2 \tag{56}$$

$$\tag{57}$$

Using the properties of the trace, it follows that :

$$\sum_{i,j,m} \left|\left| V_{\text{goal}_{\text{proj}_i}}^{(j,m)} - \sum_{k=1}^{K} \boldsymbol{\omega}_{m,k}^T \boldsymbol{B}_{i,j}^t \boldsymbol{\alpha}_k \right|\right|^2 = \sum_{i,j,m} \left|\left| V_{\text{goal}_{\text{proj}_i}}^{(j,m)} - \sum_{k} \text{Tr}\left( \boldsymbol{B}_{i,j}^t \boldsymbol{\alpha}_k \boldsymbol{\omega}_{m,k}^T \right) \right|\right|^2 \tag{58}$$

$$= \sum_{i,j,m} \left|\left| V_{\text{goal}_{\text{proj}_i}}^{(j,m)} - \text{Tr}\left( \boldsymbol{B}_{i,j}^t \overbrace{\sum_{k} \boldsymbol{\alpha}_k \boldsymbol{\omega}_{m,k}^T}^{\boldsymbol{F}_m} \right) \right|\right|^2 \tag{59}$$

$$= \sum_{i,j,m} \left|\left| V_{\text{goal}_{\text{proj}_i}}^{(j,m)} - flat(\boldsymbol{B}_{i,j}^t)^T flat(\boldsymbol{F}_m) \right|\right|^2 \tag{60}$$

$$= \sum_{i,j} \left|\left| V_{\text{goal}_{\text{proj}_i}}^{(j)} - flat(\boldsymbol{B}_{i,j}^t)^T \boldsymbol{F} \right|\right|^2 \tag{61}$$

$$\tag{62}$$

With $\boldsymbol{F} := \begin{pmatrix} flat(\boldsymbol{F}_1) & ... & flat(\boldsymbol{F}_{C[+1]}) \end{pmatrix}$.

We remark that $\boldsymbol{V}(\boldsymbol{A}, \boldsymbol{\Omega})$ is a linear function of the matrix $\boldsymbol{F}$ which implies that the solution of 56 is the same as the one for linear layer. Replacing $\boldsymbol{\Omega}\boldsymbol{A}$ by $\boldsymbol{F}$ in 47 and following the same reasoning as for linear layer, it follows that 56 is equivalent to :

$$\arg\min_{\boldsymbol{F}} \left|\left| \mathbf{S}^{\frac{1}{2}} \boldsymbol{F} - \mathbf{S}^{-\frac{1}{2}} \boldsymbol{N} \right|\right|_2 \tag{63}$$

with $\boldsymbol{S} := \sum_{i,j} flat(\boldsymbol{B}_{i,j}^t) flat(\boldsymbol{B}_{i,j}^t)^T$ and $\boldsymbol{N} := \sum_{i,j} \boldsymbol{V}_{\text{goal}_{\text{proj}}}^{j} flat(\boldsymbol{B}_{i,j}^t)^T$.

However, we remark that the dimension of $\boldsymbol{S} \in \mathbb{R}(C[-1]d[+1]^2 d^2, C[-1]d[+1]^2 d^2)$ is quite large and that computing the SVD of such matrix is costly. To avoid expensive computation, we approximate 56 by defining the matrix $\boldsymbol{S}$ and $\boldsymbol{N}$ as 98 and 100. We now prove that 3.2, 3.4 and Equation (10) still hold with such new definitions of $\boldsymbol{S}$ and $\boldsymbol{N}$.

**Lemma C.2.** *Let $r := \min(\boldsymbol{B}_{1,1}^t.shape)$, we define:*

$$\boldsymbol{S} := \frac{r}{n} \sum_{i=1}^{n} \sum_{j=1}^{H[+1]W[+1]} (\boldsymbol{B}_{i,j}^t)^T (\boldsymbol{B}_{i,j}^t) \in (C[-1]dd, C[-1]dd) \tag{64}$$

$$\boldsymbol{N}_m := \frac{1}{n} \sum_{i,j}^{n,H[+1]W[+1]} \boldsymbol{V}_{goal_{proj_{i,j,m}}} (\boldsymbol{B}_{i,j}^t)^T \in (C[-1]dd, d[+1]d[+1]) \tag{65}$$

$$\boldsymbol{N} := \begin{pmatrix} \boldsymbol{N}_1 \cdots \boldsymbol{N}_{C[+1]} \end{pmatrix} \in (C[-1]dd, C[+1]d[+1]d[+1]) \tag{66}$$

*We have:*

$$\frac{1}{n} \left|\left| \boldsymbol{V}(\boldsymbol{A}, \boldsymbol{\Omega}) - \boldsymbol{V}_{goal_{proj}} \right|\right|^2 \leqslant \left|\left| \mathbf{S}^{\frac{1}{2}} \boldsymbol{A}_F \boldsymbol{\Omega}_F - \mathbf{S}^{-\frac{1}{2}} \boldsymbol{N} \right|\right|^2 - \left|\left| \mathbf{S}^{-\frac{1}{2}} \boldsymbol{N} \right|\right|^2 + \frac{1}{n} \left|\left| \boldsymbol{V}_{goal_{proj}} \right|\right|^2 \tag{67}$$

Proof in C.2.2

**Lemma C.3.** *For $\boldsymbol{S} \in \mathbb{R}(s,s), \boldsymbol{N} \in \mathbb{R}(s,t), \boldsymbol{A} \in \mathbb{R}(s,K), \boldsymbol{\Omega} \in \mathbb{R}(K,t)$.*

*We note $\boldsymbol{U}\boldsymbol{\Lambda}\boldsymbol{V}$ the singular value decomposition of $\mathbf{S}^{-\frac{1}{2}}\boldsymbol{N}$ and $\boldsymbol{U}_K$ the first $K$ columns of $\boldsymbol{U}$, $V_K$ the first $K$ lines of $\boldsymbol{V}$, $\Lambda_K$ the first $K$ singular values of $\Lambda$ and $\Lambda_{K+1:}$ the other singular values of $\Lambda$.*

*We define:*

$$\boldsymbol{A}^* := \mathbf{S}^{-\frac{1}{2}} \boldsymbol{U}_K \sqrt{\Lambda_K} \tag{68}$$

$$\boldsymbol{\Omega}^* := \sqrt{\Lambda_K} V_K \tag{69}$$

*Then:*

$$\min_{\boldsymbol{A},\boldsymbol{\Omega}} \frac{1}{n} \left\|\boldsymbol{V}(\boldsymbol{A},\boldsymbol{\Omega}) - \boldsymbol{V}_{goal_{proj}}\right\|^2 \leqslant \frac{1}{n}\left\|\boldsymbol{V}(\boldsymbol{A}^*,\boldsymbol{\Omega}^*) - \boldsymbol{V}_{goal_{proj}}\right\|^2 = -\left\|\Lambda_K\right\|^2 + \frac{1}{n}\left\|\boldsymbol{V}_{goal_{proj}}\right\|^2 \tag{70}$$

*with equality in the linear case. Using expressivity bottlenecks notations, this rewrites:*

$$\Psi^l_{\theta \oplus \theta^{K*}_{\leftrightarrow}} \leqslant \Psi^l_\theta - \sum_{k=1}^{K} \lambda_k^2 \ . \tag{71}$$

*Proof.* Using Lemma C.5 and Lemma C.7:

$$\frac{1}{n}\left\|\boldsymbol{V}(\boldsymbol{A},\boldsymbol{\Omega}) - \boldsymbol{V}_{\text{goal}_{proj}}\right\|^2 \leqslant \left\|\mathbf{S}^{\frac{1}{2}}\boldsymbol{A}\boldsymbol{\Omega} - \mathbf{S}^{-\frac{1}{2}}\boldsymbol{N}\right\|^2 - \left\|\mathbf{S}^{-\frac{1}{2}}\boldsymbol{N}\right\|^2 + \frac{1}{n}\overbrace{\left\|\boldsymbol{V}_{\text{goal}_{proj}}\right\|^2}^{\Psi^l_\theta} \tag{72}$$

Hence we minimize the second term of the right term, that is:

$$\arg\min_{\boldsymbol{A},\boldsymbol{\Omega}} \left\|\mathbf{S}^{\frac{1}{2}}\boldsymbol{A}\boldsymbol{\Omega} - \mathbf{S}^{-\frac{1}{2}}\boldsymbol{N}\right\| \tag{73}$$

As we suppose that $S$ is invertible, we can use the change of variable $\widetilde{\boldsymbol{A}} = \mathbf{S}^{\frac{1}{2}}\boldsymbol{A}$ thus we have:

$$\min_{\boldsymbol{A},\boldsymbol{\Omega}} \left\|\mathbf{S}^{\frac{1}{2}}\boldsymbol{A}\boldsymbol{\Omega} - \mathbf{S}^{-\frac{1}{2}}\boldsymbol{N}\right\| = \min_{\widetilde{\boldsymbol{A}},\boldsymbol{\Omega}} \left\|\widetilde{\boldsymbol{A}}\boldsymbol{\Omega} - \mathbf{S}^{-\frac{1}{2}}\boldsymbol{N}\right\| \tag{74}$$

The solution of such problems is given by the paper Eckart & Young (1936) and is:

$$\widetilde{\boldsymbol{A}}^* = \boldsymbol{U}_K\sqrt{\Lambda_K} \tag{75}$$

$$\boldsymbol{\Omega}^* = \sqrt{\Lambda_K}\boldsymbol{V}_K \tag{76}$$

To recover $\boldsymbol{A}^*$ we simply have to multiply by $\mathbf{S}^{-\frac{1}{2}}$ on the left side of $\widetilde{\boldsymbol{A}}^*$. By definition of the SVD and the construction of $(\boldsymbol{A}^*,\boldsymbol{\Omega}^*)$ we have:

$$\left\|\mathbf{S}^{\frac{1}{2}}\boldsymbol{A}^*\boldsymbol{\Omega}^* - \mathbf{S}^{-\frac{1}{2}}\boldsymbol{N}\right\|^2 - \left\|\mathbf{S}^{-\frac{1}{2}}\boldsymbol{N}\right\|^2 = \left\|\Lambda_{K+1:}\right\|^2 - \left\|\Lambda\right\|^2 = -\left\|\Lambda_K\right\|^2 \tag{77}$$

Using this and Equation (72) we immediately get the desired Equation (70). To conclude, we can also rewrite this with the bottleneck expression:

$$\Psi_{\theta \oplus \theta^K_{\leftrightarrow}} := \min_{\boldsymbol{A},\boldsymbol{\Omega}} \frac{1}{n}\left\|\boldsymbol{V}(\boldsymbol{A},\boldsymbol{\Omega}) - \boldsymbol{V}_{\text{goal}_{proj}}\right\|^2 \leqslant \Psi^l_\theta - \sum_{k=1}^{K} \lambda_k^2 \tag{78}$$

$\square$

We now prove all the lemmas.

**Lemma C.4.** *For $\boldsymbol{S} \in \mathbb{R}(s,s), \boldsymbol{N} \in \mathbb{R}(s,t), \boldsymbol{C} \in \mathbb{R}(t,s)$.*

*If $\boldsymbol{N} = \mathbf{S}^{\frac{1}{2}}\mathbf{S}^{-\frac{1}{2}}\boldsymbol{N}$, we have:*

$$\left\langle \boldsymbol{C}^T, \boldsymbol{S}\boldsymbol{C}^T\right\rangle - 2\left\langle \boldsymbol{N}, \boldsymbol{C}^T\right\rangle = \left\|\mathbf{S}^{\frac{1}{2}}\boldsymbol{C}^T - \mathbf{S}^{-\frac{1}{2}}\boldsymbol{N}\right\|^2 - \left\|\mathbf{S}^{-\frac{1}{2}}\boldsymbol{N}\right\|^2 \tag{79}$$

*Proof.*    • For the first term we have:

$$\left\langle \boldsymbol{C}^T, \boldsymbol{S}\boldsymbol{C}^T \right\rangle = \left\langle \boldsymbol{C}^T, \mathbf{S}^{\frac{1}{2}}\mathbf{S}^{\frac{1}{2}}\boldsymbol{C}^T \right\rangle \tag{80}$$

$$= \left\langle \mathbf{S}^{\frac{1}{2}}\boldsymbol{C}^T, \mathbf{S}^{\frac{1}{2}}\boldsymbol{C}^T \right\rangle \tag{81}$$

$$= \left\| \mathbf{S}^{\frac{1}{2}}\boldsymbol{C}^T \right\|^2 \tag{82}$$

• For the second term we have:

$$\left\langle \boldsymbol{N}, \boldsymbol{C}^T \right\rangle = \left\langle \mathbf{S}^{\frac{1}{2}}\mathbf{S}^{-\frac{1}{2}}\boldsymbol{N}, \boldsymbol{C}^T \right\rangle \tag{83}$$

$$= \left\langle \mathbf{S}^{-\frac{1}{2}}\boldsymbol{N}, \mathbf{S}^{\frac{1}{2}}\boldsymbol{C}^T \right\rangle \tag{84}$$

Hence we have that:

$$\left\langle \boldsymbol{C}^T, \boldsymbol{S}\boldsymbol{C}^T \right\rangle - 2\left\langle \boldsymbol{N}, \boldsymbol{C}^T \right\rangle = \left\| \mathbf{S}^{\frac{1}{2}}\boldsymbol{C}^T \right\|^2 - 2\left\langle \mathbf{S}^{-\frac{1}{2}}\boldsymbol{N}, \mathbf{S}^{\frac{1}{2}}\boldsymbol{C}^T \right\rangle + \left\| \mathbf{S}^{-\frac{1}{2}}\boldsymbol{N} \right\|^2 - \left\| \mathbf{S}^{-\frac{1}{2}}\boldsymbol{N} \right\|^2 \tag{85}$$

$$= \left\| \mathbf{S}^{\frac{1}{2}}\boldsymbol{C}^T - \mathbf{S}^{-\frac{1}{2}}\boldsymbol{N} \right\|^2 - \left\| \mathbf{S}^{-\frac{1}{2}}\boldsymbol{N} \right\|^2 \tag{86}$$

$\square$

**Lemma C.5.** *Let* $\boldsymbol{Y} \in \mathbb{R}(t,n), \boldsymbol{X} \in \mathbb{R}(s,n), \boldsymbol{C} \in \mathbb{R}(t,s)$

*We define:*

$$\boldsymbol{S} := \frac{1}{n}\boldsymbol{X}\boldsymbol{X}^T \in \mathbb{R}(s,s) \tag{87}$$

$$\boldsymbol{N} := \frac{1}{n}\boldsymbol{X}\boldsymbol{Y}^T \in \mathbb{R}(s,t) \tag{88}$$

$$\frac{1}{n}\left\| \boldsymbol{C}\boldsymbol{X} - \boldsymbol{Y} \right\|^2 = \left\| \boldsymbol{C}\mathbf{S}^{\frac{1}{2}} - \boldsymbol{N}^T\mathbf{S}^{-\frac{1}{2}} \right\|^2 - \left\| \mathbf{S}^{-\frac{1}{2}}\boldsymbol{N} \right\|^2 + \frac{1}{n}\left\| \boldsymbol{Y} \right\|^2 \tag{89}$$

*Proof.* By developing the scalar product we get:

$$\frac{1}{n}\left\| \boldsymbol{C}\boldsymbol{X} - \boldsymbol{Y} \right\|^2 = \frac{1}{n}\left\| \boldsymbol{Y} \right\|^2 - 2\left\langle \boldsymbol{Y}, \frac{1}{n}\boldsymbol{C}\boldsymbol{X} \right\rangle + \frac{1}{n}\left\| \boldsymbol{C}\boldsymbol{X} \right\|^2 \tag{90}$$

$$= \frac{1}{n}\left\| \boldsymbol{Y} \right\|^2 - 2\left\langle \boldsymbol{Y}, \frac{1}{n}\boldsymbol{C}\boldsymbol{X} \right\rangle + \frac{1}{n}\left\langle \boldsymbol{C}\boldsymbol{X}, \boldsymbol{C}\boldsymbol{X} \right\rangle \tag{91}$$

$$= \frac{1}{n}\left\| \boldsymbol{Y} \right\|^2 - 2\left\langle \boldsymbol{Y}^T, \frac{1}{n}(\boldsymbol{C}\boldsymbol{X})^T \right\rangle + \frac{1}{n}\left\langle (\boldsymbol{C}\boldsymbol{X})^T, (\boldsymbol{C}\boldsymbol{X})^T \right\rangle \tag{92}$$

$$= \frac{1}{n}\left\| \boldsymbol{Y} \right\|^2 - 2\left\langle \frac{1}{n}\boldsymbol{X}\boldsymbol{Y}^T, \boldsymbol{C}^T \right\rangle + \left\langle \boldsymbol{C}^T, \frac{1}{n}\boldsymbol{X}\boldsymbol{X}^T\boldsymbol{C}^T \right\rangle \tag{93}$$

We now use the two following lemma:

**Lemma C.6.** *Let* $\boldsymbol{Y} \in \mathbb{R}(t,n), \boldsymbol{X} \in \mathbb{R}(s,n)$ *and* $\boldsymbol{S} := \boldsymbol{X}\boldsymbol{X}^T \in \mathbb{R}(s,s)$.

$$\mathbf{S}^{\frac{1}{2}}\mathbf{S}^{-\frac{1}{2}}\boldsymbol{X}\boldsymbol{Y}^T = \boldsymbol{X}\boldsymbol{Y}^T \tag{94}$$

*Proof.* Let decompose $\boldsymbol{Y}$ on $\text{Im}(\boldsymbol{X}^T) \oplus_{\perp} \ker(\boldsymbol{X})$: $\boldsymbol{Y} = \boldsymbol{X}^T I + \boldsymbol{K}$.

$$\boldsymbol{X}\boldsymbol{Y}^T = \boldsymbol{X}\boldsymbol{X}^T I + \boldsymbol{X}K = \boldsymbol{X}\boldsymbol{X}^T I = \boldsymbol{S}I$$

Hence $\boldsymbol{X}\boldsymbol{Y}^T \in \text{Im}(\boldsymbol{S})$, hence as $\boldsymbol{S}_{|\,\text{Im}(\boldsymbol{S})}$ is invertible, we have: $\mathbf{S}^{-\frac{1}{2}}\mathbf{S}^{\frac{1}{2}}\boldsymbol{X}\boldsymbol{Y}^T = \boldsymbol{X}\boldsymbol{Y}^T$.    $\square$

Continuing the demonstration from Equation (93), by applying Lemma C.4, we have:

$$\frac{1}{n}\left|\left|\boldsymbol{CX} - \boldsymbol{Y}\right|\right|^2 = \frac{1}{n}\left|\left|\boldsymbol{Y}\right|\right|^2 - 2\left\langle\boldsymbol{N}, \boldsymbol{C}^T\right\rangle + \left\langle\boldsymbol{C}^T, \boldsymbol{SC}^T\right\rangle \tag{95}$$

$$= \frac{1}{n}\left|\left|\boldsymbol{Y}\right|\right|^2 + \left|\left|\mathbf{S}^{\frac{1}{2}}\boldsymbol{C}^T - \mathbf{S}^{-\frac{1}{2}}\boldsymbol{N}\right|\right|^2 - \left|\left|\mathbf{S}^{-\frac{1}{2}}\boldsymbol{N}\right|\right|^2 \tag{96}$$

$$= \frac{1}{n}\left|\left|\boldsymbol{Y}\right|\right|^2 + \left|\left|\boldsymbol{C}\mathbf{S}^{\frac{1}{2}} - \boldsymbol{N}^T\mathbf{S}^{-\frac{1}{2}}\right|\right|^2 - \left|\left|\mathbf{S}^{-\frac{1}{2}}\boldsymbol{N}\right|\right|^2 \tag{97}$$

$$\square$$

**Lemma C.7.** *Let $r := \min(\boldsymbol{B}_{1,1}^t.shape)$, we define:*

$$\boldsymbol{S} := \frac{r}{n}\sum_{i=1}^{n}\sum_{j=1}^{H[+1]W[+1]}(\boldsymbol{B}_{i,j}^t)^T(\boldsymbol{B}_{i,j}^t) \in (C[-1]dd, C[-1]dd) \tag{98}$$

$$\boldsymbol{N}_m := \frac{1}{n}\sum_{i,j}^{n,H[+1]W[+1]}\boldsymbol{V}_{goal_{proj_{i,j,m}}}(\boldsymbol{B}_{i,j}^t)^T \in (C[-1]dd, d[+1]d[+1]) \tag{99}$$

$$\boldsymbol{N} := \left(\boldsymbol{N}_1\cdots\boldsymbol{N}_{C[+1]}\right) \in (C[-1]dd, C[+1]d[+1]d[+1]) \tag{100}$$

*We have:*

$$\frac{1}{n}\left|\left|\boldsymbol{V}(\boldsymbol{A}, \boldsymbol{\Omega}) - \boldsymbol{V}_{goal_{proj}}\right|\right|^2 \leqslant \left|\left|\mathbf{S}^{\frac{1}{2}}\boldsymbol{A}_F\boldsymbol{\Omega}_F - \mathbf{S}^{-\frac{1}{2}}\boldsymbol{N}\right|\right|^2 - \left|\left|\mathbf{S}^{-\frac{1}{2}}\boldsymbol{N}\right|\right|^2 + \frac{1}{n}\left|\left|\boldsymbol{V}_{goal_{proj}}\right|\right|^2 \tag{101}$$

*Proof.* We have:

$$\frac{1}{n}\left|\left|\boldsymbol{V}(\boldsymbol{A}, \boldsymbol{\Omega}) - \boldsymbol{V}_{\text{goal}_{proj}}\right|\right|^2 = \frac{1}{n}\left|\left|\boldsymbol{V}(\boldsymbol{A}, \boldsymbol{\Omega})\right|\right|^2 - \frac{2}{n}\left\langle\boldsymbol{V}(\boldsymbol{A}, \boldsymbol{\Omega}), \boldsymbol{V}_{\text{goal}_{proj}}\right\rangle^2 + \frac{1}{n}\left|\left|-\boldsymbol{V}_{\text{goal}_{proj}}\right|\right|^2 \tag{102}$$

We will now simplify the first two terms separately.

**Norm simplification**

**Lemma C.8.** *For any square matrix $\boldsymbol{A} \in \mathbb{R}^{(n,n)}, \mathrm{Tr}(\boldsymbol{A})^2 \leqslant \mathrm{rank}(\boldsymbol{A})\left|\left|\boldsymbol{A}\right|\right|^2$.*

*Proof.* Using the truncated SVD we have $\boldsymbol{A} = \boldsymbol{U}\Sigma\boldsymbol{V}$ with $\Sigma$ a diagonal and $\boldsymbol{U} \in \mathbb{R}^{(n,\mathrm{rank}(\boldsymbol{A}))}, \boldsymbol{V} \in \mathbb{R}^{(\mathrm{rank}(\boldsymbol{A}),n)}$ truncated orthonormal matrices.

We have:

$$\mathrm{Tr}(\boldsymbol{A})^2 = \mathrm{Tr}(\boldsymbol{U}\Sigma\boldsymbol{V})^2 \tag{103}$$

$$= \mathrm{Tr}(\boldsymbol{V}\boldsymbol{U}\Sigma)^2 \tag{104}$$

$$= \left\langle\boldsymbol{V}\boldsymbol{U}, \Sigma\right\rangle^2 \tag{105}$$

$$(\text{Cauchy-Swarz}) \quad \leqslant \left|\left|\boldsymbol{V}\boldsymbol{U}\right|\right|^2\left|\left|\Sigma\right|\right|^2 \tag{106}$$

As $\boldsymbol{U}, \boldsymbol{V}$ are truncated orthonormal matrices, we have:

$$\left|\left|\boldsymbol{V}\boldsymbol{U}\right|\right|^2 = \mathrm{Tr}(\boldsymbol{U}^T\boldsymbol{V}^T\boldsymbol{V}\boldsymbol{U}) = \mathrm{Tr}(\boldsymbol{U}^T\boldsymbol{U}) = \mathrm{Tr}(I_{\mathrm{rank}(\boldsymbol{A})}) = \mathrm{rank}(\boldsymbol{A})$$

Hence:

$$\mathrm{Tr}(\boldsymbol{A})^2 \leqslant \mathrm{rank}(\boldsymbol{A})\left|\left|\Sigma\right|\right|^2 \tag{107}$$

As $\boldsymbol{U}, \boldsymbol{V}$ are truncated orthonormal matrices, we have:

$$||\Sigma||^2 = \text{Tr}(\Sigma^T \Sigma) = \text{Tr}((\boldsymbol{V}\Sigma\boldsymbol{U})\boldsymbol{U}\Sigma\boldsymbol{V}) = \text{Tr}(\boldsymbol{A}^T\boldsymbol{A}) = ||\boldsymbol{A}||^2$$

We conclude that:

$$\text{Tr}(\boldsymbol{A})^2 \leqslant \text{rank}(\boldsymbol{A}) \, ||\boldsymbol{A}||^2 \tag{108}$$

$\square$

**Lemma C.9.** *For $\boldsymbol{M} \in (m,n), (\boldsymbol{u}_k)_{k\in[[K]]} \in (m)^K, (\boldsymbol{v}_k)_{k\in[[K]]} \in (n)^K$ and with $\boldsymbol{W} := \sum_{k\in[\![K]\!]} \boldsymbol{v}_k \boldsymbol{u}_k^T \in (n,m)$ we have:*

$$\left\| \sum_{k\in[\![K]\!]} \boldsymbol{u}_k^T \boldsymbol{M} \boldsymbol{v}_k \right\|^2 = \text{Tr}\,(\boldsymbol{M}\boldsymbol{W})^2 \tag{109}$$

*Proof.* Let $i \in I$:

$$\left\| \sum_{k\in[\![K]\!]} \boldsymbol{v}\boldsymbol{u}_k^T \boldsymbol{M} \boldsymbol{v}_k \right\|^2 = \left( \sum_{k\in[\![K]\!]} \boldsymbol{u}_k^T \boldsymbol{M} \boldsymbol{v}_k \right)^2 \tag{110}$$

$$= \text{Tr} \left( \sum_{k\in[\![K]\!]} \boldsymbol{u}_k^T \boldsymbol{M} \boldsymbol{v}_k \right)^2 \tag{111}$$

$$= \text{Tr} \left( \boldsymbol{M} \sum_{k\in[\![K]\!]} \boldsymbol{v}_k \boldsymbol{u}_k^T \right)^2 \tag{112}$$

$$= \text{Tr}\,(\boldsymbol{M}\boldsymbol{W})^2 \tag{113}$$

$\square$

**Lemma C.10.** *For $(\boldsymbol{M}_i)_{i\in I} \in (m,n)^I$ such that $\forall i \in I, \text{rank}(\boldsymbol{M}_i\boldsymbol{W}) \leqslant H$ and with $\boldsymbol{W} \in (n,m)$ we have:*

$$\sum_{i\in I} \text{Tr}\,(\boldsymbol{M}_i\boldsymbol{W})^2 \leqslant \left\langle \boldsymbol{W}, H \sum_{i\in I} \boldsymbol{M}_i^T \boldsymbol{M}_i \boldsymbol{W} \right\rangle \tag{114}$$

*Proof.* Let $i \in I$:

Using Lemma C.8 with $\boldsymbol{A} \leftarrow \boldsymbol{M}_i\boldsymbol{W}$

$$\text{Tr}\,(\boldsymbol{M}_i\boldsymbol{W})^2 \leqslant \text{rank}(\boldsymbol{M}_i\boldsymbol{W})||\boldsymbol{M}_i\boldsymbol{W}||^2 \tag{115}$$

$$\leqslant H\,||\boldsymbol{M}_i\boldsymbol{W}||^2 \quad H := \min(\boldsymbol{M}_i.shape) \tag{116}$$

Hence we have:

$$\sum_{i\in I} \left\| \sum_{k\in[\![K]\!]} \boldsymbol{u}_k^T \boldsymbol{M}_i \boldsymbol{v}_k \right\|^2 \leqslant H \sum_{i\in I} ||\boldsymbol{M}_i\boldsymbol{W}||^2 \tag{117}$$

$$= H \sum_{i\in I} \langle \boldsymbol{M}_i\boldsymbol{W}, \boldsymbol{M}_i\boldsymbol{W} \rangle \tag{118}$$

$$= H \sum_{i\in I} \langle \boldsymbol{W}, \boldsymbol{M}_i^T \boldsymbol{M}_i \boldsymbol{W} \rangle \tag{119}$$

$$= H \left\langle \boldsymbol{W}, \sum_{i\in I} \boldsymbol{M}_i^T \boldsymbol{M}_i \boldsymbol{W} \right\rangle \tag{120}$$

$$\square$$

Using first Lemma C.9 with $\boldsymbol{u}_k \leftarrow \boldsymbol{\omega}_{k,m}$, $\boldsymbol{v}_k \leftarrow \boldsymbol{\alpha}_k$ and $\boldsymbol{M} \leftarrow \boldsymbol{B}_{i,j}^t$, we have:

$$\frac{1}{n} \left\| \boldsymbol{V}(\boldsymbol{A}, \boldsymbol{\Omega}) \right\|^2 = \frac{1}{n} \sum_m^{C[+1]} \sum_{i,j}^{n,H[+1]W[+1]} \left\| \sum_k^K \boldsymbol{\omega}_{k,m}^T (\boldsymbol{B}_{i,j}^t) \boldsymbol{\alpha}_k \right\|^2 \tag{121}$$

$$= \frac{1}{n} \sum_m^{C[+1]} \sum_{i,j}^{n,H[+1]W[+1]} \mathrm{Tr} \left( \boldsymbol{B}_{i,j}^t \sum_k^K \boldsymbol{\alpha}_k \boldsymbol{\omega}_{k,m}^T \right)^2 \tag{122}$$

$$= \frac{1}{n} \sum_m^{C[+1]} \sum_{i,j}^{n,H[+1]W[+1]} \mathrm{Tr} \left( \boldsymbol{B}_{i,j}^t \boldsymbol{A}_F \boldsymbol{\Omega}[m]_F \right)^2 \tag{123}$$

Using Lemma C.10 with $i \leftarrow (i,j)$, $\boldsymbol{M}_i \leftarrow \boldsymbol{B}_{i,j}^t$ and $\boldsymbol{W} \leftarrow \boldsymbol{A}_F \boldsymbol{\Omega}[m]_F$:

$$\leqslant \sum_m^{C[+1]} \left\langle \boldsymbol{A}_F \boldsymbol{\Omega}[m]_F , \frac{r}{n} \sum_{i,j}^{n,H[+1]W[+1]} (\boldsymbol{B}_{i,j}^t)^T (\boldsymbol{B}_{i,j}^t) \boldsymbol{A}_F \boldsymbol{\Omega}[m]_F \right\rangle \tag{124}$$

$$= \sum_m^{C[+1]} \left\langle \boldsymbol{A}_F \boldsymbol{\Omega}[m]_F, \boldsymbol{S} \boldsymbol{A}_F \boldsymbol{\Omega}[m]_F \right\rangle \tag{125}$$

$$= \left\langle \boldsymbol{A}_F \boldsymbol{\Omega}_F, \boldsymbol{S} \boldsymbol{A}_F \boldsymbol{\Omega}_F \right\rangle \tag{126}$$

**Scalar product simplification**

**Lemma C.11.** *For* $\boldsymbol{M} \in (m,n), \boldsymbol{u} \in (m), \boldsymbol{v} \in (n)$ *we have:*

$$\boldsymbol{u}^T \boldsymbol{M} \boldsymbol{v} = \left\langle \boldsymbol{v} \boldsymbol{u}^T, \boldsymbol{M}^T \right\rangle \tag{127}$$

*Proof.*

$$\boldsymbol{u}^T \boldsymbol{M} \boldsymbol{v} = (\boldsymbol{u}^T \boldsymbol{M} v)^T \tag{128}$$

$$= \boldsymbol{v}^T \boldsymbol{M}^T \boldsymbol{u} \tag{129}$$

$$= \left\langle \boldsymbol{v}, \boldsymbol{M}^T \boldsymbol{u} \right\rangle \tag{130}$$

$$= \left\langle \boldsymbol{v} \boldsymbol{u}^T, \boldsymbol{M}^T \right\rangle \tag{131}$$

$$\square$$

We have:

$$\frac{1}{n} \left\langle \boldsymbol{V}(\boldsymbol{A}, \boldsymbol{\Omega}), \boldsymbol{V}_{\mathrm{goal}_{proj}} \right\rangle^2 = \frac{1}{n} \sum_m^{C[+1]} \sum_{i,j}^{n,H[+1]W[+1]} \sum_k^K \boldsymbol{\omega}_{k,m}^T \boldsymbol{B}_{i,j}^t \boldsymbol{\alpha}_k \boldsymbol{V}_{\mathrm{goal}_{proj}\,_{i,j,m}} \tag{132}$$

$$(\boldsymbol{V}_{\mathrm{goal}_{proj}\,_{i,j,m}} \in (1)) \quad = \sum_m^{C[+1]} \sum_k^K \boldsymbol{\omega}_{k,m}^T \frac{1}{n} \sum_{i,j}^{n,H[+1]W[+1]} (\boldsymbol{B}_{i,j}^t \boldsymbol{V}_{\mathrm{goal}_{proj}\,_{i,j,m}}) \boldsymbol{\alpha}_k \tag{133}$$

Using Lemma C.11 with $\boldsymbol{M} \leftarrow \sum_{i,j}^{n,H[+1]W[+1]} \boldsymbol{V}_{\text{goal}_{\text{proj}_{i,j,m}}} (\boldsymbol{B}_{i,j}^t)^T$

$$= \sum_m^{C[+1]} \left\langle \sum_k^K \boldsymbol{\alpha}_k \boldsymbol{\omega}_{k,m}^T, \frac{1}{n} \sum_{i,j}^{n,H[+1]W[+1]} \boldsymbol{V}_{\text{goal}_{proj_{i,j,m}}} (\boldsymbol{B}_{i,j}^t)^T \right\rangle \tag{134}$$

$$= \sum_m^{C[+1]} \left\langle \boldsymbol{A}_F \boldsymbol{\Omega}[m]_F, \boldsymbol{N}_m \right\rangle \tag{135}$$

$$= \left\langle \boldsymbol{A}_F \boldsymbol{\Omega}_F, \boldsymbol{N} \right\rangle \tag{136}$$

**Conclusion** In total, we get:

$$\frac{1}{n} \left|\left| \boldsymbol{V}(\boldsymbol{A}, \boldsymbol{\Omega}) - \boldsymbol{V}_{\text{goal}_{proj}} \right|\right|^2 \leqslant \left\langle \boldsymbol{A}_F \boldsymbol{\Omega}_F, \boldsymbol{S} \boldsymbol{A}_F \boldsymbol{\Omega}_F \right\rangle - 2 \left\langle \boldsymbol{A}_F \boldsymbol{\Omega}_F, \boldsymbol{N} \right\rangle + \frac{1}{n} \left|\left| \boldsymbol{V}_{\text{goal}_{proj}} \right|\right|^2 \tag{137}$$

If we suppose that $\boldsymbol{S}$ is invertible, we can apply Lemma C.4 and get the result.

$\square$

---

**Proposition C.2.** *Solving 8 is equivalent to minimizing the loss $\mathcal{L}$ at order one in $\boldsymbol{V}^l$. Furthermore performing an update of architecture with $\gamma \delta \boldsymbol{W}^*$ (5) and a neuron addition with $\gamma \theta_{\leftrightarrow}^{K*}$ (3.2), has an impact on the loss at first order in $\gamma$ as :*

$$\mathcal{L}(f_{\theta \oplus \theta_{\leftrightarrow}^K}) = \mathcal{L}(f_\theta) - \gamma \left( \sigma_{l-1}'(0) \Delta_{\theta_{\leftrightarrow}^{K,*}} + \Delta_{\delta \boldsymbol{W}^*} \right) + o(\gamma) \tag{138}$$

*with*

$$\Delta_{\theta_{\leftrightarrow}^{K,*}} := \frac{1}{n} \left\langle \boldsymbol{V}_{goal_{proj}}^l, \boldsymbol{V}^l(\theta_{\leftrightarrow}^{K,*}) \right\rangle_{\text{Tr}} = \sum_{k=1}^K \lambda_k^2 \tag{139}$$

$$\Delta_{\delta \boldsymbol{W}^*} := \frac{1}{n} \left\langle \boldsymbol{V}_{goal}^l, \boldsymbol{V}^l(\delta \boldsymbol{W}^*) \right\rangle_{\text{Tr}} \geqslant 0 \tag{140}$$

---

To prove such proposition we use the following lemma :

**Lemma C.12.** *We note $\boldsymbol{V}(\boldsymbol{A}, \boldsymbol{\Omega})$ the result of $\boldsymbol{B}$ after applying the layers parameterized by $\boldsymbol{A}$ and $\boldsymbol{\Omega}$. Let us consider $\boldsymbol{V}(\boldsymbol{A}^*, \boldsymbol{\Omega}^*)$ where $\boldsymbol{A}^*$ and $\boldsymbol{\Omega}^*$ are defined in 3.2. Then:*

$$\frac{1}{n} \left\langle \boldsymbol{V}_{goal_{proj}}, \boldsymbol{V}(\boldsymbol{A}^*, \boldsymbol{\Omega}^*) \right\rangle = ||\Lambda_K||^2 \tag{141}$$

*Proof.* Starting from Lemma C.3

$$\frac{1}{n} \left|\left| \boldsymbol{V}(\boldsymbol{A}^*, \boldsymbol{\Omega}^*) - \boldsymbol{V}_{\text{goal}_{\text{proj}}} \right|\right|^2 = - ||\Lambda_K||^2 + \frac{1}{n} \left|\left| \boldsymbol{V}_{\text{goal}_{\text{proj}}} \right|\right|^2 \tag{142}$$

Hence by developing the norm, we have:

$$\frac{1}{n} ||\boldsymbol{V}(\boldsymbol{A}^*, \boldsymbol{\Omega}^*)||^2 - \frac{2}{n} \left\langle \boldsymbol{V}(\boldsymbol{A}^*, \boldsymbol{\Omega}^*), \boldsymbol{V}_{\text{goal}_{\text{proj}}} \right\rangle = - ||\Lambda_K||^2 \tag{143}$$

Moreover by construction we have $\frac{1}{n} ||\boldsymbol{V}(\boldsymbol{A}^*, \boldsymbol{\Omega}^*)||^2 = ||\Lambda_K||^2$ and therefore we get:

$$-\frac{2}{n} \left\langle \boldsymbol{V}(\boldsymbol{A}^*, \boldsymbol{\Omega}^*), \boldsymbol{V}_{\text{goal}_{\text{proj}}} \right\rangle = -2 ||\Lambda_K||^2 \tag{144}$$

which concludes the proof. $\square$

We now prove the main proposition. Suppose that each quantity is added to the architecture with an amplitude factor $\gamma$ *i.e.* the best update is then $\gamma \, \delta \boldsymbol{W}^*$ and the new neurons are $\{\sqrt{\gamma} \boldsymbol{\alpha}_i^*, \sqrt{\gamma} \boldsymbol{\omega}_i^*\}_i$.

Using the Fréchet derivative on $\gamma$, we have the following:

$$\mathcal{L}(\boldsymbol{a}^l + \gamma \, \delta \boldsymbol{a}^l) = \mathcal{L}(\boldsymbol{a}^l) + \langle \nabla_{\boldsymbol{a}^l} \mathcal{L}, \gamma \, \delta \boldsymbol{a}^l \rangle + o(\gamma) \tag{145}$$

On one hand, performing an update of architecture, ie $\boldsymbol{W}^* \leftarrow \boldsymbol{W} + \gamma \, \delta \boldsymbol{W}^*$, changes the activation function $\boldsymbol{a}^l$ by $\gamma \, \delta \boldsymbol{a}_u^l := \boldsymbol{V}(\gamma \, \delta \boldsymbol{W}^*)$. Then, as explained in Appendix A.5, adding neurons $(\boldsymbol{A}^*, \boldsymbol{\Omega}^*)$ at layer $l-1$ changes the activation function $\boldsymbol{a}^l$ by :

$$\gamma \, \delta \boldsymbol{a}_a^l = \sigma'(0) \, \gamma \, \boldsymbol{V}(\boldsymbol{A}^*, \boldsymbol{\Omega}^*) + o(\gamma) \ . \tag{146}$$

We now suppose $\delta \boldsymbol{a}_u^l \neq -\delta \boldsymbol{a}_a^l$ and perform a first order development in $\gamma$. Then combining Equations (145) and (146), we have :

$$\mathcal{L}(\boldsymbol{A}^*, \boldsymbol{\Omega}^*) = \mathcal{L} + \langle \nabla_{\boldsymbol{a}^l} \mathcal{L}, \gamma \left(\delta \boldsymbol{a}_u^l + \delta \boldsymbol{a}_a^l\right) \rangle + o(\gamma) \ . \tag{147}$$

Using that $\boldsymbol{v}_{\text{goal}}(\boldsymbol{x}_i) := -\nabla_{\boldsymbol{a}^l(\boldsymbol{x}_i)} \ell(\boldsymbol{x}_i)$ and that $\mathcal{L} = \frac{1}{n} \sum_i \ell(\boldsymbol{x}_i)$, it follows that :

$$\mathcal{L}(\boldsymbol{A}^*, \boldsymbol{\Omega}^*) = \mathcal{L} - \frac{1}{n} \langle \boldsymbol{V}_{\text{goal}}, \gamma \left(\delta \boldsymbol{a}_u^l + \delta \boldsymbol{a}_a^l\right) \rangle + o(\gamma) \tag{148}$$

$$= \mathcal{L} - \frac{\gamma}{n} \Big( \langle \boldsymbol{V}_{\text{goal}}, \delta \boldsymbol{a}_u^l \rangle + \langle \boldsymbol{V}_{\text{goal}} - \boldsymbol{V}(\gamma \delta \boldsymbol{W}^*), \delta \boldsymbol{a}_a^l \rangle + \gamma \langle \delta \boldsymbol{a}_u^l, \delta \boldsymbol{a}_a^l \rangle \Big) + o(\gamma) \ . \tag{149}$$

Using C.12 we have :

$$\mathcal{L}(\boldsymbol{A}^*, \boldsymbol{\Omega}^*) = \mathcal{L} - \gamma \left( \sigma'(0) \sum_{k=1}^{K} \lambda_k^2 + \frac{1}{n} \langle \boldsymbol{V}_{\text{goal}}, \delta \boldsymbol{a}_u^l \rangle \right) + o(\gamma) \ . \tag{150}$$

**Note on the approximation for convolutional layer.** By developing the expression $||\boldsymbol{V} - \boldsymbol{V}_{\text{goal}_{proj}}||^2$, we remark that minimizing $||\boldsymbol{V} - \boldsymbol{V}_{\text{goal}_{proj}}||^2$ over $\boldsymbol{V}$ is equivalent to maximizing $\langle \boldsymbol{V}, \boldsymbol{V}_{\text{goal}_{proj}} \rangle$ with a constraint on the norm of $\boldsymbol{V}$. This constraint lies in the functional space of the activities and can be reformulated in the parameter space with the matrix $\boldsymbol{S}$ as $||\boldsymbol{A}\boldsymbol{\Omega}^T||_{\boldsymbol{S}} = ||\boldsymbol{V}||$. By changing the matrix $\boldsymbol{S}$ for another positive semi-definite matrix $\boldsymbol{S}_{pseudo}$, we modify the metric on $\boldsymbol{V}$ and obtain a pseudo-solution $\boldsymbol{S}_{pseudo}^{-1}\boldsymbol{N}$.

## C.3 Proposition 3.4 and Corollary 3.2

**Proposition C.3.** *3.4 Suppose $\boldsymbol{S}$ is semi definite, we note $\boldsymbol{S} = \boldsymbol{S}^{\frac{1}{2}}\boldsymbol{S}^{\frac{1}{2}}$. Solving equation 8 is equivalent to find the $K$ first eigenvectors $\boldsymbol{\alpha}_k$ associated to the $K$ largest eigenvalues $\lambda$ of the generalized eigenvalue problem:*

$$\boldsymbol{N}\boldsymbol{N}^T\boldsymbol{\alpha}_k = \lambda \, \boldsymbol{S}\boldsymbol{\alpha}_k \tag{151}$$

**Corollary 2.** *(3.2)For all integers $m, m'$ such that $m + m' \leqslant R$, at order one in $\eta$, adding $m + m'$ neurons simultaneously according to the previous method is equivalent to adding $m$ neurons then $m'$ neurons by applying successively the previous method twice.*

*Proof*

To prove 3.4, we show that the solution of 151 and the formula of 3.2 are collinear.

Solving 151 is equivalent to maximizing the following generalized Rayleigh quotient (which is solvable by the LOBPCG technique):

$$\boldsymbol{\alpha}^* = \arg\max_{\alpha} \frac{\boldsymbol{\alpha}^T \boldsymbol{N} \boldsymbol{N}^T \boldsymbol{\alpha}}{\boldsymbol{\alpha}^T \boldsymbol{S} \boldsymbol{\alpha}} \tag{152}$$

$$\boldsymbol{p}^* = \arg\max_{\boldsymbol{p} = \boldsymbol{S}^{1/2}\boldsymbol{\alpha}} \frac{\boldsymbol{p}^T \boldsymbol{S}^{-\frac{1}{2}} \boldsymbol{N} \boldsymbol{N}^T \boldsymbol{S}^{-\frac{1}{2}} \boldsymbol{p}}{\boldsymbol{p}^T \boldsymbol{p}} \tag{153}$$

$$\boldsymbol{p}^* = \arg\max_{||\boldsymbol{p}||=1} ||\boldsymbol{N}^T \boldsymbol{S}^{-\frac{1}{2}} \boldsymbol{p}|| \tag{154}$$

$$\boldsymbol{\alpha}^* = \boldsymbol{S}^{-\frac{1}{2}} \boldsymbol{p}^* \tag{155}$$

Considering the SVD of $\boldsymbol{N}^T \boldsymbol{S}^{-\frac{1}{2}} = \sum_{r=1}^{R} \lambda_r \boldsymbol{e}_r \boldsymbol{f}_r^T$, then $\boldsymbol{S}^{-\frac{1}{2}} \boldsymbol{N} \boldsymbol{N}^T \boldsymbol{S}^{-\frac{1}{2}} = \sum_{r=1}^{R} \lambda_r^2 \boldsymbol{f}_r \boldsymbol{f}_r^T$, because $j \neq i \implies \boldsymbol{e}_i^T \boldsymbol{e}_j = 0$ and $\boldsymbol{f}_i^T \boldsymbol{f}_j = 0$. Hence maximizing the first quantity is equivalent to $\boldsymbol{p}_k^* = \boldsymbol{f}_k$, then $\boldsymbol{\alpha}_k = \boldsymbol{S}^{-\frac{1}{2}} \boldsymbol{f}_k$, which matches the formula of Proposition 3.2. The same reasoning can be applied to $\boldsymbol{\omega}_k$.

We prove second corollary 3.2 by induction. Note that $\boldsymbol{v}(\theta_{\leftrightarrow}^{K,*}, \boldsymbol{x}) = o(\eta)$, then for $m = m' = 1$ :

$$\boldsymbol{a}_l(\boldsymbol{x})^{t+1} = \boldsymbol{a}_l(\boldsymbol{x})^t + \boldsymbol{v}(\theta_{\leftrightarrow}^{1,*}, \boldsymbol{x}) + o(\eta) \tag{156}$$

Remark that $\boldsymbol{v}_{\text{goal}}(\boldsymbol{x})$ is a function of $\boldsymbol{a}_l(\boldsymbol{x})$, i.e. $\boldsymbol{v}_{\text{goal}}(\boldsymbol{x}) := g(\boldsymbol{a}_l(\boldsymbol{x}))$. Then suppose that $\mathcal{L}(f(\boldsymbol{x}), \boldsymbol{y}))$ is twice differentiable in $\boldsymbol{a}_l(\boldsymbol{x})$. It follows that $g(\boldsymbol{a}_l(\boldsymbol{x}))$ is differentiable and :

$$\boldsymbol{v}_{\text{goal}}^{t+1}(\boldsymbol{x}) = g(\boldsymbol{a}_l^t(\boldsymbol{x}) + \boldsymbol{v}(\theta_{\leftrightarrow}^{1,*}, \boldsymbol{x})) \tag{157}$$

$$= g(\boldsymbol{a}_l^t(\boldsymbol{x})) + \nabla_{\boldsymbol{a}_l^t(\boldsymbol{x})} g(\boldsymbol{a}_l^t(\boldsymbol{x}))^T \boldsymbol{v}(\theta_{\leftrightarrow}^{1,*}, \boldsymbol{x}) + o(\eta^2) \tag{158}$$

$$= \boldsymbol{v}_{\text{goal}}^t(\boldsymbol{x}) + \eta \frac{\partial^2 \mathcal{L}(f_\theta(\boldsymbol{x}), \boldsymbol{y})}{\partial \boldsymbol{a}^l(\boldsymbol{x})^2} \boldsymbol{v}(\theta_{\leftrightarrow}^{1,*}, \boldsymbol{x}) + o(\eta^2) \tag{159}$$

$$= \boldsymbol{v}_{\text{goal}}^t(\boldsymbol{x}) + o(\eta) \tag{160}$$

Adding the second neuron we obtain the minimization problem:

$$\arg\min_{\boldsymbol{\alpha}_2, \boldsymbol{\omega}_2} ||\boldsymbol{V}_{\text{goal}}^t - \boldsymbol{V}(\boldsymbol{\alpha}_2, \boldsymbol{\omega}_2)|| + o(\eta) \tag{161}$$

$\square$

## C.4 About equivalence of quadratic problems

Problems 8 and 7 are generally not equivalent, but might be very close, depending on layer sizes and number of samples. The difference between the two problems is that in one case one minimizes the quadratic quantity:

$$\left\| \boldsymbol{V}^l(\theta_{\leftrightarrow}^K) + \boldsymbol{V}^l(\boldsymbol{M}) - \boldsymbol{V}_{\text{goal}}^l \right\|^2$$

w.r.t. $\boldsymbol{M}$ and $\theta_{\leftrightarrow}^K$ **jointly**, while in the other case the problem is first minimized w.r.t. $\boldsymbol{M}$ and then w.r.t. $\theta_{\leftrightarrow}^K$. The latter process, being greedy, might thus provide a solution that is not as optimal as the joint optimization.

We chose this two-step process as it intuitively relates to the spirit of improving upon a standard gradient descent: we aim at adding neurons that complement what the other ones have already done. This choice is debatable and one could solve the joint problem instead, with the same techniques.

The topic of this section is to check how close the two problems are. To study this further, note that $\boldsymbol{V}^l(\boldsymbol{M}) = \delta\boldsymbol{W}_l \boldsymbol{B}_{l-1}$ while $\boldsymbol{V}^l(\theta_{\leftrightarrow}^K) = \sum_{k=1}^{K} \boldsymbol{\omega}_k \boldsymbol{B}_{l-2}^T \boldsymbol{\alpha}_k$. The rank of $\boldsymbol{B}_{l-1}$ is $\min(n_S, n_{l-1})$ where $n_S$ is the number of samples and $n_{l-1}$ the number of neurons (post-activities) in layer $l-1$, while the rank of $\boldsymbol{B}_{l-2}$ is $\min(n_S, n_{l-2})$ where $n_{l-2}$ is the number of neurons (post-activities) in layer $l-2$. Note also that the number of degrees of freedom in the optimization variables $\delta\boldsymbol{W}_l$ and $\theta_{\leftrightarrow}^K = (\boldsymbol{\omega}_k, \boldsymbol{\alpha}_k)$ is much larger than these ranks.

**Small sample case.** If the number $n_S$ of samples is lower than the number of neurons $n_{l-1}$ and $n_{l-2}$ (which is potentially problematic, see Section E.1), then it is possible to find suitable variables $\delta \boldsymbol{W}_l$ and $\theta_{\leftrightarrow}^K$ to form any desired $\boldsymbol{V}^l(\boldsymbol{M})$ and $\boldsymbol{V}^l(\theta_{\leftrightarrow}^K)$. In particular, if $n_S \leqslant n_{l-1} \leqslant n_{l-2}$, one can choose $\boldsymbol{V}^l(\theta_{\leftrightarrow}^K)$ to be $\boldsymbol{V}_{\text{goal}}{}^l - \boldsymbol{V}^l(\boldsymbol{M})$ and thus cancel any effect due to the greedy process in two steps. The two problems are then equivalent.

**Large sample case.** On the opposite, if the number of samples is very large (compared to the number of neurons $n_{l-1}$ and $n_{l-2}$), then the lines of matrices $\boldsymbol{B}_{l-1}$ and $\boldsymbol{B}_{l-2}$ become asymptotically uncorrelated, under the assumption of their independence (which is debatable, depending on the type of layers and activation functions). Thus the optimization directions available to $\boldsymbol{V}^l(\boldsymbol{M})$ and $\boldsymbol{V}^l(\theta_{\leftrightarrow}^K)$ become orthogonal, and proceeding greedily does not affect the result, the two problems are asymptotically equivalent.

In the general case, matrices $\boldsymbol{B}_{l-1}$ and $\boldsymbol{B}_{l-2}$ are not independent, though not fully correlated, and the number of samples (in the minibatch) is typically larger than the number of neurons; the problems are then different.

Note that technically the ranks could be lower, in the improbable case where some neurons are perfectly redundant, or, e.g., if some samples yield exactly the same activities.

## D Section *About greedy growth sufficiency and TINY convergence* with more details and proofs

One might wonder whether a greedy approach on layer growth might get stuck in a non-optimal state. By *greedy* we mean that every neuron added has to decrease the loss. We derive the following series of propositions in this regard. Since in this work, we add neurons layer per layer independently, we study here the case of a single hidden layer network, to spot potential layer growth issues. For the sake of simplicity, we consider the task of least square regression towards an explicit continuous target $f^*$, defined on a compact set. That is, we aim at minimizing the loss:

$$\inf \sum_{\boldsymbol{x} \in \mathcal{D}} \|f(\boldsymbol{x}) - f^*(\boldsymbol{x})\|^2 \tag{162}$$

where $f(\boldsymbol{x})$ is the output of the neural network and $\mathcal{D}$ is the training set.

We start with an optional introductory section D.1 about greedy growth possibilities, then prepare lemmas in Sections D.2 and D.3 that will be used in Section D.4 to show that one can keep on adding neurons to a network (without modifying already existing weights) to make it converge exponentially fast towards the optimal function. Then in Section D.6 we present a growth method that explicitly overfits each dataset sample one by one, thus requiring only $n$ neurons, thanks to existing weights modification. Finally, more importantly, in Section D.7, we show that actually any reasonable growth method that follows a certain optimization protocol (this includes TINY completed by random neuron additions if necessary) will reach 0 training error in at most $n$ neuron additions.

### D.1 Possibility of greedy growth

**Proposition D.1** (Greedy completion of an existing network). *If $f$ is not $f^*$ yet, there exists a set of neurons to add to the hidden layer such that the new function $f'$ will have a lower loss than $f$.*

One can even choose the added neurons such that the loss is arbitrarily well minimized.

*Proof.* The classic universal approximation theorem about neural networks with one hidden layer Pinkus (1999) states that for any continuous function $g^*$ defined on a compact set $\boldsymbol{\omega}$, for any desired precision $\gamma$, and for any activation function $\sigma$ provided it is not a polynomial, then there exists a neural network $g$ with one hidden layer (possibly quite large when $\gamma$ is small) and with this activation function $\sigma$, such that

$$\forall x, \|g(x) - g^*(x)\| \leqslant \gamma \tag{163}$$

We apply this theorem to the case where $g^* = f^* - f$, which is continuous as $f^*$ is continuous, and $f$ is a shallow neural network and as such is a composition of linear functions and of the function $\sigma$, that we will suppose to be continuous for the sake of simplicity. We will suppose that $f$ is real-valued for the sake of simplicity as well, but the result is trivially extendable to vector-valued functions (just concatenate the networks obtained for each output independently). We choose $\gamma = \frac{1}{10}\|f^* - f\|_{L^2}$, where $\langle a|b\rangle_{L^2} = \frac{1}{|\boldsymbol{\omega}|}\int_{\boldsymbol{x}\in\boldsymbol{\omega}} a(\boldsymbol{x})\, b(\boldsymbol{x})\, d\boldsymbol{x}$. This way we obtain a one-hidden-layer neural network $g$ with activation function $\sigma$, and we write $a(\boldsymbol{x}) = g(\boldsymbol{x}) - g^*(\boldsymbol{x})$ the error term, it follows that:

$$\forall \boldsymbol{x} \in \boldsymbol{\omega}, \;\; -\gamma \leqslant g(\boldsymbol{x}) - g^*(\boldsymbol{x}) \leqslant \gamma \tag{164}$$

$$\forall \boldsymbol{x} \in \boldsymbol{\omega}, \;\; g(\boldsymbol{x}) = f^*(\boldsymbol{x}) - f(\boldsymbol{x}) + a(\boldsymbol{x}) \tag{165}$$

with $\forall \boldsymbol{x} \in \boldsymbol{\omega}, |a(\boldsymbol{x})| \leqslant \gamma$.

Then:

$$\forall \boldsymbol{x} \in \boldsymbol{\omega}, \;\; f^*(\boldsymbol{x}) - (f(\boldsymbol{x}) + g(\boldsymbol{x})) = -a(\boldsymbol{x}) \tag{166}$$

$$\forall \boldsymbol{x} \in \boldsymbol{\omega}, \;\; (f^*(\boldsymbol{x}) - h(\boldsymbol{x}))^2 = a^2(\boldsymbol{x}) \tag{167}$$

with $h$ being the function corresponding to a neural network consisting of concatenating the hidden layer neurons of $f$ and $g$, and consequently summing their outputs.

$$\|f^* - h\|_{L^2}^2 = \|a\|_{L^2}^2 \tag{168}$$

$$\|f^* - h\|_{L^2}^2 \leqslant \gamma^2 = \frac{1}{100}\|f^* - f\|_{L^2}^2 \tag{169}$$

and consequently the loss is reduced indeed (by a factor of 100 in this construction).

The same holds in expectation or sum over a training set, by choosing $\gamma = \frac{1}{10}\sqrt{\frac{1}{|\mathcal{D}|}\sum_{\boldsymbol{x}\in\mathcal{D}}\|f(\boldsymbol{x}) - f^*(\boldsymbol{x})\|^2}$, as Equation (167) then yields:

$$\sum_{\boldsymbol{x}\in\mathcal{D}} (f^*(\boldsymbol{x}) - h(\boldsymbol{x}))^2 = \sum_{\boldsymbol{x}\in\mathcal{D}} a^2(\boldsymbol{x}) \leqslant \frac{1}{100}\sum_{\boldsymbol{x}\in\mathcal{D}} (f^*(\boldsymbol{x}) - f(\boldsymbol{x}))^2 \tag{170}$$

which proves the proposition as stated.

For more general losses, one can consider order-1 (linear) development of the loss and ask for a network $g$ that is close to (the opposite of) the gradient of the loss.

$\square$

*Proof of the additional remark.* The proof in Pinkus (1999) relies on the existence of real values $c_n$ such that the $n$-th order derivatives $\sigma^{(n)}(c_n)$ are not 0. Then, by considering appropriate values arbitrarily close to $c_n$, one can approximate the $n$-th derivative of $\sigma$ at $c_n$ and consequently the polynomial $c^n$ of order $n$. This standard proof then concludes by density of polynomials in continuous functions.

Provided the activation function $\sigma$ is not a polynomial, these values $c_n$ can actually be chosen arbitrarily, in particular arbitrarily close to 0. This corresponds to choosing neuron input weights arbitrarily close to 0. $\square$

**Proposition D.2** (Greedy completion by one single neuron)**.** *If $f$ is not $f^*$ yet, there exists a neuron to add to the hidden layer such that the new function $f'$ will have a lower loss than $f$.*

*Proof.* From the previous proposition, there exists a finite set of neurons to add such that the loss will be decreased. In this particular setting of $L^2$ regression, or for more general losses if considering small function moves, this means that the function represented by this set of neurons has a strictly negative component over the gradient $g$ of the loss ($g = -2(f^* - f)$ in the case of the $L^2$ regression). That is, denoting by $a_i\sigma(\boldsymbol{W}_i\cdot\boldsymbol{x})$ these $N$ neurons:

$$\left\langle \sum_{i=1}^{N} a_i\sigma(\boldsymbol{w}_i\cdot\boldsymbol{x}) \,\Big|\, g \right\rangle_{L^2} = K < 0 \tag{171}$$

i.e.

$$\sum_{i=1}^{N} \langle a_i \sigma(\boldsymbol{w}_i \cdot \boldsymbol{x}) |\, g \rangle_{L^2} = K < 0 \tag{172}$$

We have:

$$0 > \frac{1}{N} K = \frac{1}{N} \sum_{i=1}^{N} \langle a_i \sigma(\boldsymbol{w}_i \cdot \boldsymbol{x}) |\, g \rangle_{L^2} \geqslant \min_{i=1}^{N} \langle a_i \sigma(\boldsymbol{w}_i \cdot \boldsymbol{x}) |\, g \rangle_{L^2} \tag{173}$$

Then necessarily at least one of the $N$ neurons satisfies

$$\langle a_i \sigma(\boldsymbol{w}_i \cdot \boldsymbol{x}) |\, g \rangle_{L^2} \leqslant \frac{1}{N} K < 0 \tag{174}$$

and thus decreases the loss when added to the hidden layer of the neural network representing $f$. Moreover this decrease is at least $\frac{1}{N}$ of the loss decrease resulting from the addition of all neurons.

$\square$

As a consequence, there exists no situation where one would need to add many neurons simultaneously to decrease the loss: it is always feasible with a single neuron. Note that finding the optimal neuron to add is actually NP-hard (Bach, 2017), so we will not necessarily search for the optimal one. A constructive lower bound on how much the loss can be improved will be given later in this section.

**Proposition D.3** (Greedy completion by one infinitesimal neuron)**.** *The neuron in the previous proposition can be chosen to have arbitrarily small input weights.*

*Proof.* This is straightforward, as, following a previous remark, the neurons found to collectively decrease the loss can be supposed to all have arbitrarily small input weights. $\square$

This detail is important in that our approach is based on the tangent space of the function $f$ and thus manipulates infinitesimal quantities. Our optimization problem indeed relies on the linearization of the activation function by requiring the added neuron to have infinitely small input weights, to make the problem easier to solve. This proposition confirms that such neuron exists indeed.

**Correlations and higher orders.** Note that, as a matter of fact, our approach exploits linear correlations between inputs of a layer and desired output variations. It might happen that the loss is not minimized yet but there is no such correlation to exploit anymore. In that case the optimization problem (8) will not find neurons to add. Yet following Prop. D.3 there does exist a neuron with arbitrarily small input weights that can reduce the loss. This paradox can be explained by pushing further the Taylor expansion of that neuron output in terms of weight amplitude (single factor $\varepsilon$ on all of its input weights), for instance $\sigma(\varepsilon \boldsymbol{\alpha} \cdot \boldsymbol{x}) \simeq \sigma(0) + \sigma'(0)\varepsilon \boldsymbol{\alpha} \cdot \boldsymbol{x} + \frac{1}{2}\sigma''(0)\varepsilon^2(\boldsymbol{\alpha} \cdot \boldsymbol{x})^2 + O(\varepsilon^3)$. Though the linear term $\boldsymbol{\alpha} \cdot \boldsymbol{x}$ might be uncorrelated over the dataset with desired output variation $v(\boldsymbol{x})$, i.e. $\mathbb{E}_{\boldsymbol{x} \sim \mathcal{D}}[\boldsymbol{x}\, v(\boldsymbol{x})] = 0$, the quadratic term $(\boldsymbol{\alpha} \cdot \boldsymbol{x})^2$, or higher-order ones otherwise, might be correlated with $v(\boldsymbol{x})$. Finding neurons with such higher-order correlations can be done by increasing accordingly the power of $(\boldsymbol{\alpha} \cdot \boldsymbol{x})$ in the optimization problem (7). Note that one could consider other function bases than the polynomials from Taylor expansion, such as Hermite or Legendre polynomials, for their orthogonality properties. In all cases, one does not need to solve such problems exactly but just to find an approximate solution, i.e. a neuron improving the loss.

**Adding random neurons.** Another possibility to suggest additional neurons, when expressivity bottlenecks are detected but no correlation (up to order $p$) can be exploited anymore, is to add random neurons. The first $p$ order Taylor expansions will show 0 correlation with desired output variation, hence no loss improvement nor worsening, but the correlation of the $p + 1$-th order will be non-0, with probability 1, in the spirit of random projections. Furthermore, in the spirit of common neural network training practice, one could consider brute force combinatorics by adding many random neurons and hoping some will be close enough to the desired direction (Frankle & Carbin, 2018). The difference with the usual training is that we would perform such computationally costly searches only when and where relevant, exploiting all simple information first (linear correlations in each layer).

### D.2 Loss decreases with a line search on a quadratic energy

Let $\mathcal{L}$ be a quadratic loss over $\mathbb{R}^d$ and $g$ be a vector in $\mathbb{R}^d$. The loss $\mathcal{L}$ can be written as:

$$\mathcal{L}(g) = g^T Q g + v^T g + K \tag{175}$$

where $Q$ is a matrix that we will suppose to be symmetric positive definite. This is to ensure that all eigenvalues of $Q$ are positive, hence modeling a local minimum without a saddle point. $v$ is a vector in $\mathbb{R}^d$ and $K$ is a real constant.

For instance, the mean square loss $\mathbb{E}_{x \in \mathcal{D}} \left[ \|f(x) - f^*(x)\|_S^2 \right]$, where $\mathcal{D}$ is a finite dataset of $N$ samples, $f^*$ a target function, and $S$ is a symmetric positive definite matrix used as a metric, fits these hypotheses, considering $g = (f(x_1), f(x_2), ...)$ as a vector. Indeed this loss rewrites as

$$\sum_{i=1}^N f(x_i)^T S f(x_i) - 2 \sum_i f^{*T}(x_i) S f(x_i) + K \ = \ g^T Q\, g \ + \ v^T g \ + \ K \tag{176}$$

by flattening and concatenating the vectors $f(x_i)$ and considering $Q = S \otimes S \otimes S \otimes ...$ the tensor product of $N$ times the same matrix $S$, i.e. a diagonal-block matrix with $N$ identical blocks $S$. Note that for the standard regression with the $L^2$ metric, this matrix $Q$ is just the Identity.

Starting from point $g$, and given a direction $h \in \mathbb{R}^d$, the question is to perform a line search in that direction, i.e. to optimize the factor $\lambda \in \mathbb{R}$ in order to minimize $\mathcal{L}(g + \lambda h)$.

Developing that expression, we get:

$$\mathcal{L}(g + \lambda h) \ = \ (g + \lambda h)^T Q\, (g + \lambda h) + v^T(g + \lambda h) + K \ = \ \lambda^2 h^T Q h + \lambda(2h^T Q g + v^T h) + \mathcal{L}(g) \tag{177}$$

which is a second-order polynomial in $\lambda$ with a positive quadratic coefficient. Note that the linear coefficient is $h^T \nabla_g \mathcal{L}(g)$, where $\nabla_g \mathcal{L}(g) = 2Qg + v$ is the gradient of $\mathcal{L}$ at point $g$. The unique minimum of the polynomial in $\lambda$ is then:

$$\lambda^* = -\frac{1}{2} \frac{h^T \nabla_g \mathcal{L}(g)}{h^T Q h} \tag{178}$$

which leads to

$$\min_\lambda \mathcal{L}(g + \lambda h) = \lambda^{*2} h^T Q h + \lambda^* h^T \nabla_g \mathcal{L}(g) + \mathcal{L}(g) \tag{179}$$

$$= \mathcal{L}(g) - \frac{1}{4} \frac{\left( h^T \nabla_g \mathcal{L}(g) \right)^2}{h^T Q h} \tag{180}$$

$$= \mathcal{L}(g) - \frac{1}{4} \left\langle \frac{h}{\|h\|_Q} \,\middle|\, \nabla_g^Q \mathcal{L}(g) \right\rangle_Q^2 . \tag{181}$$

Thus the loss gain obtained by a line search in a direction $h$ is quadratic in the angle between that direction and the gradient of the loss, in the sense of the $Q$ norm (and it is also quadratic in the norm of the gradient). Note that inner products with the gradient do not depend on the metric, in the sense that $\langle h \,|\, \nabla_g \mathcal{L}(g) \rangle_{L^2} \ = \ \langle h \,|\, \nabla_g^S \mathcal{L}(g) \rangle_S \quad \forall h$ for any metric $S$, i.e. any symmetric definite positive matrix $S$, associated to the norm $\|h\|_S^2 = h^T S h$ and to the gradient $\nabla_g^S \mathcal{L}(g) = S^{-1} \nabla_g^{L^2} \mathcal{L}(g)$.

In the case of a standard $L^2$ regression this boils down to:

$$\min_\lambda \|g + \lambda h\|_{L^2}^2 = \|g\|^2 - \left\langle \frac{h}{\|h\|} \,\middle|\, g \right\rangle_{L^2}^2 \tag{182}$$

i.e. considering $\mathcal{L}(f) := \mathbb{E}_{x \in \mathcal{D}} \left[ \|f(x) - f^*(x)\|^2 \right]$ :

$$\min_\lambda \mathcal{L}(f + \lambda h) \ = \ \mathcal{L}(f) - \left\langle \frac{h}{\|h\|} \,\middle|\, f^* - f \right\rangle_{L^2}^2 \ = \ \mathcal{L}(f) - \frac{\mathbb{E}_{x \in \mathcal{D}} \left[ (f^* - f)\, h \right]^2}{\mathbb{E}_{x \in \mathcal{D}} \left[ \|h\|^2 \right]} . \tag{183}$$

A result that is useful in the next sections.

### D.3 Expected loss gain with a line search in a random direction

Using Appendix D.2 above, the loss gain when performing a line search on a quadratic loss is quadratic in the angle $\alpha = \left\langle \frac{\boldsymbol{V}(X)}{\|\boldsymbol{V}(X)\|} \;\middle|\; \frac{\boldsymbol{V}_{\text{goal}}(X)}{\|\boldsymbol{V}_{\text{goal}}(X)\|} \right\rangle_{L^2}$ between the random search direction $\boldsymbol{V}(X)$ and the gradient $\boldsymbol{V}_{\text{goal}}(X)$.

This angle has average 0 and is of standard deviation $\frac{1}{nd}$, as described in Section 5.2. The loss gain is thus of the order of magnitude of $\frac{1}{d}$ in the best case (single-sample minibatch).

### D.4 Exponential convergence to 0 training error

Considering a regression to a target $f^*$ with the quadratic loss, the function $f$ represented by the current neural network (fully-connected, one hidden layer, with ReLU activation function) can be improved to reach 0 loss by an addition of $n$ neurons $(h_i)_{1 \leqslant i \leqslant n}$, with $n$ is the dataset size, using Zhang et al. (2017). Unfortunately there is no guarantee that if one adds each of these neurons one by one, the loss decreases each time. We will prove that one of these neurons does decrease the loss, and we will quantify by how much, relying on the explicit construction in Zhang et al. (2017). This decrease will actually be a constant factor of the loss, thus leading to exponential convergence towards the target $f^*$ on the training set.

As in the proof of Proposition D.2 in Appendix D, at least one of the added neurons satisfies that its inner product with the gradient direction is at least $1/n$. While one could consequently hope for a loss gain in $O(\frac{1}{n})$, one has to see that this decrease would be the one of a gradient step, which is multiplied by a step size $\eta$, and asks for multiple steps to be done. Instead in our approach we actually perform a line search over the direction of the new neuron. In both cases (line search or multiple small gradient steps), one has to take into account at least order-2 changes of the loss to compute the line search or estimate suitable $\eta$ and/or its associated number of steps. Luckily in our case of least square regression, the loss is exactly equal to its second order Taylor development, and all following computations are exact.

We consider the mean square regression loss $\mathcal{L}(f) = \mathbb{E}_{x \in \mathcal{D}} \left[ \|f(x) - f^*(x)\|_S^2 \right]$, where $\mathcal{D}$ is a finite training dataset of $N$ samples. Its functional gradient $\nabla \mathcal{L}(f)$ at point $f$ is $2(f - f^*)$, which is proportional to the optimal change to add to $f$, that is, $f^* - f$. The $n$ neurons $(h_i)_{1 \leqslant i \leqslant n}$ to be added to $f$ following Zhang et al. (2017) satisfy $\sum_i h_i = f^* - f = -\frac{1}{2}\nabla \mathcal{L}(f)$. Thus

$$\left\langle \sum_i h_i \;\middle|\; f^* - f \right\rangle_{L^2} \;=\; \|f^* - f\|_{L^2}^2 \;=\; \mathcal{L}(f). \tag{184}$$

Then like in the proof of D.2 we use that the maximum is greater or equal to the mean to get that there exists a neuron $h_i$ that satisfies:

$$\langle h_i | \; f^* - f \rangle_{L^2} \;\geqslant\; \mathcal{L}(f)/n. \tag{185}$$

By applying Appendix D.2 one obtains that the new loss after line search into the direction of $h_i$ yields:

$$\min_\lambda \mathcal{L}(f + \lambda h_i) \;=\; \mathcal{L}(f) - \frac{\langle h_i \mid f^* - f \rangle_{L^2}^2}{\|h_i\|^2} \;\leqslant\; \mathcal{L}(f) \times \left( 1 - \frac{\mathcal{L}(f)}{n^2 \|h_i\|^2} \right). \tag{186}$$

From the particular construction in Zhang et al. (2017) it is possible to bound the square norm of the neuron $\|h_i\|^2$ by $n\, d' \left( \frac{d_M}{d_m} \right)^2 \mathcal{L}(f)$, where $d_M$ is related to the maximum distance between 2 points in the dataset, $d_m$ is another geometric quantity related to the minimum distance, and $d'$ is the network output dimension. To ease the reading of this proof, we defer the construction of this bound to the next section, Appendix D.5.

Then the loss at each neuron addition decreases by a factor which is at least $\gamma = 1 - \frac{1}{n^3 d'} \left( \frac{d_m}{d_M} \right)^2 < 1$. This factor is a constant, as it is a bound that depends only on the geometry of the dataset (not on $f$).

Thus it is possible to decrease the loss exponentially fast with the number $t$ of added neurons, i.e. $\mathcal{L}(f_t) \leqslant \gamma^t \mathcal{L}(f)$, towards 0 training loss, and this in a greedy way, that is, by adding neuron one by one, with the property that each neuron addition decreases the loss.

Note that, in the proof of Zhang et al. (2017), the added neurons could be chosen to have arbitrarily small input weights. This corresponds to choosing $a$ with small norm instead of unit norm in Equation 187.

The number of neuron additions expected to reach good performance according to this bound is in the order of magnitude of $n^3$, which is to be compared to $n$ (number of neurons needed to overfit the dataset, without the constraint that each addition decreases the loss). This bound might be improved using other constructions than Zhang et al. (2017), though with this proof the bound cannot be better than $n^2$ (supposing $\|h_i\|$ can be made not to depend on $n$).

Note also that with ReLU activation functions, all points that are on the convex hull of the dataset (which is necessarily the case of all points if the input dimension is higher that the number of points) can easily in turn be perfectly predicted (0 loss) by just one neuron addition each (without changing the outputs for the other points), by choosing an hyperplane that separates the current convex hull point from the rest of the dataset, and setting a ReLU neuron in that direction.

### D.5 Bound on the norm of the neurons

Here we prove that the neurons obtained by Zhang et al. (2017) can be chosen so as to bound the square norm of any neuron $\|h_i\|^2$ by $n\, d' \left(\frac{d_M}{d_m}\right)^2 \mathcal{L}(f)$, where $d_M$ is related to the maximum distance between 2 points in the dataset, and $d_m$ is another geometric quantity related to the minimum distance. For the sake of simplicity, we first consider the case where the output dimension is $d' = 1$.

In Zhang et al. (2017), the $n$ neurons are obtained by solving $y = Aw$, where $y = (y_1, y_2...)$ is the target function (here $(f^* - f)$ at each $x_j$), $A$ is the matrix given by $A_{jk} = \text{ReLU}(a \cdot x_j - b_k)$, representing neuron activations, and $a$ is any vector that separates the dataset points, i.e. $a \cdot x_j \neq a \cdot x_{j'} \; \forall j \neq j'$, that is, $a$ could be almost any vector in $\mathbb{R}^d$ (in the sense of random projections, that is, the set of vectors that do not satisfy this is of measure 0).

Here we will pick a particular unit direction $a$, one that maximizes the distance between any two samples after projection:

$$a \in \underset{\|a\|=1}{\arg\max} \; \underset{j,j'}{\min} |a \cdot (x_j - x_{j'})| \tag{187}$$

and let us denote $d'_m$ the associated value: $d'_m = \min_{j,j'} |a \cdot (x_j - x_{j'})|$ for that $a$. Note that $d'_m \leqslant \min_{j,j'} \|x_j - x_{j'}\|$ and that it depends only on the training set. The quantity $d'_m$ is likely to be also lower-bounded (over all possible datasets) by $\min_{j,j'} \|x_j - x_{j'}\|$ times a factor depending on the embedding dimension $d$ and the number of points $n$.

Now, let us sort the samples according to increasing $a \cdot x_j$, that is, let us re-index the samples such that $(a \cdot x_j)$ now grows with $j$. By definition of $a$, the difference between any two consecutive $a \cdot x_j$ is at least $d'_m$.

We now choose biases $b_j = a \cdot x_j - d'_m + \varepsilon$ for some very small $\varepsilon$. The neurons are then defined as $h_k(x) = w_k \text{ReLU}(a \cdot x - b_k)$. The induced activation matrix $A_{jk} = \text{ReLU}(a \cdot x_j - b_k)$ then satisfies $\forall j < k; A_{jk} = 0$ and $\forall j \geqslant k; A_{jk} \geqslant d'_m - \varepsilon$. The matrix $A$ is lower triangular with diagonal elements above $d_m := d'_m - \varepsilon$, hence invertible. Recall that $y = Aw$.

Consequently, $w = A^{-1}y$, and hence $\|w\|^2 \leqslant \||A^{-1}\||^2 \|y\|^2$, that is,

$$\|w\|^2 \leqslant \frac{1}{d_m^2} \mathcal{L}(f) \tag{188}$$

as the target $y$ is the vector $f^* - f$ in our case. Consequently, for any neuron $h_i$, one has:

$$w_i^2 \leqslant \frac{1}{d_m^2} \mathcal{L}(f). \tag{189}$$

As the norm of the neuron is $\|h_i\|^2 = w_i^2 \sum_j A_{ji}^2$, one still has to bound the activities $A_{ji} = \text{ReLU}(a \cdot x_j - b_i)$. As $a$ was chosen a unit direction, the values $a \cdot x_j$ span a domain smaller than the diameter of the dataset $\mathcal{D}$: $|a \cdot (x_j - x_{j'})| \leqslant \|x_j - x_{j'}\| \leqslant \text{diam}(\mathcal{D}) \; \forall j, j'$. Hence all values $\forall i, j, |A_{ij}| = |a \cdot x_i - b_j| = |a \cdot x_i - a \cdot x_j + d_m| < d_M := \text{diam}(\mathcal{D}) + d_m$. Note that $d_M$ depends only the dataset geometry, as for $d_m$.

We now have:

$$\|h_i\|^2 = w_i^2 \sum_j A_{ji}^2 \leqslant n \frac{d_M^2}{d_m^2} \mathcal{L}(f) \tag{190}$$

which ends the proof.

For higher output dimensions $d'$, one vector $w$ of output weights is estimated per dimension, independently, leading to the same bound for each dimension. The square norms of neurons are summed over all dimensions and thus multiplied by at most $d'$.

### D.6  Reaching 0 training error in $n$ neuron additions by overfitting each dataset sample in turn

If one allows updating already existing output weights at the same time as one adds new neurons, then it is possible to reach 0 training error in only $n$ steps (where $n$ is the size of the dataset) while decreasing the loss at each addition.

This scenario is closer to the one we consider with TINY, as we compute the optimal update of existing weights inside the layer, as a byproduct of new neuron estimation, and apply them.

However the existence proof here follows a very different way to create new neurons, tailored to obtain a constructive proof, and inspired by the previous section. See Appendix D.7 for another, more generic proof, applicable to a wide range of growth methods.

Here we consider the same approach as in Appendix D.5 above, but introducing neurons one by one instead of $n$ neurons at once. After computing $a$ and the biases $b_j$, thus forming the activity matrix $A$, we add only the last neuron $h_n$. The activity of this neuron is 0 for all input samples $x_j$ except for the last one, for which it is $A_{nn} > 0$. Thus, the neuron $h_n$ separates the sample $x_n$ from the rest of the dataset, and it is easy to find $w_n$ so that the loss gets to 0 on that training sample, without changing the outputs for other samples.

Similarly, one can then add neuron $h_{n-1}$, which is active only for samples $x_{n-1}$ and $x_n$. However designing $w_{n-1}$ so that the loss becomes 0 at point $x_{n-1}$ disturbs the output for point $x_n$ (and for that point only). Luckily if one allows updating $w_n$ then there exists a (unique) solution $(w_{n-1}, w_n)$ to achieve 0 loss at both points. This is done exactly as previously, by solving $y = Aw$, but considering only the last 2 lines and rows of $A$, leading to a smaller $2 \times 2$ system which is also lower-triangular with positive diagonal.

Proceeding iteratively this way adds neuron one by one in a way that sends each time one more sample to 0 loss. Thus adding $n$ neurons is sufficient to achieve 0 loss on the full training set, and this in a way that each time decreases the loss.

Note that updating existing output weights $w_i$ while adding a new neuron, to decrease optimally the loss, is actually what TINY does. However, the construction in this Appendix completely overfits each sample in turn, by design, without being able to generalize to new test points. On the opposite, TINY exploits correlations over the whole dataset to extract the main tendencies.

### D.7  TINY reaches 0 training error in $n$ neuron additions

We will now show that the TINY approach, as well as any other suitable greedy growth method, implemented within the right optimization procedure, reaches 0 training error in at most $n$ steps (where $n$ is the size of the dataset), almost surely.

Before stating it formally, we need to introduce the optimization protocol, growth completion and a probability measure over activation functions.

**Optimization protocol.** For this we consider the following optimization protocol conditions, that has to be applied at least during the last, $n$-th addition step:

- a **full batch** approach,

- when adding new neurons, also compute and add the **optimal moves of already existing parameters** (i.e. of output weights $w$).

The first point is to ensure that all dataset samples will be taken into account in the loss during the $n$-th update. Otherwise, for instance if using minibatches instead, the optimization of output weights $w$ will not be able to overfit the training loss.

The second point is to make sure that, after update, the output weights $w$ will be optimal for the training loss. Note that in the mean square regression case, this is easy to do, as the loss is quadratic in $w$: the optimal move (leading to the global optimum $f^*$) can be obtained by line search over the natural gradient (which is obtained for free as a by-product of TINY's projection of $\boldsymbol{V}_{\text{goal}}$, and is proportional to $f^* - f$). This is precisely what we do in practice with TINY when training networks (except when comparing with other methods and using their own protocol).

**Growth completion.** For this proof to make sense, we will need the growth method to actually be able to perform $n$ neuron additions, if it has not reached 0 training loss before. A counter-example would be a growth method that gets stuck at a place where the training loss is not 0 while being unable to propose new neuron to add. In the case of TINY, this can happen when no correlation between inputs $x_i$ and desired output variations $f^*(x_i) - f(x_i)$ can be found anymore. To prevent this, one can choose any auxiliary method to add neurons in such cases, for instance random directions, solutions of higher-order expressivity bottleneck formulations using further developments of the activation function, or locally optimal neurons found by gradient descent. Some auxiliary methods are guaranteed to further decrease the loss by a neuron addition (cf. Appendices D.2, D.3, D.4), while any other one is guaranteed not to increase the loss if combined with a line search along that neuron direction.

We will name *completed-TINY* the completion of TINY by any such auxiliary method.

**Activation function.** For technical reasons, the result will stand *almost surely* only, depending on the invertibility of a certain matrix, namely, the activation matrix $A$, defined as $A_{ij} = \sigma(\boldsymbol{v}_j \cdot \boldsymbol{x}_i + b_j)$, indexed by samples $i$ and neurons $j$.

Generally speaking, kernels induced by neurons $k_j : \ \boldsymbol{x} \mapsto \sigma(\boldsymbol{v}_j \cdot \boldsymbol{x} + b_j)$ form free families, in the sense that they are linearly independent (to the notable exception of the linear kernel). This linear independency means that a linear combination of kernels cannot be equal, as a function, to another kernel with different parameters. Equality is to be understood as *for all possible points $\boldsymbol{x}$ ever*. However here we will evaluate the functions only at a finite number $n$ of points (the dataset samples), therefore linear independence will be considered among the rows of the activation matrix $A$. This notion of linear dependence is much weaker: kernels might form a free family as functions but be linearly dependent once restricted to the dataset samples, by mere chance. While this is not likely (over dataset samples), this is not impossible in general (though of measure 0), and it is difficult to express an explicit, simple condition on the activation function to be sure that the activation matrix $A$ is *always* invertible (up to slight changes of parameters). Thus instead we will express results *almost surely* over activation functions and neuron parameters.

For most activation functions in the space of smooth functions, the activation matrix $A$ will be invertible almost surely over all possible datasets. In the unlucky case where the matrix is not invertible, an infinitesimal move of the neurons' parameters will be sufficient to make it invertible. For some activation functions, however, such as linear or piecewise-linear ones (e.g., ReLU), the matrix might remain non-invertible over a wide range of parameter variations (unless further assumptions are made on the neurons added by the growth process). Yet, in such cases, slight perturbations of the activation function (i.e., choosing another, smooth, activation function, arbitrarily close to the original one) will yield invertibility.

To properly define "*almost surely*" regarding activation functions, let us restrict the activation function $\sigma$ to belong to the space $\mathfrak{P}$ of polynomials of order at least $n^2$, that is:

$$\sigma(x) \;=\; \sum_{k=0}^{K} \gamma_k \, x^k \tag{191}$$

with $n^2 \leqslant K < +\infty$, and non-0 highest-order amplitude $\gamma_K \neq 0$. This set $\mathfrak{P}$ is dense in the set of all continuous functions over the set $\Omega = [-r_M, r_M]^d$ which is a hypercube of sufficient radius $r_M$ to cover all samples from the given dataset. One can define probability distributions over $\mathfrak{P}$, for instance consider the density $p(\sigma) = \frac{\alpha}{K^2} \prod_{k=0}^{K} \frac{e^{-\gamma_k^2}}{\sqrt{2\pi}}$ with a factor $\alpha = \left( \frac{\pi^2}{6} - \sum_{k<n^2} \frac{1}{k^2} \right)^{-1}$ to normalize the distribution, and where $K$ is the order of the polynomial and thus depends on $\sigma$. This density is continuous in the space of parameters $\gamma_k$ (though not continuous in the usual functional metric spaces). Note that the decomposition of any $\sigma \in \mathfrak{P}$ as a finite-order polynomial is unique, as monomials of different orders are linearly independent.

We can now state the following lemma (that we will prove later):

**Lemma D.1** (Invertibility of the activation matrix). *Let $\mathcal{D} = \{x_i, 1 \leqslant i \leqslant n\}$ be a dataset of $n$ distinct points, and let $\sigma : \mathbb{R} \to \mathbb{R}$ be a function in $\mathfrak{P}$, that is, a polynomial of order at least $n^2$. Then with probability 1 over function and neuron parameters $(\gamma_k)$, $(\boldsymbol{v}_j)$ and $(b_j)$, the activity matrix $A$ defined by $A_{ij} = \sigma(\boldsymbol{v}_j \cdot \boldsymbol{x}_i + b_j)$ is full rank.*

and the following proposition:

**Proposition D.4** (Reaching 0 training error in at most $n$ neuron additions). *Under the assumptions above (polynomial activation function of order $\geqslant n^2$, full-batch optimization and computation of the optimal moves of already existing parameters), completed-TINY reaches 0 training error in at most $n$ neuron additions almost surely.*

*Proof.* If the growth method reaches 0 training error before $n$ neuron additions, the proof is done. Otherwise, let us consider the $n$-th neuron addition. We will show in Lemma D.1 that the activity matrix $A$, defined by $A_{ij} = \sigma(\boldsymbol{v}_j \cdot \boldsymbol{x}_i + b_j)$, indexed by samples $i$ and neurons $j$, is invertible. Then there exists a unique $\boldsymbol{w} \in \mathbb{R}^n$ such that $A\boldsymbol{w} = f^*$, i.e. $\sum_j w_j \sigma(\boldsymbol{v}_j \cdot \boldsymbol{x}_i + b_j) = f^*(\boldsymbol{x}_i)$ for each point $\boldsymbol{x}_i$ of the dataset. This vector of output parameters $\boldsymbol{w}$ realizes the global minimum of the loss over already existing weights: $\inf_{\boldsymbol{w}} \mathcal{L}(f_{\boldsymbol{v},\boldsymbol{w}}) = \inf_{\boldsymbol{w}} \|A\boldsymbol{w} - f^*\|^2$. They are also the ones found by a natural gradient step over the loss (up to a factor 2, that can easily be found by line search as the loss is convex). Then after that update the training loss is exactly 0. $\qquad\square$

Note: piecewise-linear activation functions such as ReLU are not covered by this proposition. However the result might still hold with further assumptions over the growth process. For instance, with the method in Zhang et al. (2017), the ReLU neurons are chosen in such a way that the matrix $A$ is full rank by construction.

*Proof of Lemma D.1.* Let us first show that if, unluckily, for a given activation function $\sigma$ and given parameters $(\boldsymbol{v}_j, b_j)$, the matrix $A$ is not full rank, then upon infinitesimal variation of the parameters, the matrix $A$ becomes full rank.

Indeed, if all pre-activities $a_{i,j} := \boldsymbol{v}_j \cdot \boldsymbol{x}_i + b_j$ are not distinct for all $i, j$, then an infinitesimal variation of the vectors $\boldsymbol{v}_j$ can make them distinct. For this, one can see that the set of directions $\boldsymbol{v}_j$ on which any two dataset points $\boldsymbol{x}_i$ and $\boldsymbol{x}_{i'}$ have the same projection is finite (since it has to be the direction of $\boldsymbol{x}_i - \boldsymbol{x}_{i'}$, for a given pair of dataset samples $(i, i')$) and thus of measure 0. As a consequence with probability 1 over neuron parameters $\boldsymbol{v}_j$ and $b_j$, all pre-activities are distinct.

Now, if the matrix $A$ is not invertible, as invertible matrices are dense in the space of matrices, one can easily find an infinitesimal change $\delta A$ to apply to $A$ to make it invertible. This corresponds to changing the activation function $\sigma$ accordingly at each of the $n^2$ distinct pre-activity values. Since $\sigma$ has more than $n^2$ parameters, this is doable. For instance, one can select the $n^2$ first parameters and search for a suitable variation $\boldsymbol{g} := (\delta\gamma_k)_{0 \leqslant k < n^2}$ of them by solving the linear system $S\,\boldsymbol{g} = \delta A$ where the $n^2 \times n^2$ matrix $S$ is

defined by $S_{ij,k} = a_{i,j}^k = (\boldsymbol{v}_j \cdot \boldsymbol{x}_i + b_j)^k$. This matrix $S$ is invertible because any $\boldsymbol{g}$ such that $S\boldsymbol{g} = 0$ would induce:

$$\forall i, j, \; \sum_{k=0}^{n^2-1} \delta\gamma_k \, a_{i,j}^k = 0 \tag{192}$$

and thus the polynomial $P(x) = \sum_{k=0}^{n^2-1} \delta\gamma_k \, x^k$ has at least $n^2$ roots while being of order at most $n^2 - 1$. Thus $S\boldsymbol{g} = 0 \implies \boldsymbol{g} = 0$ and $S$ is invertible. Note that as $\delta A$ is infinitesimal, $\boldsymbol{g} = S^{-1}\delta A$ will be infinitesimal as well, and so is the change brought to the activation function $\sigma$.

Consequently we have that the set of activation functions $\sigma$ and neuron parameters $(\boldsymbol{v}_j, b_j)$ for which the matrix $A$ is full rank is dense in the set of polynomials $\mathfrak{P}$ of order at least $n^2$ and of neuron parameters $\mathfrak{N}$.

Now, the function $\det : \mathfrak{P} \times \mathfrak{N} \to \mathbb{R}$, $((\gamma_k)_k, (\boldsymbol{v}_j, b_j)_j) \mapsto \det A = \det(\sigma_\gamma(\boldsymbol{v}_j \cdot \boldsymbol{x}_i + b_j))$ is smooth as a function of its input parameters (the determinant being a polynomial function of the matrix coefficients). As this continuous function is non-0 on a dense set of its inputs, the pre-image $\det^{-1}\{0\}$ is closed and contains no open subset. This is not yet sufficient to prove that this pre-image is of measure 0 (e.g., fat Cantor set).

For a fixed order $K$, one can see this function as a polynomial of its inputs $\gamma_k$ and $\boldsymbol{v}_j, b_j$, and conclude[2] that the set of its roots is of measure 0. As a consequence, the probability, over coefficients $\gamma_k$ or equivalently over polynomials $\sigma$ of order $K$, that $\det A$ is non-0, is 1. As this stands for all $K$, we have that the probability that the matrix $A$ is invertible is at least the mass of polynomials of all orders $K$, i.e. $\sum_{k \geqslant n^2} \frac{\alpha}{k^2} = 1$. Thus $A$ is invertible with probability 1. $\qquad\square$

# E  Technical details

## E.1  Batch size to estimate the new neuron and the best update

In this section we study the variance of the matrices $\boldsymbol{M}^*$ and $\boldsymbol{S}^{-1/2}\boldsymbol{N}$ computed using a minibatch of $n$ samples, seeing the samples as random variables, and the matrices computed as estimators of the true matrices one would obtain by considering the full distribution of samples. Those two matrices are the solutions of the multiple linear regression problems defined in (35) and in (46), as we are trying to regress the desired update noted $Y$ onto the span of the activities noted $X$. We suppose we have the following setting :

$$Y \sim \boldsymbol{A}X + \varepsilon, \; \varepsilon \sim \mathcal{N}(0, \sigma^2), \; \mathbb{E}[\varepsilon|X] = 0 \tag{193}$$

where the $(X_i, Y_i)$ are *i.i.d.* and $A$ is the oracle for $\boldsymbol{M}^*$ or matrix $\boldsymbol{S}^{-1/2}\boldsymbol{N}$. If $Y$ is multidimensional, the total variance of our estimator can be seen as the sum of the variances of the estimator on each dimension of $Y$.

We now suppose that $Y \in \mathbb{R}$ and note $\hat{A} := \boldsymbol{Y}\boldsymbol{X}^T(\mathbf{X}\mathbf{X}^T)^+$ the solution of 193. We first remark that $\hat{\boldsymbol{A}}\boldsymbol{X} = \boldsymbol{Y}\boldsymbol{P}$ with $\boldsymbol{P} = \boldsymbol{X}^T(\boldsymbol{X}\boldsymbol{X}^T)^+\boldsymbol{X} \in \mathbb{R}(n,n)$ . It follows that when $n \leqslant p$, almost surely we have $rk(\boldsymbol{P}) = n$ and $\boldsymbol{Y}\boldsymbol{P} = \boldsymbol{Y}$, resulting in a zero expressivity bottleneck for that specific mini-batch, *i.e.* $\boldsymbol{Y} = \hat{\boldsymbol{A}}\boldsymbol{X}$. In practice, we do not consider $n < p$ as the solution ($\delta\boldsymbol{W}$ or $\boldsymbol{A}, \boldsymbol{\Omega}$) would overfit a specific mini-batch and would increase the expressivity bottleneck for the rest of the dataset.

We now suppose that $n > p$, we now have almost surely that $rk(\boldsymbol{P}) = p$ and $\boldsymbol{Y}\boldsymbol{P} \neq \boldsymbol{Y}$. We now study the variance of the estimator $\hat{\boldsymbol{A}} \in \mathbb{R}^p$. We have almost surely that $\boldsymbol{X}\boldsymbol{X}^T$ is invertible and note $(\boldsymbol{X}\boldsymbol{X}^T)^{-1}$ its inverse. Taking the expectation on variable $\varepsilon$, we have $\text{cov}(\hat{\boldsymbol{A}}) = \sigma^2(\mathbf{X}\mathbf{X}^T)^{-1}$. If $n$ is large, and if matrix $\frac{1}{n}\mathbf{X}\mathbf{X}^T \to \boldsymbol{Q}$, with $\boldsymbol{Q}$ non-singular, then, asymptotically, we have $\hat{\boldsymbol{A}} \sim \mathcal{N}(\boldsymbol{A}, \sigma^2\frac{\boldsymbol{Q}^{-1}}{n})$, which is equivalent to $(\hat{\boldsymbol{A}} - \boldsymbol{A})\frac{\sqrt{n}}{\sigma}\boldsymbol{Q}^{1/2} \sim \mathcal{N}(0, I)$. Then $||(\hat{\boldsymbol{A}} - \boldsymbol{A})\frac{\sqrt{n}}{\sigma}\boldsymbol{Q}^{1/2}||^2 \sim \chi^2(k)$ where $k$ is the dimension of $X$. It follows that $\mathbb{E}\left[||(\hat{\boldsymbol{A}} - \boldsymbol{A})\boldsymbol{Q}^{1/2}||^2\right] = \frac{k\sigma^2}{n}$ and as $\boldsymbol{Q}^{1/2}\boldsymbol{Q}^{1/2}{}^T$ is positive definite, we conclude that $\text{var}(\hat{\boldsymbol{A}}) \leqslant \frac{k\sigma^2}{n\lambda_{\min}(\boldsymbol{Q})}$.

---

[2]See for instance a proof by recurrence that roots of a polynomial are always of measure 0: `https://math.stackexchange.com/questions/1920302/the-lebesgue-measure-of-zero-set-of-a-polynomial-function-is-zero` .

In practice we aim to keep the variance of our estimators stable during architecture growth. To ensure this we can choose the batch size $n$ to make the bound constant. With the notations defined in Figure 12, we estimate a matrix of size $k \leftarrow (SW)^2$. For $n$ images, as each input sample contains $P$ quantities, and that each is a realization of the random variable $\boldsymbol{X}$ (total $nP$ variables), we have in total $n \leftarrow nP$ data points for the estimation of the best neuron. Hence to add new neurons with a (asymptotically) fixed variance, we use batch size

$$n \propto \frac{(SW)^2}{P} \ .$$

For convolutional layers, we take $n = 0.001 \times \frac{(SW)^2}{P} \times 2^k$ (Figure 5) and $n = 0.01 \times \frac{(SW)^2}{P} \times 2^k$ (Figure 8), where $k$ is equal to $\sqrt{\frac{32 \times 32}{P}}$, and this $2^k$ factor is found empirically to somehow account for the variances of the estimators even when the same input is used multiple times, as are the $\{\boldsymbol{B}_{i,j}^t\}_{j \in P}$ in Equation (55).

### E.2 Batch size for learning

We adjust the batch size for gradient descent as follows: the batch size is set to $b_{t=0} = 32$ at the beginning of each experiment, and it is scheduled to increase as the square root of the complexity of the model (*i.e.* number of parameters). If at time $t$ the network has complexity $C_t$ parameters, then at time $t+1$ the training batch size is equal to $b_{t+1} = b_t \times \sqrt{\frac{C_{t+1}}{C_t}}$.

### E.3 Normalization

#### E.3.1 Figures 5, 6 and 16 : Usual normalization

For the GradMax method of Figures 5 and 16, before adding the new neurons to the architecture, we normalize the outgoing weight of the new neurons according to Evci et al. (2022), *i.e.* :

$$\alpha_k^* \leftarrow 0 \tag{194}$$

$$\text{for Figures 5 and 6} \quad \omega_k^* \leftarrow \omega_k^* \times \frac{10^{-3}}{\sqrt{||(\omega_j^*)_{j=1}^{n_d}||_2^2 / n_d}} \tag{195}$$

$$\text{for Figure 16} \quad \omega_k^* \leftarrow \omega_k^* \times \sqrt{\frac{10^{-3}}{||(\omega_j^*)_{j=1}^{n_d}||_2^2 / n_d}} \tag{196}$$

For TINY method of both figures, the previous normalization process is mimicked by normalizing the in and out going weights by their norms and multiplying them by $\sqrt{10^{-3}}$, *i.e.* :

$$\alpha_k \leftarrow \alpha_k^* \times \sqrt{\frac{10^{-3}}{||(\alpha_j^*)_{j=1}^{n_d}||_2^2 / n_d}} \tag{197}$$

$$\omega_k \leftarrow \omega_k^* \times \sqrt{\frac{10^{-3}}{||(\omega_j^*)_{j=1}^{n_d}||_2^2 / n_d}} \tag{198}$$

#### E.3.2 Figure 8 : Amplitude Factor

For the Random and the TINY methods of Figure 8, we first normalize the parameters as :

For the new neurons

$$\alpha_k^* \leftarrow \alpha_k^* \times \frac{1}{\sqrt{||(\alpha_j^*)_{j=1}^{n_d}||_2^2 / n_d}}$$

$$\omega_k^* \leftarrow \omega_k^* \times \frac{1}{\sqrt{||(\omega_j^*)_{j=1}^{n_d}||_2^2 / n_d}}$$

For the best update

$$\boldsymbol{W}^* \leftarrow \boldsymbol{W}^* \times \frac{1}{\sqrt{||\boldsymbol{W}^*||_2^2 / n_d}}$$

Then, we multiply them by the amplitude factor $\gamma^*$ :

For the new neurons :

$$\alpha_k^*, \ \omega_k^* \ \leftarrow \alpha_k^* \gamma^*, \omega_k^* \gamma^*$$

$$\gamma^* := \arg\min_{\gamma \in [-L, L]} \sum_i \mathcal{L}(f_{\theta \oplus \gamma \theta_\leftrightarrow^K}(\boldsymbol{x}_i), \boldsymbol{y}_i)$$

For the best update :

$$\boldsymbol{W}_l^* \ \leftarrow \gamma^* \boldsymbol{W}_l$$

$$\gamma^* := \arg\min_{\gamma \in [-L, L]} \sum_i \mathcal{L}(f_{\theta + \gamma \boldsymbol{W}^*}(\boldsymbol{x}_i), \boldsymbol{y}_i)$$

where the operation $\gamma \theta_\leftrightarrow^{K^*} = (\gamma \alpha_k^*, \gamma \omega_k^*)_k^K$ is the concatenation of the neural network with the new neurons and $\theta + \gamma \boldsymbol{W}^*$ is the update of one layer with its best update. The batch on which $\gamma^*$ is computed is different from the one used to estimate the new parameters and its size is fixed to 1000 for all experiments.

### E.4   Full algorithm

In this section we describe in detail the pseudo-code to plot Figures 5, 6 and 8. The function NewNeurons($l$), in Algorithm 2, computes the new neurons defined at Proposition 3.2 for layer $l$ sorted by decreasing eigenvalues. The function BestUpdate($l$), in Algorithm 4 computes the best update at Proposition 3.1 for layer $l$.

---

**Algorithm 1:** Algorithm to plot Figures 5 and 8.

**1** **for** *each method [TINY, MethodToCompareWith]* **do**
**2**     Start from neural network $N$ with initial structure $s \in \{1/4, 1/64\}$;
**3**     **while** *N architecture does not match ResNet18 width* **do**
**4**         **for** *d in {depths to grow}* **do**
**5**             $\theta_\leftrightarrow^{K^*} = \text{NewNeurons}(d, method)$ ;
**6**             Normalize $\theta_\leftrightarrow^{K^*}$ according to E.3;
**7**             Add the neurons at layer $d$ ;
**8**             Train $N$ for $\Delta t$ epochs ;
**9**             Save model $N$ and its performance ;
**10**         **end**
**11**     **end**
**12** **end**

---

---

**Algorithm 2:** NewNeurons

**Data:** $l$, method $= TINY$
**Result:** Best neurons at $l$
1 **if** $method = TINY$ **then**
2    $\boldsymbol{M} = \text{BestUpdate}(l+1)$;
3    $\boldsymbol{S}, \boldsymbol{N} = \text{MatrixSN}(l-1, l+1, \boldsymbol{M} = \boldsymbol{M})$;
4    Compute the SVD of $\boldsymbol{S} := \boldsymbol{O}\Sigma\boldsymbol{O}^T$;
5    Compute the SVD of
     $\boldsymbol{O}\sqrt{\Sigma}^{-1}\boldsymbol{O}\boldsymbol{N} := \boldsymbol{A}\Lambda\boldsymbol{\Omega}$;
6    Use the columns of $\boldsymbol{A}$, the lines of $\boldsymbol{\Omega}$ and the diagonal of $\Lambda$ to construct the new neurons of Prop. 3.2;
7 **else if** $method = GradMax$ **then**
8    $\boldsymbol{M} = None$ ;
9    $\_, \boldsymbol{N} = \text{MatrixSN}(l-1, l+1, \boldsymbol{M} = \boldsymbol{M})$ ;
10   Compute the SVD of $\boldsymbol{N}^T\boldsymbol{N}$ ;
11   Use the eigenvectors to define the new out-going weights ;
12   Set the new in-going weight to 0;
13 **else if** $method = Random$ **then**
14   $(\boldsymbol{\alpha}_k, \boldsymbol{\omega}_k)_{k=1}^{n_d} \sim \mathcal{N}(0, Id)$;
15 **end**

---

**Algorithm 3:** MatrixSN

**Data:** $p_1, p_2$ (layer indexes), $\boldsymbol{M} = None$
**Result:** Construct matrices $\boldsymbol{S}$ and $\boldsymbol{N}$
1 Take a minibatch $\mathbf{X}$ of size $\propto \frac{(SW)^2}{P}$;
2 Propagate and backpropagate $\mathbf{X}$;
3 Compute $\mathbf{V}_{goal}$ at $p_2$, i.e. $-\frac{\partial\mathcal{L}^{tot}}{\partial\mathbf{A_{p2}}}$;
4 **if** $\boldsymbol{M} \neq None$ **then**
5   $\mathbf{V}_{goal} -= \boldsymbol{M}\boldsymbol{B}_{p_1}$
6 **end**
7 $\mathbf{S}, \mathbf{N} = \boldsymbol{B}_{p_1}\boldsymbol{B}_{p_1}^T, \boldsymbol{B}_{p_1}\mathbf{V}_{goal}^T$;

---

**Algorithm 4:** BestUpdate

**Data:** $l$, index of a layer
**Result:** Best update at $l$
1 Take a minibatch $\mathbf{X}$ of size $\propto \frac{(SW)^2}{P}$;
2 Compute $(\mathbf{S}, \mathbf{N})$ with MatrixSN$(l, l)$;
3 $\boldsymbol{M} = \boldsymbol{N}^T\boldsymbol{S}^{-1}$;

---

### E.5 Computational complexity

We estimate here the computational complexity of the above algorithm for architecture growth.

**Theoretical estimate.** We use the following notations:

- number of layers: $L$

- layer width, or number of kernels if convolutions: $W$ (assuming for simplicity that all layers have same width or kernels)

- number of pixels in the image: $P$ ($P = 1$ for fully-connected)

- kernel filter size: $S$ ($S = 1$ if fully-connected)

- minibatch size used for standard gradient descent: $M$

- minibatch size used for new neuron estimation: $M'$

- minibatch size used in the line-search to estimate amplitude factor: $M''$

- number of classical gradients steps performed between 2 addition tentatives: $T$

Complexity, estimated as the number of basic operations, cumulated over all calls of the functions:

- of the standard training part: $TMLSW^2P$

- of the computation of matrices of interest (function MatrixSN): $LM'(SW)^2P$

- of SVD computations (function NewNeurons): $L(SW)^3$

- of line-searches (function AmplitudeFactor): $L^2M''SW^2P$

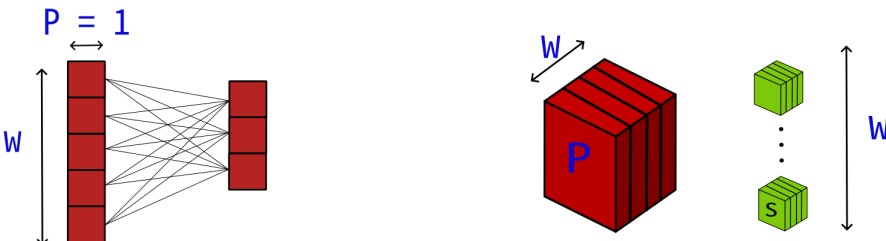

Figure 12: Notation and size for convolutional and linear layers

- of weight updates (function BestUpdate): $LSW$

The relative added complexity w.r.t. the standard training part is thus:

$$M'S/TM \;+\; S^2W/TMP \;+\; M''L/TM \;+\; 1/WTMP.$$

**SVD cost is negligible.** The relative cost of the SVD w.r.t. the standard training part is $S^2W/TMP$. In the fully-connected network case, $S = 1$, $P = 1$, and the relative cost of the SVD is then $W/TM$. It is then negligible, as layer width $W$ is usually much smaller than $TM$, which is typically $10 \times 100$ for instance. In the convolutional case, $S = 9$ for $3 \times 3$ kernels, and $P \approx 1000$ for CIFAR, $P \approx 100000$ for ImageNet, so the SVD cost is negligible as long as layer width $W \ll 10000$ or $1\,000\,000$ respectively. So one needs no worrying about SVD cost.

Likewise, the update of existing weights using the "optimal move" (already computed as a by-product) is computationally negligible, and the relative cost of the line searches is limited as long as the network is not extremely deep ($L < TM/M''$).

On the opposite, the estimation of the matrices (to which SVD is applied) can be more ressource-demanding. The factor $M'S/TM$ can be large if the minibatch size $M'$ needs to be large for statistical significance reasons. One can show that an upper bound to the value required for $M'$ to ensure estimator precision (see Appendix E.1) is $(SW)^2/P$. In that case, if $W > \sqrt{TMP/S^3}$, these matrix estimations will get costly. In the fully-connected network case, this means $W > \sqrt{TM} \approx 30$ for $T = 10$ and $M = 100$. In the convolutional case, this means $W > \sqrt{TMP/S^3} \approx 30$ for CIFAR and $\approx 300$ for ImageNet. We are working on finer variance estimation and on other types of estimators to decrease $M'$ and consequently this cost. Actually $(SW)^2/P$ is just an upper bound on the value required for $M'$, which might be much lower, depending on the rank of computed matrices.

**In practice.** In practice the cost of a full training with our architecture growth approach is similar (sometimes a bit faster, sometimes a bit slower) than a standard gradient descent training using the final architecture from scratch. This is great as the right comparison should take into account the number of different architectures to try in the classical neural architecture search approach. Therefore we get layer width hyper-optimization for free.

## F    Additional experimental results and remarks

### F.1    ResNet18 on CIFAR-100

**Figures.** In all plots the black line represents the average performance over two independent runs, and the colored regions indicate the confidence interval.

**Technical details of Figures 5 and 8** The experiments were performed on 1 GPU. The optimizer is SGD($lr = 1e-2$) with the starting batch size 32 E.2. At each depth $l$ we set the number $n_l$ of neurons to be

added at this depth 2. These numbers do not depend on the starting architecture and have been chosen such that each depth will reach its final width with the same number of layer extensions. For the initial structure $s = 1/4$, resp. $1/64$, we set the number of layer extensions to 16, resp. 21, such that at depth 2 (named Conv2 in Table 3), $n_2 = (\text{Size}_2^{final} - \text{Size}_2^{start})/\text{nb of layer extensions} = (64 - 16)/16 = (64 - 1)/21 = 3$. The initial architecture is described in Table 3.

| depth $l$ | Conv2 | Conv3 | Conv5 | Conv6 | Conv8 | Conv9 | Conv11 | Conv12 |
|---|---|---|---|---|---|---|---|---|
| $n_l$ | 3 | 3 | 6 | 6 | 12 | 12 | 24 | 24 |

Table 2: Number of neurons to add per layer. The depth is identified by its name on Table 3.

### F.1.1    Performance for gradient-based methods

We present in Table 5 two indicators of performance on classical visual datasets for four gradient-based methods: DART (Liu et al. (2019)), NORTH Maile et al. (2022), GradMax (Evci et al. (2022)), and TINY. We use the adjective *gradient-based* when the method uses the information from the propagation of the loss to search for an architecture. While GradMax, NORTH, and TINY increase the size of existing architectures (VGG and ResNet), DARTS uses cells to create the architecture, starting from a large graph of cells and removing what is considered unnecessary connections. We make the following remarks for the indicator *Time*, which is the time in GPU days to search for an architecture and train it, and the indicator *Acc.*, which is the accuracy on the test set:

- For NORTH, GradMax, and TINY, the accuracy on the test set is comparable, as all methods share a lot in terms of methodology, growth process, and neuron initialization (cf. B). Nonetheless, NORTH search time complexity is much larger as for each increase of architecture its strategy relies on trial-and-error methodology (going up to 1000 generations before actually increasing the architecture).

- The time search complexity of TINY will always be slightly higher than GradMax as it needs to project the desired update and compute the matrix $S$ (cf B).

- Although DARTS reaches good performance, its time search complexity is always greater than one GPU as it starts from a large architecture and decreases its size by removing connections between cells. For the ImageNet dataset, the indicator *Time* does not take into account the time spent searching for the basic cells, which are then used to create the graph.

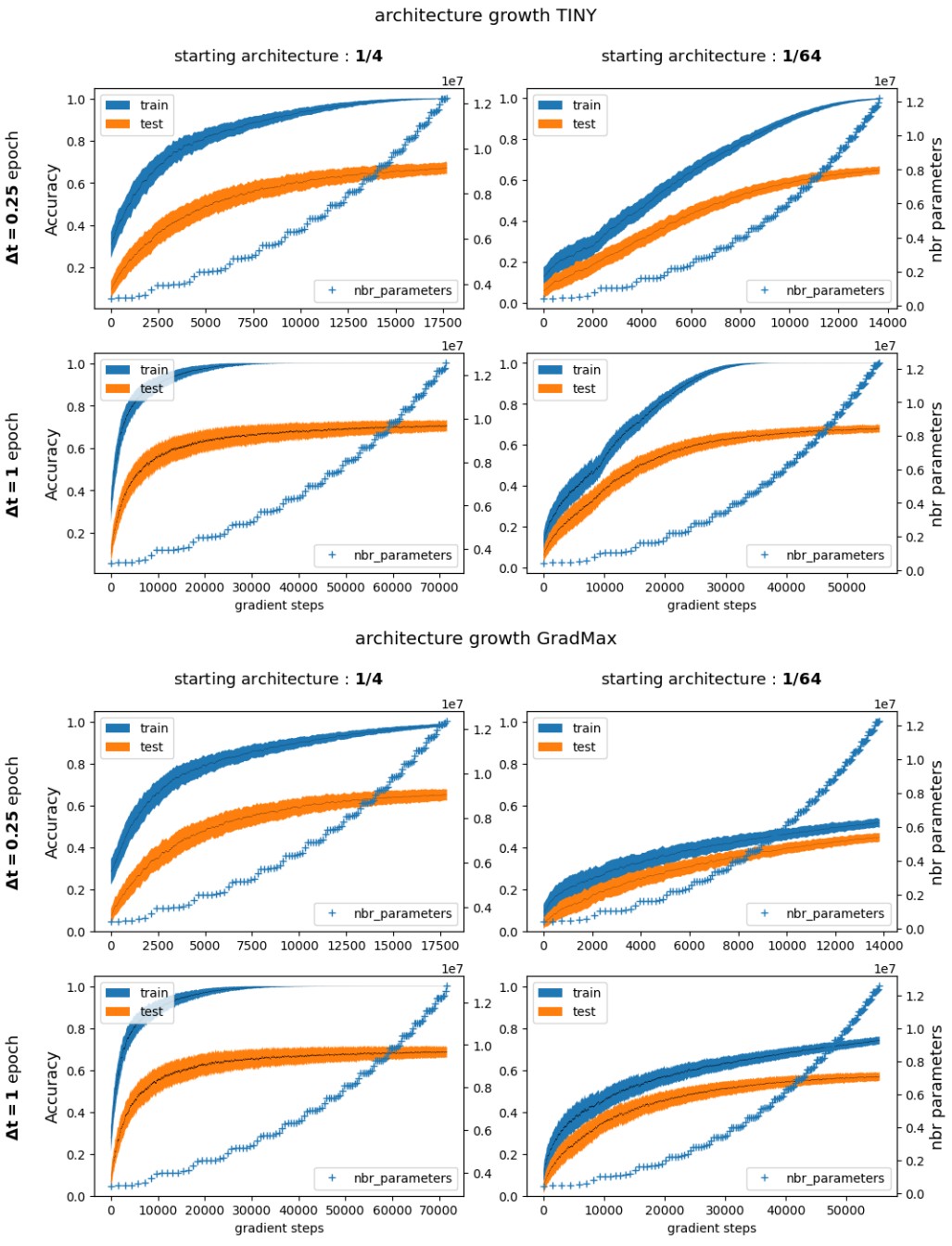

Figure 13: Accuracy and number of parameters during architecture growth for methods TINY and GradMax as a function of the gradient step. Mean and standard deviation for four independent runs.

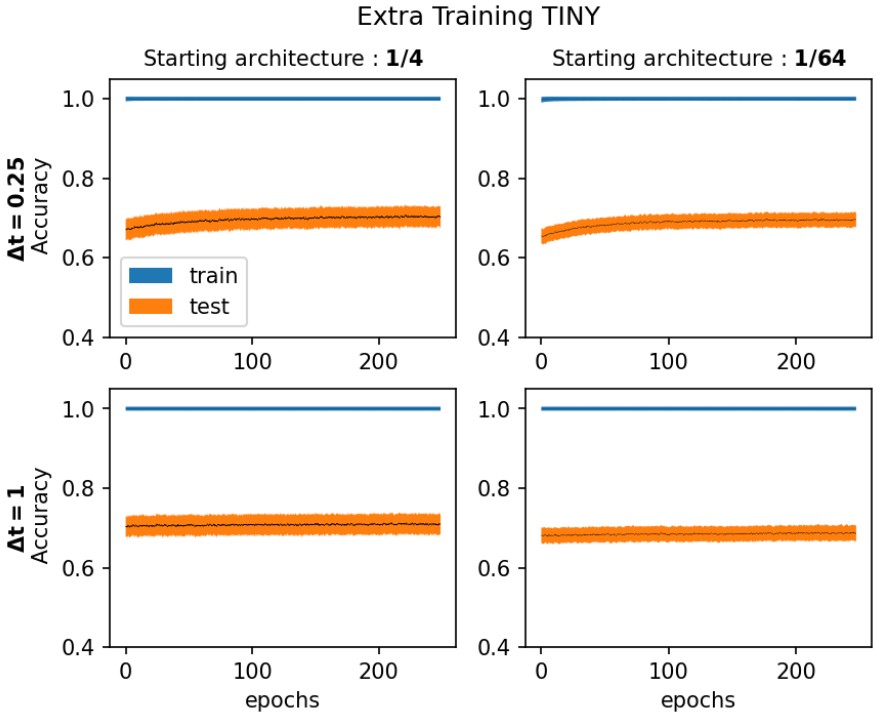

Figure 14: Accuracy as a function of the number of epochs during extra training for TINY on four independent runs.

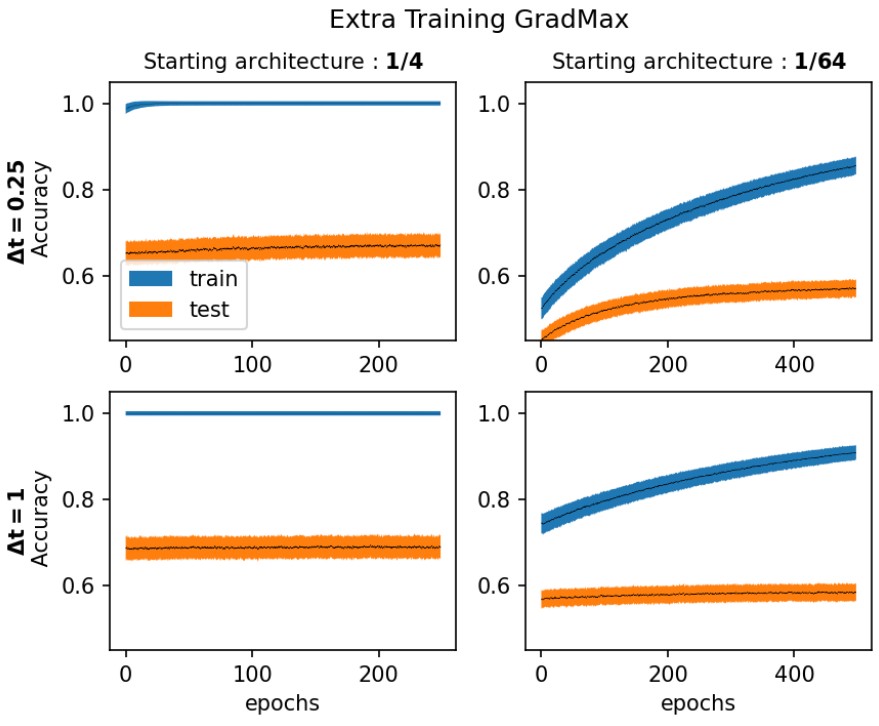

Figure 15: Accuracy curves as a function of the number of epochs during extra training for GradMax on four independent runs.

Table 3: Initial and final architecture for the models of Figure 5. Numbers in color indicate where the methods were allowed to add neurons (middle of ResNet blocks). In blue the initial structure for the model 1/64 and in green the initial structure for the model 1/4, i.e., 1|16 indicates that the model 1/64 started with 1 neuron at this layer while the model 1/4 started with 16 neurons at the same layer.

| ResNet18 | | | |
|---|---|---|---|
| Layer name | Output size | Initial layers (kernel=(3,3), padd.=1) | Final layers (end of Fig 5) |
| Conv 1 | $32 \times 32 \times 64$ | $\begin{bmatrix} 3 \times 3, \end{bmatrix}$ | $\begin{bmatrix} 3 \times 3, 64 \end{bmatrix}$ |
| Conv 2 | $32 \times 32 \times 64$ | $\begin{bmatrix} 3 \times 3, 64 \\ 3 \times 3, 1\|16 \end{bmatrix} \begin{bmatrix} 3 \times 3, 1\|16 \\ 3 \times 3, 64 \end{bmatrix}$ | $\begin{bmatrix} 3 \times 3, 64 \\ 3 \times 3, 64 \end{bmatrix} \begin{bmatrix} 3 \times 3, 64 \\ 3 \times 3, 64 \end{bmatrix}$ |
| Conv 3 | $32 \times 32 \times 64$ | $\begin{bmatrix} 3 \times 3, 64 \\ 3 \times 3, 1\|16 \end{bmatrix} \begin{bmatrix} 3 \times 3, 1\|16 \\ 3 \times 3, 64 \end{bmatrix}$ | $\begin{bmatrix} 3 \times 3, 64 \\ 3 \times 3, 64 \end{bmatrix} \begin{bmatrix} 3 \times 3, 64 \\ 3 \times 3, 64 \end{bmatrix}$ |
| Conv 4 | $16 \times 16 \times 64$ | $\begin{bmatrix} 3 \times 3, 128 \end{bmatrix}$ | $\begin{bmatrix} 3 \times 3, 128 \end{bmatrix}$ |
| Conv 5 | $16 \times 16 \times 128$ | $\begin{bmatrix} 3 \times 3, 128 \\ 3 \times 3, 2\|32 \end{bmatrix} \begin{bmatrix} 3 \times 3, 2\|32 \\ 3 \times 3, 128 \end{bmatrix}$ | $\begin{bmatrix} 3 \times 3, 128 \\ 3 \times 3, 128 \end{bmatrix} \begin{bmatrix} 3 \times 3, 128 \\ 3 \times 3, 128 \end{bmatrix}$ |
| Conv 6 | $16 \times 16 \times 128$ | $\begin{bmatrix} 3 \times 3, 128 \\ 3 \times 3, 2\|32 \end{bmatrix} \begin{bmatrix} 3 \times 3, 2\|32 \\ 3 \times 3, 128 \end{bmatrix}$ | $\begin{bmatrix} 3 \times 3, 128 \\ 3 \times 3, 128 \end{bmatrix} \begin{bmatrix} 3 \times 3, 128 \\ 3 \times 3, 128 \end{bmatrix}$ |
| Conv 7 | $8 \times 8 \times 256$ | $\begin{bmatrix} 3 \times 3, 256 \end{bmatrix}$ | $\begin{bmatrix} 3 \times 3, 256 \end{bmatrix}$ |
| Conv 8 | $8 \times 8 \times 256$ | $\begin{bmatrix} 3 \times 3, 256 \\ 3 \times 3, 4\|64 \end{bmatrix} \begin{bmatrix} 3 \times 3, 4\|64 \\ 3 \times 3, 256 \end{bmatrix}$ | $\begin{bmatrix} 3 \times 3, 256 \\ 3 \times 3, 256 \end{bmatrix} \begin{bmatrix} 3 \times 3, 256 \\ 3 \times 3, 256 \end{bmatrix}$ |
| Conv 9 | $8 \times 8 \times 256$ | $\begin{bmatrix} 3 \times 3, 256 \\ 3 \times 3, 4\|64 \end{bmatrix} \begin{bmatrix} 3 \times 3, 4\|64 \\ 3 \times 3, 256 \end{bmatrix}$ | $\begin{bmatrix} 3 \times 3, 256 \\ 3 \times 3, 256 \end{bmatrix} \begin{bmatrix} 3 \times 3, 256 \\ 3 \times 3, 256 \end{bmatrix}$ |
| Conv 10 | $4 \times 4 \times 512$ | $\begin{bmatrix} 3 \times 3, 512 \end{bmatrix}$ | $\begin{bmatrix} 3 \times 3, 512 \end{bmatrix}$ |
| Conv 11 | $4 \times 4 \times 512$ | $\begin{bmatrix} 3 \times 3, 512 \\ 3 \times 3, 8\|128 \end{bmatrix} \begin{bmatrix} 3 \times 3, 8\|128 \\ 3 \times 3, 512 \end{bmatrix}$ | $\begin{bmatrix} 3 \times 3, 512 \\ 3 \times 3, 512 \end{bmatrix} \begin{bmatrix} 3 \times 3, 512 \\ 3 \times 3, 512 \end{bmatrix}$ |
| Conv 12 | $4 \times 4 \times 512$ | $\begin{bmatrix} 3 \times 3, 512 \\ 3 \times 3, 8\|128 \end{bmatrix} \begin{bmatrix} 3 \times 3, 8\|128 \\ 3 \times 3, 512 \end{bmatrix}$ | $\begin{bmatrix} 3 \times 3, 512 \\ 3 \times 3, 512 \end{bmatrix} \begin{bmatrix} 3 \times 3, 512 \\ 3 \times 3, 512 \end{bmatrix}$ |
| AvgPool2d | $1 \times 1 \times 512$ | | |
| FC 1 | 100 | $512 \times 100$ | $512 \times 100$ |
| SoftMax | 100 | | |

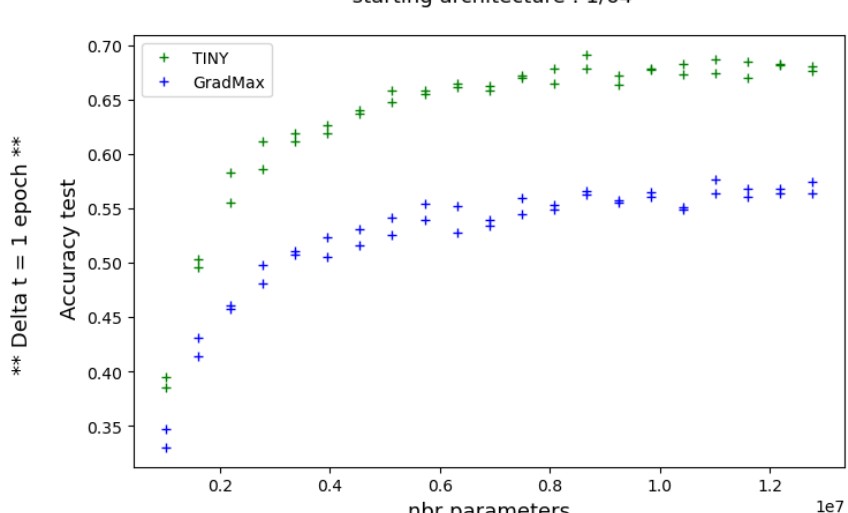

Figure 16: Accuracy on test split of as a function of the number of parameters during architecture growth from ResNet$_{1/64}$ to ResNet18. **The normalization for GradMax is $\sqrt{10^{-3}}$.**

|  | $\Delta t = 1$ | | Baseline |
|---|---|---|---|
|  | TINY | GradMax |  |
| s = 1/64 | $68.1 \pm 0.5$ $68.7 \pm 0.6\,^{5*}$ | $57.2 \pm 0.3$ $57.7 \pm 0.3\,^{3*}$ | $72.8 \pm 0.3\,^{5*}$ |

Table 4: Final accuracy on test split of ResNet18 of 16 after the architecture growth (*grey*) and after convergence (*blue*). The number of stars indicates the multiple of 50 epochs needed to achieve convergence. With the starting architecture ResNet$_{1/64}$ and $\Delta t = 1$ the method TINY achieves $68.1 \pm 0.5$ on test split after its growth and it reaches $68.7 \pm 0.6\,^{5*}$after $* := 5 \times 50$ epochs.

| Method | | Indicators | | | Dataset |
|---|---|---|---|---|---|
| | | Arch. | Time | Acc. | |
| DARTS | | $\times$ | 4 | $97.0 \pm 0.1$ | CIFAR-10 |
| | | $\times$ | 6.5 | $97.8 \pm 0.1$ | |
| | | $\times$ | 4 | 73.3 | ImageNet |
| NORTH | | VGG-11 | 3.3 | $\sim 76.4$ | CIFAR-10 |
| | | ResNet-28 | 0.4 | $\sim 83.3$ | |
| | | VGG-11 | 1.6 | $\sim 55.7$ | CIFAR-100 |
| GradMax$^{\ddagger}$ | $\Delta t = 0.25, s = 1/64$ | ResNet-18 | 0.12 | $45.0 \pm 0.4$ | CIFAR-100 |
| | $\Delta t = 1, s = 1/64$ | ResNet-18 | 0.26 | $56.8 \pm 0.2$ | |
| TINY$^{\ddagger}$ | $\Delta t = 0.25, s = 1/64$ | ResNet-18 | 0.13 | $65.8 \pm 0.1$ | |
| | $\Delta t = 0.1, s = 1/64$ | ResNet-18 | 0.28 | $68.1 \pm 0.5$ | |

Table 5: *Time*: GPU days spent to search for the architecture and to train it. *Acc.*: accuracy on test set (%). $\ddagger$ : estimated with our implementation. For the NORTH method, we computed the average performance over their different initializations and strategies based on their Figure 4.

