# OpenReview forum: "Growing Tiny Networks: Spotting Expressivity Bottlenecks and Fixing Them Optimally"
_TMLR — Accepted by TMLR_

### Review · Reviewer_pEe5 · 2024-07-30

**Summary Of Contributions:**

The paper proposes a novel method of growing neural networks. The method aims to add parameters to a network so as to move the network in the direction of the functional gradient. The authors demonstrate theoretically that their method can eventually interpolate the training set with sufficiently many parameters added. Experiments demonstrate that the proposed method outperforms a baseline on CIFAR-100.

**Audience:**

Yes

**Broader Impact Concerns:**

No broader impact concerns

**Claims And Evidence:**

Yes

**Requested Changes:**

**Would strengthen**
- Please add another dataset such as ImageNet
- Please compare with an additional baseline method
- In some experiments, mean and std are only computed over two runs; please increase to at least four
- Please use booktabs style tables

**Strengths And Weaknesses:**

**Strengths**

Despite the existence of prior work on growing neural network, the paper's proposed method is novel. Notably, the theoretical analysis of the method is comprehensive, and one of the paper's biggest strengths. The experiments are also thorough and adequate to validate the theory.

The paper is quite well written. Concepts are gradually presented and well motivated, and explanations and notations are thorough. The authors make great use of text coloring throughout the paper.

**Weaknesses**

In my view, the paper has few weaknesses. The main one in my view is that the experiments only test on one dataset, CIFAR-100. I would encourage the authors to include results on a more complex dataset as well, such as ImageNet. It would also be helpful to include an additional baseline, in addition to GradMax.

Minor comments:
- In some experiments, mean and std are only computed over two runs; please increase to at least four
- Please use booktabs style tables

---

> ### Author Response · Authors · 2024-09-25
>
> - To increase statistical consistency, we increased all experiments to 4 independent runs. We also changed the graphical style by plotting the mean and the standard deviation instead of the curve for each individual run. Figures 6 and 7 are thus updated accordingly.
> - As a supplementary baseline, we used another method in Figure 8 by initializing both the incoming and outgoing weights of the new neurons with a uniform random variable. The conclusion for that method is the same as for the Gaussian initialization: the independence between the desired update and the suggested update induces only small spurious correlations between these two quantities, and therefore yields just a small decrease of loss. We refer also to Proposition 3.3 (which has been reformulated) when performing only the addition of neurons:
> \begin{align*}
> \mathcal{L}(f_{\theta \oplus \theta_{\leftrightarrow}^K}) &=  \mathcal{L}(f_{\theta})- \gamma \sigma_{l-1}'(0)\Delta_{\theta_{\leftrightarrow}^{K, \*}}  + o(\gamma)
> \end{align*}
> with
> \begin{align*}
>     \Delta_{\theta_{\leftrightarrow}^{K, \*}} &:=\frac{1}{n}\left\langle  {v_{goal_{proj}}^l},\; v^{l}(\theta_{\leftrightarrow}^{K, \*}) \right\rangle_{Tr} = \sum_{k=1}^K\lambda_k^2
> \end{align*}
> We remark that a decorrelation between  ${v_{goal_{proj}}}$ and $v^l$ induces a small $\Delta_{\theta_{\leftrightarrow}^{K, *}}$ and consequently a very small decrease in loss. In practice in Figure 8 our method reaches about 10 times better accuracy.
> - We changed the table style booktabs as recommended.
> - As the work is already validated on CIFAR-100, we do not expect a tremendous difference on ImageNet. We also have results on MNIST and CIFAR-10, that we have not mentioned in the paper, as they are smaller datasets and yield similar conclusions as for CIFAR-100. They can be found in the supplementary materials DEMO/CL and DEMO/MLP. While we agree that validating with MNIST only would not have been sufficient, we think that CIFAR-100 is complex enough and that reaching the same accuracy as a standard ResNet on CIFAR-100 is a sufficient proof of concept.
> Also, scaling to ImageNet means (at first glance) multiplying all training times by 10 and/or moving to different hardware (GPUs with more memory, and parallelizing over GPUs). We do reserve some time for this but rather for the next paper. Indeed, experiments will be more interesting and enlightening when we will have neuron addition strategies and when we will be able to develop any computational graph (that is, add layers as well). This is work in progress and we will definitely test on ImageNet then, as well as other kinds of tasks, to see which type of architecture appears.

---

### Review · Reviewer_jtZt · 2024-08-20

**Summary Of Contributions:**

This paper studies how to grow neural networks by adding neurons.  The paper first formalizes the notion of an 'expressivity bottleneck' -- a layer that is currently not expressive enough to represent the desired function.  It then proposes a method for adding neurons so as to fix expressivity bottlenecks.  The paper proves that an instantiation of this method provably yields global convergence of the training objective   Finally, it is shown that the algorithm improves over the baseline GradMax in the narrow-width regime.

**Audience:**

Yes

**Claims And Evidence:**

Yes

**Requested Changes:**

please see questions above

**Strengths And Weaknesses:**

Strengths:
 - The way of formulating expressivity bottlenecks, based on the functional gradient, is an interesting perspective and could potentially be useful for other works studying related questions.
 - The proposed method outperforms the baseline of GradMax.

Questions:
 - In practice with neural networks, how often is there a nonzero expressivity bottleneck?  If we have a  hidden layer with dimension $h$, and there are $n$ data points in a minibatch, then the functional gradient is just a vector of size $h n$, and if the layer has more than this number of parameters, shouldn't there be zero expressivity bottleneck?  If my math is wrong, is there a different dimension-counting argument that one could use to reason about when we expect to see an expressivity bottleneck?
 - The paper needs to more clearly state, in the main paper (rather than the appendix), what is the overall algorithm being used to train the network.   Section 3.2 tells us how to add a specified number of neurons to a specified layer, but it is left unclear how to leverage this to train a network.  Do we keep looping over the layers, from first to last, over and over again?  How do we know how many neurons to add to each layer?  In addition to pseudocode in the appendix, there should be a description of the overall training algorithm in English in the main paper.
 - in section 3.2, given that $\delta \mathbf{W}$ is ultimately taken to be the best move of the already-existing parameters, is there a good  reason to initially say that your goal is to _jointly_ optimize  $\delta \mathbf{W}$ and $\theta^K_\leftrightarrow$, as in equations (6), (7)?
 - I did not understand section 5.2, starting with the first sentence: "in this section, we focus on the impact of the new neurons' initialization".  What is meant by "initialization"?  Aren't we tuning the neurons as we add them?

Minor issues:
  - the second-to-last paragraph on page 2, beginning with "In this article", is very confusing to read.  One improvement would be to take away "aim at" => "In this article, we measure lack of expressivity...".  But the sentence should probably be split up into several smaller ones.
 - top of page 3 has a typo - "enables the localize expressivity bottlenecks"
 - in page 4, it would be good to state explicitly that $T_\mathcal{A}^{f_\theta}$ is a set of functions.  In general, it would be better throughout this section to consistently clarify the 'types' of all the symbols, provided that this can be done without using excessive notation.
 - in page 5, "the solution of 4" should be "the solution of (4)"

Suggestions (feel free to ignore):
 - I think the paper would be considerably improved if section 1 were split into separate introduction and related work sections.  The paper currently spends a lot of time talking about loosely related prior works before it mentions its actual contributions.

---

> ### Author Response · Authors · 2024-09-25
>
> - There is indeed such a setting where zero expressivity bottlenecks can be encountered, as a layer with sufficiently many parameters is able to overfit any quantity for few samples. In that case, there is no need for adding neurons, and this is coherent with an absence of expressivity bottleneck.  This situation may be encountered if the target function ($v_{goal}$) is very simple (e.g., linear), or if the function is provided on too few samples to realize its actual complexity, in which case looking at more samples would make the expressivity bottleneck appear again. Ideally, one should consider the whole dataset to compute the expressivity bottleneck, but, in the same spirit as stochastic gradient descent, we estimate this quantity on a large enough minibatch to save computational power. The quality of this estimator depends on the minibatch size, and this dependency is studied in Appendix E.1. Thus we can choose the batch size so as to be sure that expressivity bottlenecks are correctly estimated. In the end, the minibatch size is set to be always larger than the layer size, on purpose (unless the layer width exceeds the dataset size, which would not be "frugal"), and therefore expressivity bottlenecks are never 0 just for dimensional reasons.
>  Technically, starting from Eq. (5) and considering the expressivity bottleneck at layer $l$, a minibatch of size $n$, post-activities $A\in\mathbb{R}^{n, p}$, and desired update $v_{goal} \in \mathbb{R}^{n, q}$, Equation (5) can be written as: $$\min_{\delta w} \,|| A\,\delta w - v_{goal}||^2 .$$ Either we suppose that the matrix $A^TA$ is full rank, ie $rk(A^TA) = p$, either we note $p = rk(A^TA)$. If $n < p$ then, it exists $\delta\hat{w}$ such that $v_{goal} = A \,\delta \hat{w}$, resulting in a zero expressivity bottleneck. However, the batch on which we estimate the expressivity bottleneck is always sufficient to avoid this issue, as explained in Appendix E.1. We have added this remark to Appendix E.1 and have changed the experimental part to point towards correct appendix part.
> - This remark is partly shared with reviewer VRpd; we think that designing intelligent strategies for finding an optimal architecture (such as number of neurons to add) needs further studies, and is out of the scope of this article. We updated the abstract and the claim in that spirit and to reflect the conducted experiments. Regarding the strategy to add neurons, we used a naive strategy by looping through all depths until the final size of ResNet-18 is reached. The number of neurons to add at each layer is constant throughout time and is indicated in Table 2 of Appendix F.1. This strategy is used in the original paper GradMax, and we employed it to provide a fair comparison with their method. To keep the article length within the TMLR standard, the algorithms are detailed in the appendices.
> - Whether we should optimize a joint problem on $\delta W$ and $\theta^{K}_{ \leftrightarrow}$ or whether we should solve a sequence of 2 problems independently is debated in Appendix C.4. In a nutshell, the joint problem might provide better solutions, but the two-step process more intuitively relates to the spirit of improving upon a standard gradient descent: we aim at adding neurons that complement what the other ones have already done. To detail this point in the paper, we added a reference to appendix C.4 just after Eq. (7) and (8). If you think it would improve clarity, we could skip the joint optimization problem in the main paper and directly state the 2-step one.
>
> - Regarding the impact of the new neurons' initialization: we wish to check whether our way to add new neurons (following Proposition 3.2 + line search) significantly improves over just adding random neurons (+ line search). The results confirm that our "initialization" (way to set new neurons) is much better than adding random neurons. The reason for the line search is detailed as follows. In Section 5.1, the amplitude of the neurons is set to $\gamma =$ 1e-3 and the whole architecture is trained between each increase of architecture ($\Delta t \ne 0$). Thus, the weights of the new neurons are adjusted by gradient descent during this training period. If there is no training ($\Delta t = 0$ as in part 5.2), once the new neurons are added, their associated weights are not changed until the end of the experiment. For that reason, we choose the right amplitude $\gamma$ before adding them to the structure. A good initialization, i.e., good directions, will result in large $\gamma$ and a significant improvement in performance, while bad initializations yield poor optimization landscapes over the proposed directions for the line search, and consequently very small amplitude factors $\gamma$.
>
> - We thank the reviewer for noting the typos; we corrected them all.

---

> > ### Comment · Reviewer_jtZt · 2024-10-21
> > **response**
> >
> > Thanks for the authors' time.  I do think that the paper should skip the discussion of the joint problem, and just present the two step solution.  You could say "see appendix for a discussion of a joint approach."

---

### Review · Reviewer_VRpd · 2024-09-13

**Summary Of Contributions:**

The paper proposes a new method for growing neural networks. The proposed method, TINY, includes a mathematical definition of expressivity bottlenecks, which enables an optimal solution by adding neurons. The authors take a functional perspective of training neural networks which allows them to present a mathematical definition of expressivity bottlenecks. The expressivity bottleneck (shown in Figure 2) is the L2 norm of the parametric gradient $\mathcal{v}(x) = -(\nabla_\theta \mathcal{L}(f_\theta))(x)$ minus the functional gradient $\mathcal{v}\_{\text{goal}}(x) = -(\nabla_f \mathcal{L}(f))(x)$. On CIFAR-100, the proposed method achieves better performance and converges faster (when starting with smaller architectures) than GradMax (Evci et al., 2022), a baseline for growing neural networks most comparable with TINY.

Utku Evci, Bart van Merrienboer, Thomas Unterthiner, Fabian Pedregosa, and Max Vladymyrov. GradMax: Growing Neural Networks using Gradient Information. In *International Conference on Learning Representations* (ICLR), 2022.

**Audience:**

Yes

**Claims And Evidence:**

Yes

**Requested Changes:**

* See weakness.
* It would be nice to see a comparison with Xavier initialization (Glorot and Bengio, 2010) in Figure 8. Also should $d$ in the Gaussian distribution for initializing new neurons be a subscript? This seems like a large variance to be initializing neurons with.

Xavier Glorot and Yoshua Bengio. Understanding the difficulty of training deep feedforward neural networks. In *Proceedings of the Thirteenth International Conference on Artificial Intelligence and Statistics (AISTATS)*, 2010.

**Strengths And Weaknesses:**

Strengths:
* The paper is well written. The paper does a good job of explaining the difference between $\mathcal{v}$ and $\mathcal{v}_{\text{goal}}$ (shown in Figure 2).
* The paper includes appendices for details like functional gradient descent, parametric gradient descent, expressivity bottleneck examples, and extensions of their proposed method to convolutional layers.

Weaknesses:
* The experiments do not do a good job of demonstrating the method. The paper talks about being able to adapt architecture on the fly during training and the importance of searching architecture hyperparameters but in the experiments TINY is used to grow a ResNet-18 from 1/4 and 1/64 the width to the full size. The experiments don’t demonstrate the ability for the method to find an optimal architecture.
* In the abstract, introduction, and contributions, the authors talk about the importance of training time and computational resources. However, there is no comparison of training times or computational resources in the main paper.
* Another weakness related to training time and computational resources is the lack of discussion of alternative methods. Using TINY to grow a ResNet-18 achieves similar performance to a ResNet-18 trained from scratch on CIFAR-100 but practitioners with limited compute looking to achieve high performance on CIFAR-100 would likely reach for a pre-trained model and use methods from transfer learning like linear probing or fine-tuning to achieve high performance at little cost.

---

> ### Author Response · Authors · 2024-09-25
>
> - We agree with the reviewer that the abstract should better reflect the experimental part. Hence, we modified the abstract and the last claim to clarify this point. The proposed experimental part is now consistent with the claims, as it shows the effectiveness of our expressivity bottleneck mitigation. Designing intelligent strategies for finding an optimal architecture needs further studies, and is out of the scope of this article.
> - Computational time is indeed important in our approach; usual NAS search methods take usually more than one GPU day for  visual tasks as complex as CIFAR-100, while TINY as well as GradMax take a few hours to solve that problem. It is also important to note that we grow the network at the same time as we train it, whereas GradMax needs extra training as shown on Fig. 7. We mentionned this aspect in Section 5.1 and we refer the reader to more complete analysis in Table 5 (in appendix) which compares computation times for relevant existing algorithms.
> - Transfer learning is indeed a common and efficient approach which requires little computational time for fine-tuning. However it is not available when facing new types of data or tasks. Furthermore, pre-trained models are typically pretty large, since they need to be generic-purpose enough, and as a consequence the fine-tuned models also consume computational power at test time. On the opposite, our method aims at developping small architectures tailored to the tasks, in order to consume less at test time. We added this remark to the conclusion.
>
> - The Glorot initialization differs from the Gaussian one only by a variance factor, which depends on the previous and next layers sizes. This factor disappears when performing a line search over amplitude factors (cf. Figure 8). Indeed, the search for the best amplitude factor is equivalent to finding the best variance scaling for the new neurons. As a consequence, in practice our initialization strategy is equivalent to the Glorot initialization. We added this remark in Section 5.2.

---

### Author Response · Authors · 2024-09-25

First, we would like to thank the reviewers for their work and relevant questions. We uploaded a new version of the paper where the changes are notified in red. (As the reviewers'  questions were about different aspects of the paper, we responded to each individually and do not make any general comment.)

---

### Decision · Action_Editor_UcFX · 2024-10-28

**Recommendation:** Accept as is

**Comment:**

Reviewers were unanimous in their recommendation that the paper be accepted.  Certain critiques were raised, and the authors addressed many of them during the discussion.  One point which remains is the breadth of experimentation---the committee felt that for a paper claiming to contribute to computational efficiency of neural network training, larger scale experiments are in order.  I encourage the authors to pursue this route in follow up work.

**Audience:**

The topic would definitely be of interest to some of TMLR's readership

**Claims And Evidence:**

Claims in the paper are supported by theory and experiments, though there is room for broadening the latter (see below)